# Mitigating Instability in High Residual Adaptive Sampling for PINNs via Langevin Dynamics

**Minseok Jeong**[*][†]  **Giup Seo**[*]  **Euiseok Hwang**[‡]
Electrical Engineering and Computer Science
Gwangju Institute of Science and Technology (GIST)
dsa950115@kaist.ac.kr, guseo1120@gm.gist.ac.kr, euiseokh@gist.ac.kr

## Abstract

Recently, physics-informed neural networks (PINNs) have gained attention in the scientific community for their potential to solve partial differential equations (PDEs). However, they face challenges related to resource efficiency and slow convergence. Adaptive sampling methods, which prioritize collocation points with high residuals, improve both efficiency and accuracy. However, these methods often neglect points with medium or low residuals, which can affect stability as the complexity of the model increases. In this paper, we investigate this limitation and show that high residual-based approaches require stricter learning rate bounds to ensure stability. To address this, we propose a Langevin dynamics-based Adaptive Sampling (LAS) framework that is robust to various learning rates and model complexities. Our experiments demonstrate that the proposed method outperforms existing approaches in terms of relative $L^2$ error, and stability across a range of environments, including high-dimensional PDEs where Monte Carlo integration-based methods typically suffer from instability. The implementation code is publicly available at https://github.com/neurips2025-las/LAS-implementation.

## 1 Introduction

Partial differential equations (PDEs) describe a wide range of physical phenomena, including heat transfer [17, 8], fluid flow [37, 18, 33], wave propagation [34, 5], optics, and epidemiology [26, 36]. Accurate and efficient PDE solutions are vital across many industries. With recent advances in deep learning, physics-informed neural networks (PINNs) have emerged as a promising approach for solving PDEs. PINNs train by minimizing errors from initial conditions (IC), boundary conditions (BC), and PDE residuals at collocation points [31, 49, 12, 41, 23]. These error terms are treated as soft constraints, guiding the model to satisfy essential physical requirements.

This collocation-based learning method enhances the capability of PINNs by reducing the need for extensive experimental data collection across spatio-temporal ranges, demonstrating success in various industries as a promising alternative to traditional numerical methods such as the finite difference method and the finite element method [50, 2, 25]. However, collocation-based PINN (hereafter referred to as PINNs) encounter challenges in efficiently setting collocation points within the constraints of a limited sampling budget and in achieving fast convergence to accurate solutions. A key challenge arises from the presence of small regions with abrupt changes, in contrast to larger, smoother regions. This issue is particularly evident in stiff PDEs, which are often characterized by discontinuities, such as sudden transitions or jumps across the spatio-temporal domain.

---

[*]Equal contribution.
[†]Work done while at GIST. Now at KAIST, ACSS.
[‡]Corresponding author.

39th Conference on Neural Information Processing Systems (NeurIPS 2025).

Adaptive sampling in PINNs mainly follows two strategies: residual distribution-based methods, which resample points proportionally to residual magnitudes, and high residual-based methods, which prioritize large-residual points while neglecting low-residual regions. The latter often yields strong empirical results but risks distorting the residual distribution. This raises a key question: *should adaptive sampling focus exclusively on high residuals?* Addressing this question requires a careful analysis of the trade-offs and risks inherent in different sampling schemes.

In this work, we respond by proposing a principled alternative: Langevin dynamics-based adaptive sampling (LAS). Through a series of theoretical and empirical analyses, we demonstrate that LAS consistently achieves reliable relative $L^2$ errors and stable convergence across diverse PDEs, architectures, and hyperparameters, outperforming existing methods in robustness and scalability.

## 2 Background and Related Work

**Physics-informed neural networks.** The basic PINN framework [35] utilizes deep neural networks as function approximators $f_\theta$ to estimate the solution $u$ of a non-linear PDE:

$$u_t + \mathcal{N}_x[u] = 0, \quad x \in \mathcal{X} \subset \mathbb{R}^d, \quad t \in [0, T]; \tag{2.1}$$

$$u(x, 0) = h(x), \quad x \in \mathcal{X} \subset \mathbb{R}^d; \tag{2.2}$$

$$u(x, t) = g(x, t), \quad x \in \partial\mathcal{X} \subset \mathbb{R}^d, \quad t \in [0, T], \tag{2.3}$$

where $u(x, t)$ denotes the hidden solution at spatial and temporal coordinates $x, t$, $\mathcal{N}_x[\cdot]$ is the non-linear differential operator, $\mathcal{X}$ is the spatial domain, $\partial\mathcal{X}$ is the boundary, and $T$ is the time range. The spatio-temporal domain is $\Omega = \mathcal{X} \times [0, T]$, with collocation point $\mathbf{x} = (x, t) \in \Omega$. The PDE residuals $\mathcal{R}_\theta(\mathbf{x})$ and loss function on collocation points $\mathcal{P} = \{\mathbf{x}_n\}_{n=1}^{N_{\text{pde}}} \subset \Omega$ are calculated as:

$$\mathcal{R}_\theta(\mathbf{x}) = \frac{\partial}{\partial t} f_\theta(\mathbf{x}) + \mathcal{N}_x[f_\theta](\mathbf{x}), \mathbf{x} \in \Omega; \tag{2.4}$$

$$\mathcal{L}_{\text{pde}}(\{\mathbf{x}_n\}; \theta) = \mathbb{E}_{\mathbf{x} \sim \mathcal{U}(\Omega)} |\mathcal{R}_\theta(\mathbf{x})|^k \approx \frac{1}{N_{\text{pde}}} \sum_{n=1}^{N_{\text{pde}}} |\mathcal{R}_\theta(\mathbf{x}_n)|^k, \tag{2.5}$$

where $\mathcal{U}(\Omega)$ is the uniform distribution over $\Omega$ and $N_{\text{pde}}$ represents the number of sample points of PDE loss. Then, in a similar manner, the total loss function $\mathcal{L}$ is defined as:

$$\mathcal{L}(\{\mathbf{x}_n^{\text{total}}\}; \theta) = \lambda_{\text{pde}} \mathcal{L}_{\text{pde}}(\{\mathbf{x}_n\}; \theta) + \lambda_{\text{ic}} \mathcal{L}_{\text{ic}}(\{\mathbf{x}_n^{\text{ic}}\}; \theta) + \lambda_{\text{bc}} \mathcal{L}_{\text{bc}}(\{\mathbf{x}_n^{\text{bc}}\}; \theta).$$

Hyperparameters $\lambda_{\text{pde}}$, $\lambda_{\text{ic}}$, and $\lambda_{\text{bc}}$ control the balance between the PDE, IC, and BC loss terms. Then, $f_\theta$ is trained to estimate appropriate solution $u$ for PDEs by minimizing the total loss $\mathcal{L}$.

**Adaptive sampling based on residual distribution.** Classical PINNs generally adopt uniform collocation sampling. To improve efficiency, a residual-based adaptive sampling strategy was proposed [31], where each point $\mathbf{x}_n$ is drawn with probability $p(\mathbf{x}_n) := \frac{|\mathcal{R}_\theta(\mathbf{x}_n)|^k}{\sum_m |\mathcal{R}_\theta(\mathbf{x}_m)|^k}$. Building on this idea, the residual-based adaptive distribution (RAD) [47] introduces a hyperparameter $c$: $p(\mathbf{x}) \propto \frac{|\mathcal{R}_\theta(\mathbf{x})|^k}{\mathbb{E}[|\mathcal{R}_\theta(\mathbf{x})|^k]} + c$. Here, $k$ highlights high-residual regions, while $c$ enforces a degree of uniformity. By tuning these parameters, RAD flexibly balances exploration and exploitation depending on the problem structure. More recently, Gaussian mixture distribution-based adaptive sampling (GAS) has been proposed [20], where high-residual validation points serve as the means of a Gaussian mixture model (GMM), and the reciprocals of residual gradients determine the diagonal covariances.

**Adaptive sampling focused on high residuals.** Alongside residual distribution-based approaches, another major line of work explores sampling methods that do not explicitly approximate the underlying distribution. These methods can often be viewed as special cases of RAD under extreme $k, c$ settings, but we categorize them here based on whether they estimate a sampling distribution.

A representative approach is residual-based adaptive refinement (RAR) [27], which iteratively selects the top-$M$ high-residual points until the mean residual falls below a tolerance. Although effective, RAR continually accumulates points and increases computational cost. To improve efficiency, the R3 method [10] retains high-residual points, uniformly resamples for diversity, and discards low-residual points. Similarly, failure-informed PINNs (FI-PINNs) adaptively sample from regions where

residuals exceed a threshold, using strategies such as self-adaptive importance sampling or subset simulation [14, 13]. Beyond these sampling strategies, recent work has questioned the appropriateness of the $L^2$ loss for solving Hamilton-Jacobi-Bellman (HJB) equations [43]. In response, adversarial training methods targeting the $L^\infty$ norm have been introduced. As we discuss later, this framework leverages partial gradient information and inherently emphasizes high-residual regions in sampling.

**Theoretical analysis of the concentration effect.** While many adaptive sampling methods have shown promising results, several theoretical aspects remain unclear. In particular, there is a lack of theoretical analysis regarding the concentration effect. Although numerous studies report success with adaptive sampling, there is limited analysis on the impact of focusing primarily on high residuals.

To bridge this analytical gap, we investigate how learning stability is influenced by the extent to which high residuals are emphasized relative to model complexity. Our analysis reveals that an excessive focus on high residuals may lead to performance degradation in PINN training.

## 3  Analysis of the Learning Stability

### 3.1  The Effect of Sampling Concentration

In this work, we define sampling concentration as the spatial aggregation of collocation points within regions characterized by large residual magnitudes. Such strategies have been shown to improve accuracy, efficiency [28, 24], stability [7, 45], and physical consistency [21, 46, 39] in PINN training. In this section, we analyze the effects of sampling concentration theoretically.

**Setup.** Consider the partial differential equation defined over the domain $\Omega = \mathcal{X} \times [0, T]$. Assume that we have $N$ collocation points forming the sample population $\mathcal{P} = \{\mathbf{x}_n\}_{n=1}^N \subset \Omega$, sampled from a uniform distribution $\mathcal{U}(\Omega)$.

**Assumption 3.1.** *For analytical simplicity, we assume that the residual error of the PDE at each collocation point $\mathbf{x}_n$ can be expressed as a linear combination of feature-mapped vectors, given an appropriate feature map $\phi : \Omega \to \mathbb{R}^D$. Specifically, we represent the residual error as follows:*

$$\mathcal{R}_\theta(\mathbf{x}_n) = \frac{\partial}{\partial t} f_\theta(\mathbf{x}_n) + \mathcal{N}_x[f_\theta](\mathbf{x}_n) \tag{3.1}$$

$$= a(\theta)^\top \phi(\mathbf{x}_n; \theta) \tag{3.2}$$

$$= \sum_{d=1}^D a_d(\theta)\phi_d(\mathbf{x}_n; \theta). \tag{3.3}$$

We regard the sampling methodology as a weighting of each sample point depending on the residual $\mathcal{R}_\theta(\mathbf{x}_n)$ and set $k = 2$. Thus, we can represent the loss function $\mathcal{L}(\mathcal{P}; \theta) = \sum_{n=1}^N w_n |\mathcal{R}_\theta(\mathbf{x}_n)|^2$. Assume that we are solving for the solution based on the gradient descent (GD) algorithm. Then,

$$\theta^{l+1} = \theta^l - \eta \nabla_\theta \left( \sum_{n=1}^N w_n^l |\mathcal{R}_{\theta^l}(\mathbf{x}_n)|^2 \right), \tag{3.4}$$

where the weights assigned to each sample point for iteration $l$ are determined as follows:

$$w_n^l \propto \exp\left( \frac{|\mathcal{R}_{\tilde{\theta}^l}(\mathbf{x}_n)|^2}{\beta^2} \right), n \in \{1, ..., N\}. \tag{3.5}$$

Additionally, $w_n^l$ is normalized to satisfy $\sum_{\mathbf{x} \in \mathcal{P}} w_n^l(\mathbf{x}) = 1$ and the parameter $\beta > 0$ preceding the residual controls the concentration of sampling with respect to the residuals. Note that the parameters $\tilde{\theta}^l = (\tilde{\theta}_1^l, \ldots, \tilde{\theta}_D^l)$ used to calculate the importance weights do not participate in the model parameter update process. Furthermore, in contexts where the meaning is clear, we will no longer explicitly indicate that $\phi$ is parameterized by $\theta$, i.e., denote $\phi(\mathbf{x}; \theta)$ as $\phi(\mathbf{x})$.

For iteration $l$, we focus on two extreme cases of interest: when $\beta$ is too large (uniform sampling), most samples receive uniform weights, resulting in uniform sampling. Conversely, when $\beta$ is close to 0 (high residual sampling), the effect is dominated by the sample with the highest residual. To explore this in more depth, consider the following propositions.

**Proposition 3.1** (Steepness of the uniform sampling). *When the sampling concentration parameter $\beta$ is sufficiently large, the maximum eigenvalue of the Hessian of the loss function can be approximated as $2\lambda_{\max}(\Sigma)$, where $\Sigma = \mathbb{E}_{\mathbf{x} \sim \mathcal{U}(\Omega)}[\phi(\mathbf{x})\phi(\mathbf{x})^\top]$ and $\lambda_{\max}(\Sigma)$ is the maximum eigenvalue of $\Sigma$.*

**Proposition 3.2** (Steepness of the high residual sampling). *When the sampling concentration parameter $\beta$ is sufficiently small, the maximum eigenvalue of the Hessian of the loss function can be approximated as $2\|\phi(\mathbf{x}^*)\|^2$, where $\mathbf{x}^* = \arg\max_{\mathbf{x} \in \mathcal{P}} |\mathcal{R}_\theta(\mathbf{x})|^2$ is the maximum residual point.*

Detailed proof can be found in Appendix B.1, B.2. It is well known that to ensure the convergence of GD algorithms, the learning rate $\eta$ must satisfy the following relationship with the largest eigenvalue $\lambda_{\max}$ of the Hessian of the loss function: $\eta < \frac{2}{\lambda_{\max}}$ [6]. Therefore, we consequently aim to examine the relationship of the largest eigenvalue in two extreme cases of $\beta$. Before presenting the main result, we introduce two additional assumptions that formalize phenomena typically observed in neural network-based models as their complexity increases.

**Assumption 3.2.** *In high-dimensional feature space, $\|\phi(\mathbf{x})\|$ follows a heavy-tailed distribution. More specifically, $\mathbb{P}(\|\phi(\mathbf{x})\| > \zeta) \sim \frac{g(\zeta)}{\zeta^\alpha}$ for large $\zeta$, where $\sim$ represents asymptotic equivalence, $g(\zeta)$ satisfies $\forall t > 0, \lim_{\zeta \to \infty} \frac{g(t\zeta)}{g(\zeta)} = 1$ and $\alpha > 0$ indicates the thickness of the tail.*

This assumption is substantiated by both empirical evidence and theoretical insights. The heavy-tailed nature of feature vectors has been documented in several studies [29, 30, 3], and theoretically, in high-dimensional settings with complex dependencies, classical assumptions of Gaussianity often break down, and heavy-tailed models provide a more accurate fit to the observed distributional behavior [4, 16, 40]. Next, we impose an assumption concerning the representative characteristics encoded in the norms of feature vectors.

**Assumption 3.3.** *As the model complexity $D$ increases, the feature vector with the maximal norm becomes increasingly representative, which results in the following: $\max_{\mathbf{x} \in \mathcal{P}} \|\phi(\mathbf{x})\| \approx \|\phi(\mathbf{x}^*)\|$ where $\mathbf{x}^* = \arg\max_{\mathbf{x} \in \mathcal{P}} |\mathcal{R}_\theta(\mathbf{x})|^2$ is the maximum residual point.*

This assumption can be seen as a concentration of measure phenomenon in high-dimensional spaces [11, 42, 32, 15]. We note that Assumptions 3.2 and 3.3 do not generally hold in isotropic settings. However, they become statistically valid under (i) the low-temperature limit $\beta \to 0$, and (ii) the high-dimensional regime $D \gg 1$ with heavy-tailed feature norms. Specifically, writing $\mathcal{R}_\theta(\mathbf{x}) = a(\theta)^\top \phi(\mathbf{x})$, we have $|\mathcal{R}_\theta(\mathbf{x})|^2 = \phi(\mathbf{x})^\top Q \phi(\mathbf{x})$ where $Q = a(\theta)a(\theta)^\top$ is rank-one. In this context, we reason as follows to determine under what conditions $\phi(\mathbf{x})^\top Q \phi(\mathbf{x}) \approx \phi(\mathbf{x})^\top \phi(\mathbf{x})$ holds.

In the limit $\beta \to 0$, the samples concentrate on regions with large $\mathcal{R}_\theta(\mathbf{x})$, thereby biasing $\phi(\mathbf{x})$ toward alignment with $a(\theta)$ and large norm. Moreover, for $D \gg 1$, feature vectors are nearly orthogonal, so large $|\mathcal{R}_\theta(\mathbf{x})|^2$ occurs only when $\phi(\mathbf{x})$ is both high-norm and well-aligned. Consequently, $\arg\max_{\mathbf{x}} |\mathcal{R}_\theta(\mathbf{x})|^2 \approx \arg\max_{\mathbf{x}} \|\phi(\mathbf{x})\|^2$ holds with high probability in this structured regime.

To support the validity of Assumptions 3.1 through 3.3, we provide a consolidated empirical verification in Appendix A, demonstrating their general applicability across a broad range of settings. Finally, under the scaling assumption that the number of samples $N$ grows linearly with model size $D$ as $N = cD$, we derive our main theoretical result as follows.

**Theorem 3.1.** *Given the heavy-tailed nature of $\|\phi(\mathbf{x})\|$ and sufficiently large model complexity $D$, we have $2\|\phi(\mathbf{x}^*)\|^2 \gg 2\lambda_{\max}(\Sigma)$. This inequality establishes a tighter upper bound on the learning rate for ensuring the convergence of the GD algorithm under the high residual sampling method.*

The detailed proof can be found in the Appendix B.3. This indicates that the stability of the algorithm can vary significantly depending on the sampling strategy and model complexity. Specifically, in these two extreme cases ($\beta \ll 1$ and $\beta \gg 1$), uniform sampling may struggle to find an appropriate solution due to the difficulty of the stiff PDE problems, while high residual sampling may fail due to instability in the learning process.

## 3.2 Characteristics of Adaptive Sampling Algorithms

Building upon the previous discussion, we now briefly review how existing adaptive sampling methods operate. In particular, we focus on how the trajectory of the sample population $\mathcal{P}$ evolves under these algorithms.

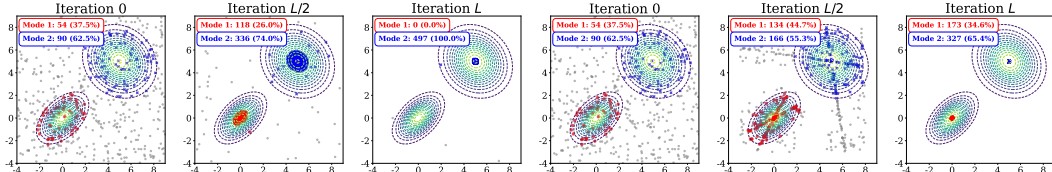

Figure 1: Schematic sampling diagram of the benchmark adaptive sampling methods: (a) R3 (first three figures), (b) $L^\infty$ (last three figures) where $|\mathcal{R}_\theta(\mathbf{x})|^k = 0.3 \times \mathcal{N}\left(\mathbf{x}; \mathbf{0}, [[1, 0.5], [0.5, 1]]\right) + 0.7 \times \mathcal{N}\left(\mathbf{x}; \mathbf{5}, [[2, -0.3], [-0.3, 2]]\right)$.

- **RAD** [47]: The modeling of residual distribution is relatively straightforward and relies on Monte Carlo integration (MCI) over the expectation $p(\mathbf{x}) \propto \frac{|\mathcal{R}_\theta(\mathbf{x})|^k}{\mathbb{E}|\mathcal{R}_\theta(\mathbf{x})|^k} + c$. In general, a larger value of $k$ corresponds to a high residual regime with smaller $\beta$, whereas a larger value of $c$ indicates a tendency toward uniform sampling regime with higher $\beta$.

- **R3** [10]: R3 employs a strategy that consistently maintains high residuals, resulting in an excessive skew in the distribution of collocation points as iterations progress. Moreover, this approach may fail to effectively handle multi-modal landscapes in the long-term, which, as demonstrated in our previous theoretical analysis, results in a scenario where the sampling concentration parameter $\beta$ becomes extremely small.

- **$L^\infty$** [43]: During the adversarial training, to estimate the inner maximal value $\sup_{\mathbf{x} \in \Omega} |\mathcal{R}_\theta(\mathbf{x})|^k$, $L^\infty$ iteratively utilizes *gradient information* $\mathrm{sign}\nabla_{\mathbf{x}}|\mathcal{R}_\theta(\mathbf{x})|^k$, allowing for some degree of access to local modes. However, there is no guarantee that the relative proportions between modes of different heights are preserved.

To facilitate an intuitive understanding of time evolving sampling methods (R3, $L^\infty$), we have illustrated the working mechanisms in a schematic diagram shown in Figure 1. For a detailed visualization of the sampling trajectories, we refer the reader to Appendix D.

## 4 Proposed Approach: Langevin Adaptive Sampling (LAS)

Similar to other residual distribution-based methodologies, our primary objective is to estimate the residual-based sampling distribution. However, unlike previous methods that directly model the distribution using residuals, we employ Langevin dynamics to model the target distribution. An intuitive visualization of our LAS framework is depicted in Figure 2.

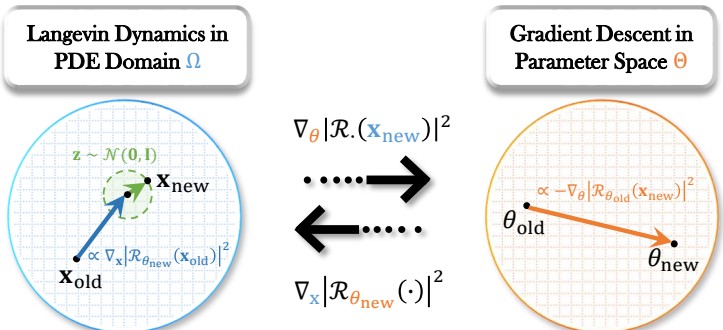

Figure 2: Bidirectional update: (Left) In the PDE domain, our LAS framework uses Langevin dynamics to adaptively update collocation points based on PDE residuals while keeping the PINN model $f_\theta$ fixed. (Right) In parameter space, the PINN model $f_\theta$ minimizes PDE residuals with the updated collocation points.

---

**Algorithm 1** Single LAS Sampling Iteration for Physics-Informed Neural Networks

---

 1: **Input: initial population** $\mathcal{P} = \mathcal{P}^0$ **with** $N_{\text{pde}}$ **collocation points.**
 2: **Output: updated population** $\mathcal{P} = \mathcal{P}^{l_{\text{L}}}$**.**
 3: **for** $l = 0$ **to** $l_{\text{L}} - 1$ **do**
 4:     **for** $\mathbf{x}_n^l \in \mathcal{P}^l$ **do**
 5:         **Calculate the residual gradient:** $\nabla_{\mathbf{x}} |\mathcal{R}_\theta(\mathbf{x}_n^l)|^2 = \nabla_{\mathbf{x}} \left| \frac{\partial}{\partial t} f_\theta(\mathbf{x}_n^l) + \mathcal{N}_x[f_\theta(\mathbf{x}_n^l)] \right|^2.$
 6:         **Sample white Gaussian noise:** $\mathbf{z}_n^l \sim \mathcal{N}(\mathbf{z}_n^l; \mathbf{0}, \mathbf{I}).$
 7:         **Follow the Langevin dynamics:** $\mathbf{x}_n^{l+1} \leftarrow \mathbf{x}_n^l + \frac{\tau}{2} \nabla_{\mathbf{x}} |\mathcal{R}_\theta(\mathbf{x}_n^l)|^2 + \beta \sqrt{\tau} \mathbf{z}_n^l.$
 8:     **end for**
 9:     **Update collocation population:** $\mathcal{P}^{l+1} \leftarrow \{\mathbf{x}_n^{l+1}\}_{n=1}^{N_{\text{pde}}}.$
10: **end for**

---

### 4.1 Langevin Dynamics and Stationary Distriburion

The dynamics of the collocation points $\mathcal{P}^l \subset \Omega$ at the $l$-th iteration in LAS are given as follows:

$$\mathbf{x}_n^{l+1} = \mathbf{x}_n^l + \frac{\tau}{2} \nabla_{\mathbf{x}} |\mathcal{R}_\theta(\mathbf{x}_n^l)|^2 + \beta \sqrt{\tau} \mathbf{z}_n^l, \tag{4.1}$$

where $\tau > 0$ is the Langevin step size, $\mathbf{z}_n^l \sim \mathcal{N}(\mathbf{z}_n^l; \mathbf{0}, \mathbf{I})$ represents the white Gaussian noise, and $\beta$ is the sampling concentration coefficient. Additionally, the residual exponent $k$ is set to 2. Unlike other methods that estimate the sampling distribution based on residuals at every iteration, LAS dynamically updates the data points without requiring the estimation of the sampling distribution. If the Langevin dynamics are allowed to run for a *sufficient number of iterations* $l_{\text{L}}$ with a *sufficiently small step size* $\tau$, we can theoretically derive the following result regarding the collocation points.

**Theorem 4.1** (Stationary distribution)**.** *For fixed $f_\theta$ and concentration parameter $\beta > 0$, sample population $\mathcal{P}^l$ asymptotically follows* $\lim_{l \to \infty} p_l(\mathbf{x}) = p_\infty(\mathbf{x}) \propto \exp\left( \frac{|\mathcal{R}_\theta(\mathbf{x})|^2}{\beta^2} \right).$

Although the proof is well known [9], to complete the formulation of the distribution under consideration, we include the full derivation in Appendix C.1. The detailed operational procedure is summarized in Algorithm 1.

Unlike R3 and RAD, which use MCI to resample collocation points independently at each step, LAS refines the sampling iteratively by reusing the current population. Similar to $L^\infty$, it exploits gradient information. However, instead of relying solely on raw gradients, LAS injects noise into the signal—a design choice whose implications are outlined below.

**Theorem 4.2** (LAS favors flat residual surfaces)**.** *Given a fixed residual landscape $\mathcal{R}_\theta$ and an initial set of randomly sampled collocation points $\mathcal{P}^0$, the LAS framework progressively refines the sampling towards flatter local maxima while avoiding less stable and sharp regions.*

An intuition-based explanation is provided in Appendix C.2 as a substitute for a formal proof. This phenomenon reflects a desirable property of Langevin dynamics, which inherently favors flatter regions of the residual landscape. As a result, when two local maxima exhibit similar residual values, the collocation points are more likely to concentrate near the flatter one. Consequently, models trained around such flat residual regions tend to exhibit more stable learning behavior.

### 4.2 Key Strengths (and Advantages) of the Proposed LAS Framework

**Robustness in high-dimensional PDEs.** In high-dimensional PDE settings, the MCI-based expectation $\mathbb{E}|\mathcal{R}_\theta(\mathbf{x})|^k \approx \frac{1}{N} \sum_{n=1}^N |\mathcal{R}_\theta(\mathbf{x}_n)|^k$ becomes unstable due to the curse of dimensionality, which demands exponentially more samples for accurate estimation. In contrast, LAS leverages local gradient information, maintaining stability even under high-dimensional conditions.

**Enhanced stability through noise injection.** Stochastic gradient descent (SGD) is known to prefer flatter minima, which are associated with better generalization [22, 19, 48]. Similarly, LAS injects noise into the gradient signal, resulting in significantly improved training stability compared to methods that rely solely on raw gradients [43]. It is reasonable to expect that sampling schemes biased toward flatter regions inherently promote more stable training dynamics.

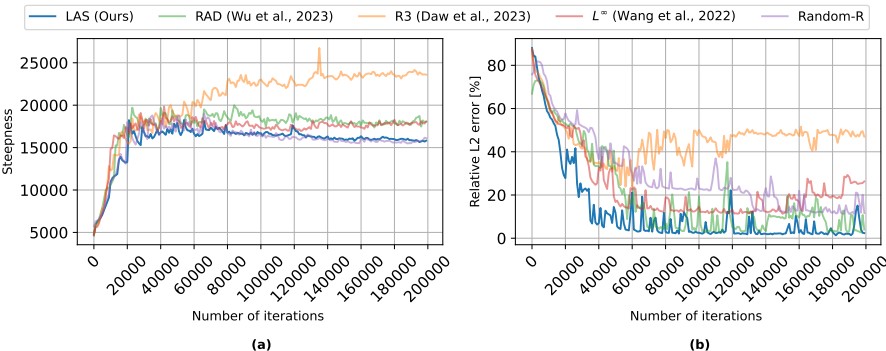

Figure 3: With fixed learning rate $\eta = 0.002$ and $4$ hidden layers, (a) The maximal eigenvalue of the Hessian (steepness) for the loss function, (b) the relative $L^2$ error curve.

# 5 Experiments

This section presents an experimental evaluation of high-residual sampling under varying model complexities and learning rates, with the number of collocation points fixed. We compare the performance of our proposed LAS method against other adaptive sampling approaches, including RAD, R3, and $L^\infty$, each evaluated under the default settings provided in their original papers, random sampling with resampling (Random-R), which essentially corresponds to uniform sampling.

**Experimental setup.** As the default settings, unless otherwise specified, the models utilized a multilayer perceptron (MLP) with $128$ nodes per layer and $4$ hidden layers, employing a hyperbolic tangent activation function in each hidden layer. The Adam optimizer was utilized with the learning rate of $\eta = 0.001$ and a decay factor of $0.9$ applied every $5,000$ iterations. Training was conducted with $200,000$ iterations, and the number of collocation points was set to $N_{\text{pde}} = 1,000$. For the LAS configuration, the residual exponent $k = 2$, the Langevin step size $\tau = 0.002$ for 1-2D PDEs and $\tau = 0.01$ for 4-8D PDEs, and the concentration parameter $\beta = 0.2$. This hyperparameter setting represents the empirically obtained optimum, with further discussion presented in the following subsection 5.5.

## 5.1 Ablation Studies

First and foremost, we sought to verify how the analytical results regarding stability and model complexity, presented in Section 3, operate and apply to the functioning of each algorithm. In this context, we performed the following key ablation studies based on the Allen-Cahn equation using 5 different random seeds.

**Steepness of the loss landscape.** In our stability analysis, we hypothesized that sampling algorithms targeting extremely high residuals—such as R3 and $L^\infty$—would induce sharper loss landscapes. To test this, we tracked the maximum eigenvalue of the Hessian of the loss throughout training, as shown in Figure 3-(a). The results support our hypothesis, indicating that high-residual-focused sampling leads to increased steepness in the loss surface. The proposed LAS method, like $L^\infty$, leverages gradient information but achieves more stable training by reducing residual surface steepness, similar to Random-R. This stability likely stems from the injected noise term. Overall, LAS outperforms Random-R and other baselines in relative $L^2$ error while maintaining the lowest steepness.

**Different number of hidden layers.** We employed MLP architectures with hidden layers ranging from 4 to 10 across all sampling methods, maintaining a learning rate of 0.001 and utilizing a step scheduler. As illustrated in Figure 4-(a), it can be observed that as the number of layers increases, the overall performance improves; however, for most sampling strategies at 10 layers, except for LAS, the performance diverges. In particular, for R3, it is evident that it fails to converge more prominently compared to other algorithms. These results indicate that the sensitivity to model complexity varies depending on the sampling method. In particular, high residual methods are more susceptible to increasing model complexity. For more clarity, regarding the loss curves during the training of PINNs based on model complexity, analyses are presented in Appendix E, not only from the perspective of depth but also from the perspective of width expansion.

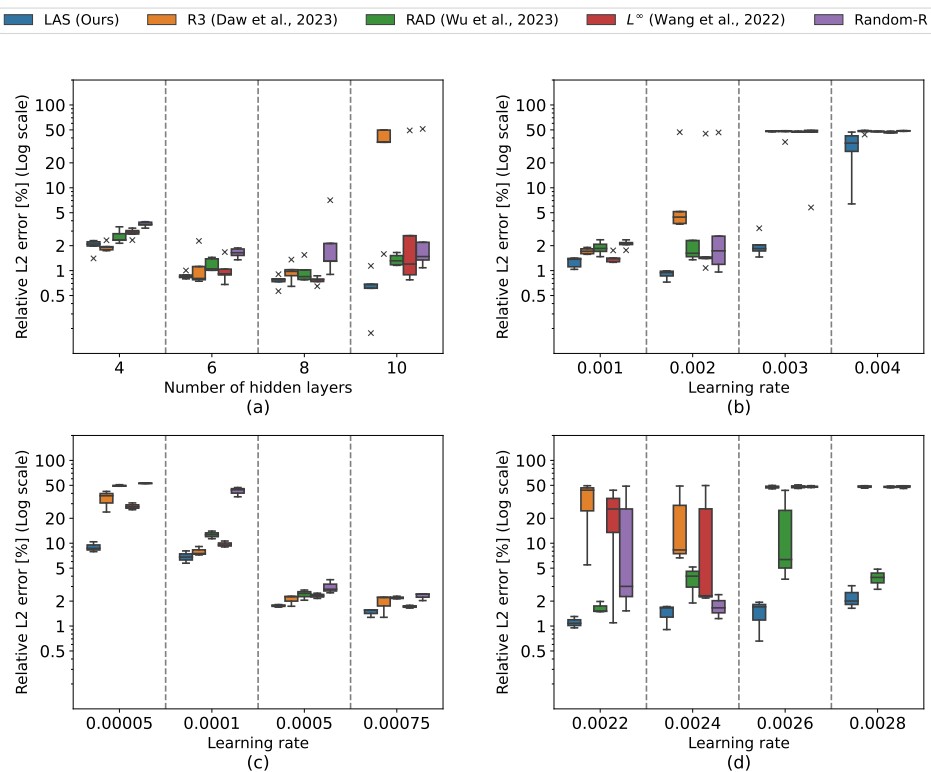

Figure 4: Relative $L^2$ error (Log scale) for the Allen-Cahn equation: (a) varying layers with $\eta = 0.001$ (with scheduler), (b)-(d) fixed 4 layers with different learning rates (no scheduler). Each boxplot is based on 5 random seeds.

**Varying learning rate $\eta$ without decaying.** We evaluated MLPs with four hidden layers across learning rates ranging from $0.001$ to $0.004$ without applying decay. As shown in Figure 4 -(b), the benchmark algorithms demonstrated performance degradation at $\eta = 0.002$ compared to $\eta = 0.001$, whereas LAS showed improvement. At $\eta = 0.003$, all methods exhibited reduced performance; however, LAS was able to partially mitigate this degradation. At $\eta = 0.004$, none of the methods produced correct solutions. In particular, we visualized the performance for very low learning rates in Figure 4-(c) and highlighted the range between $\eta = 0.002$ and $0.003$, where all algorithms begin to exhibit instability in Figure 4-(d). From this, we observe that learning does not proceed properly at very low learning rates, and for layer $4$, most algorithms become unstable at a learning rate as low as approximately $0.0022$.

## 5.2 Experiments on Representative 1-Dimensional PDEs

The proposed LAS framework is further evaluated on representative 1-dimensional PDEs derived from various benchmark problems tackled by several established algorithms, including RAD, R3, $L^\infty$, Random-R. In these evaluations, we also employ the default experimental settings as outlined earlier. The specific configurations for the PDE parameters and the hyperparameters of the baseline algorithms are detailed in Appendix F.

**Experimental results.** We evaluated each sampling method on five benchmark PDEs—Burgers', Convection, Allen-Cahn, Korteweg-De Vries, Schrödinger—using five random seeds. As summarized in the 1D case of Table 1, Random-R tended to outperform other adaptive sampling strategies in high-complexity model settings. Meanwhile, LAS consistently achieved either the best or second-best relative $L^2$ errors across all cases. In particular, LAS outperformed all methods on the Allen-Cahn and Schrödinger equations, while remaining competitive on the others.

Table 1: Relative $L^2$ error (mean $\pm$ std) across PDEs for increasing model complexity. **Bold** indicates best, underline second-best.

| Sampling method | | LAS (Ours) | | Random-R | | RAD | | R3 | | $L^\infty$ | |
|---|---|---|---|---|---|---|---|---|---|---|---|
| Number of layers | | 8 | 10 | 8 | 10 | 8 | 10 | 8 | 10 | 8 | 10 |
| 1D | Burgers' | **0.01 ± 0.00** | **0.01 ± 0.00** | **0.01 ± 0.00** | 0.02 ± 0.00 | 0.17 ± 0.02 | 0.27 ± 0.14 | **0.01 ± 0.00** | 0.02 ± 0.00 | 0.03 ± 0.01 | 0.06 ± 0.06 |
| | Allen-Cahn | 0.77 ± 0.20 | **0.62 ± 0.27** | 2.54 ± 2.30 | 11.49 ± 19.93 | 0.99 ± 0.29 | 1.36 ± 0.19 | 0.97 ± 0.23 | 34.47 ± 17.64 | 0.76 ± 0.07 | 10.95 ± 19.16 |
| | KdV | 2.68 ± 1.74 | 1.99 ± 0.50 | **1.64 ± 0.63** | 2.89 ± 1.80 | 7.44 ± 1.83 | 7.97 ± 1.45 | 3.92 ± 2.93 | 7.02 ± 8.77 | 5.70 ± 1.45 | 4.44 ± 1.45 |
| | Schrödinger | **0.08 ± 0.01** | 0.09 ± 0.01 | 0.09 ± 0.00 | 0.11 ± 0.01 | 1.68 ± 0.15 | 2.89 ± 0.69 | 0.11 ± 0.01 | 0.15 ± 0.02 | 0.22 ± 0.06 | 0.19 ± 0.03 |
| | Convection | 0.34 ± 0.12 | 0.27 ± 0.03 | 0.30 ± 0.05 | 0.41 ± 0.10 | **0.25 ± 0.02** | 0.28 ± 0.09 | 0.39 ± 0.24 | 0.27 ± 0.05 | 73.87 ± 5.07 | 54.17 ± 27.33 |
| 2D | Burgers' | **0.05 ± 0.00** | **0.05 ± 0.00** | 0.06 ± 0.01 | 0.06 ± 0.00 | **0.05 ± 0.00** | 0.06 ± 0.00 | 0.07 ± 0.01 | 0.06 ± 0.02 | 0.06 ± 0.00 | 0.05 ± 0.01 |
| | Heat | 0.26 ± 0.00 | 0.22 ± 0.03 | **0.18 ± 0.01** | 0.20 ± 0.04 | 1.44 ± 0.17 | 1.54 ± 0.15 | 8.97 ± 1.95 | 6.66 ± 4.20 | 14.29 ± 1.01 | 13.07 ± 6.21 |
| 4D | DF-Heat | **1.72 ± 0.23** | 2.14 ± 0.18 | 7.73 ± 1.85 | 75.91 ± 41.68 | 5.72 ± 0.42 | 78.20 ± 37.10 | 2.15 ± 0.25 | 80.23 ± 34.24 | 2.46 ± 0.67 | 53.78 ± 46.31 |
| 6D | DF-Heat | **3.49 ± 0.20** | 31.16 ± 39.74 | 5.53 ± 0.81 | 100.00 ± 0.00 | 6.39 ± 1.14 | 100.00 ± 0.00 | 4.98 ± 0.18 | 100.00 ± 0.00 | 4.39 ± 0.61 | 100.00 ± 0.00 |
| 8D | DF-Heat | **6.92 ± 0.31** | 34.68 ± 38.21 | 17.63 ± 1.78 | 100.00 ± 0.00 | 67.50 ± 39.83 | 100.00 ± 0.00 | 65.03 ± 43.11 | 100.00 ± 0.00 | 13.21 ± 4.46 | 100.00 ± 0.00 |

## 5.3 Experiments on High Dimensional PDEs

Beyond the one-dimensional setting, we applied the existing approaches and the proposed algorithm to higher-dimensional PDE problems. Benchmark problems include 2D Burgers', 2D heat [1], and dimension flexible heat (DF-heat) 4 to 8D equations [49]. These PDEs have analytics solutions, so the performance of all the sampling methods could be fairly observed and evaluated in high dimensional PDE cases. Especially, we modified the DF-heat PDE to pose a more challenging problem setting by increasing frequency. The details for PDEs are also described in Appendix F. We first report the performance of all sampling methods under the default settings from the original papers (Table 1).

**Experimental results.** In the 2D Burgers' and heat equations, Table 1 shows that Random-R and LAS exhibited competitive performance. RAD, R3, and $L^\infty$, on the other hand, experienced notable performance degradation, primarily because the 2D PDEs represent smooth cases rather than high-frequency or stiff regimes. For the 4D, 6D and 8D cases, LAS significantly outperformed all other adaptive sampling strategies. In the 4D, Random-R and $L^\infty$ produced plausible solutions, but their performance remained inferior to that of LAS. Meanwhile, MCI-based methods such as RAD and R3 suffered from instability. Notably, for the 6D and 8D cases, all the other sampling approaches failed to find appropriate solutions. Although convergence was observed under different balance terms, the resulting performance was inferior and highly sensitive, as discussed in the following subsection.

## 5.4 Further Discussion on Loss Balancing, Model Hyperparameters, and Applicability

To ensure fair comparison, we conducted an extensive search for the optimal loss balance terms ($\lambda_{\mathrm{ic}}, \lambda_{\mathrm{bc}}, \lambda_{\mathrm{pde}}$) and hyperparameter configurations of the baseline methods, as detailed in Appendix G. Additional experiments demonstrate that RAD and R3 outperform Random-R when equipped with the optimal loss balance terms and hyperparameter settings identified in Appendix G. However, both RAD and R3 exhibit sensitivity to the problem dimensionality and the choice of loss weighting. In contrast, LAS consistently attains either the best or second-best relative $L^2$ error across all evaluated scenarios, including smooth, stiff, low-dimensional, and high-dimensional cases. In terms of practical application, we further evaluate robustness across different model architectures (Appendix H), a factor that may be of particular relevance to practitioners.

## 5.5 LAS Hyperparameter Tuning and Computational Complexity

The practical applicability of LAS depends on the number of Langevin iterations $l_{\mathrm{L}}$, which directly affects computational complexity. We therefore compare its cost with existing sampling methods in Appendix I, and the results are summarized in Figure 5. The computational time and memory usage are in the order: Random-R, R3, RAD, LAS, and $L^\infty$. Gradient-based methods such as LAS and $L^\infty$ generally incur higher costs; in particular, $L^\infty$ is much slower because it requires many iterations (20 by default) with re-initialization each epoch, whereas LAS needs only $l_{\mathrm{L}} = 1$ without re-initialization. Thus, increasing the number of iterations should be considered with caution.

Detailed tuning results in Appendix J show that LAS with $l_{\mathrm{L}} = 1$ and without re-initialization achieves competitive performance, indicating that LAS can be applied in practice with computational cost comparable to Random-R, R3, and RAD.

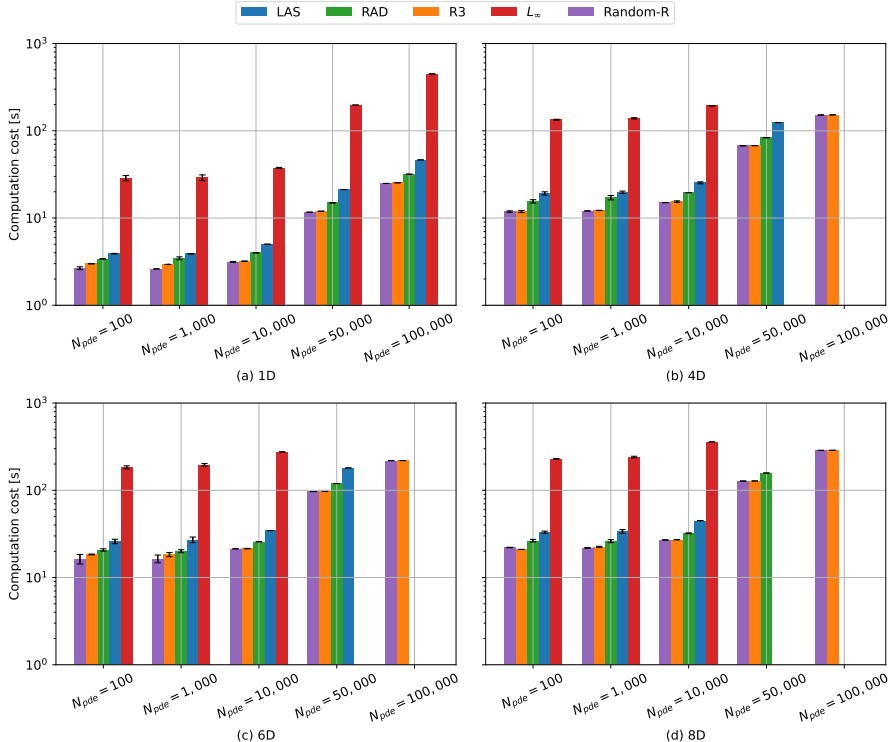

Figure 5: Comparison of computational time (seconds per 1,000 epochs) for each method across varying problem dimensions on an RTX 4090. Empty bars denote out-of-memory failures. Subplots: (a) 1D, (b) 4D, (c) 6D, and (d) 8D.

## 6 Conclusion and Future Research Directions

This paper investigates the stability of training physics-informed neural networks (PINNs) under adaptive sampling strategies, particularly as model complexity increases. Theoretical analysis shows that methods focusing excessively on high residuals may undermine stability, especially in deeper networks or with larger learning rates. To address this, we introduce Langevin dynamics-based Adaptive Sampling (LAS), which updates collocation points using residual-weighted Langevin dynamics. Empirical results demonstrate that LAS ensures stable convergence even in high-dimensional PDEs, where conventional methods often fail.

Our Langevin-based sampling scheme builds on a theoretically grounded MCMC formulation that is both simple and effective. While the current approach shows strong performance, future work could explore more efficient variants for high-dimensional problems and develop principled strategies for hyperparameter selection.

### Acknowledgments

This research was supported in part by the National Research Foundation of Korea(NRF) grant funded by the Korea government(MSIT)(RS-2024-00349582) and in part by the National Research Foundation (NRF) Grants, Basic Research Laboratory, under Grant RS-2023-00218908.

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

# Appendix

## A  Empirical Validation of Assumptions

By definition, the residual $\mathcal{R}_\theta$ involves transforming the neural network output $f_\theta$ through an operator within a certain function space. This implies that the features of the function, resulting from the neural network output combined with an additional operator, are not explicitly defined. Consequently, we extract the feature vector of the resulting function through local feature vector estimation based on the linearization of the residual function. Before delving into the main discussion, we first explain the logic behind how the feature vector $\phi$ is inferred.

### A.1  Local Approximation of the Feature Vector $\phi$

Let $\mathcal{R}_\theta(\mathbf{x}) = \frac{\partial}{\partial t} f_\theta(\mathbf{x}) + \mathcal{N}_x[f_\theta](\mathbf{x})$ represent $[g(f_\theta)](\mathbf{x})$. To validate the assumption that a suitable linearization exists, our goal is to derive a proper linear approximation of $[g(f_\theta)](\mathbf{x})$ at a specific point $\mathbf{x} \in \Omega = \mathcal{X} \times [0, T]$, given a specific function $f_\theta$. In this process, we will utilize a Taylor expansion for $[g(f_\theta)](\mathbf{x})$. It is important to note that since $g(f_\theta)$ represents the behavior in the function space, understanding how $g$ responds to small perturbations in $f_\theta$ is crucial. This analysis employs the Fréchet derivative.

To summarize briefly, $g(f_{\theta+\Delta\theta}) \approx g(f_\theta) + D_g(f_\theta)(f_{\theta+\Delta\theta} - f_\theta)$, which implies that the result can be linearized around a baseline function $f_\theta$ where $D_g(f_\theta) = \lim_{\Delta\theta \to 0} \frac{\|g(f_{\theta+\Delta\theta}) - g(f)\|}{\|\Delta\theta\|}$. Since our focus is on the linearization of $[g(f_\theta)](\mathbf{x})$, it is essential to ensure that $f_{\theta+\Delta\theta}$ is a function close to $f_\theta$ within the function space. To achieve this, small noise perturbations $\Delta\theta$ are added to the neural network $f_\theta$. In conclusion, to approximate the value at a specific point $\mathbf{x}$, we proceed as follows:

$$[g(f_{\theta+\Delta\theta})](\mathbf{x}) \approx [g(f_\theta) + D_g(f_\theta)(f_{\theta+\Delta\theta} - f_\theta)](\mathbf{x}) \tag{A.1}$$

$$= [g(f_\theta)](\mathbf{x}) + [D_g(f_\theta)(f_{\theta+\Delta\theta} - f_\theta)](\mathbf{x}) \tag{A.2}$$

$$= [g(f_\theta)](\mathbf{x}) + [D_g(f_\theta)](\mathbf{x})\big(f_{\theta+\Delta\theta}(\mathbf{x}) - f_\theta(\mathbf{x})\big). \tag{A.3}$$

Here, if $f_\theta$ is assumed to be a well-trained PINN model and perturbation $\Delta\theta$ is sufficiently small, we can readily infer the following for the first term:

$$[g(f_\theta)](\mathbf{x}) \approx 0, \quad \forall \mathbf{x} \in \Omega.$$

Consequently, the linear approximation of the function $[g(f_{\theta+\Delta\theta})](\mathbf{x})$ can be expressed using the Fréchet derivative. The aspect that conflicts with our assumption is that, in this context, the Fréchet derivative can act as a function dependent on $\mathbf{x}$. Therefore, we refer to this as a local approximation.

**Approximation of Fréchet derivative.** According to the problem formulation of PINN, $g$ is an operator that takes the function $f$ as input and generates new values through partial derivatives such as $\partial_t f, \partial_x f, \partial_{tt} f, \partial_{tx} f$, and their combinations. Thus, we can assume $g(f) = G(f, \partial_t f, \partial_x f, \partial_{tt} f, \partial_{tx} f, \cdots)$. Here, $G$ is a multivariate function that combines the derivative terms. Next, considering a scenario where a slight perturbation $\Delta\theta$ is applied to $f_\theta$, the Fréchet derivative can be approximated as follows:

$$D_g(f_{\theta+\Delta\theta})\left(f_{\theta+\Delta\theta} - f_\theta\right) \tag{A.4}$$

$$\approx g(f_{\theta+\Delta\theta}) - g(f_\theta) \tag{A.5}$$

$$= G\left(f_\theta + \Delta f_\theta, \partial_t f_\theta + \Delta\partial_t f_\theta, \partial_x f_\theta + \Delta\partial_x f_\theta, \cdots\right) - G(f_\theta, \partial_t f_\theta, \partial_x f_\theta, \cdots) \tag{A.6}$$

$$\approx \frac{\partial G}{\partial\{f_\theta\}}\Delta f_\theta + \frac{\partial G}{\partial\{\partial_t f_\theta\}}\Delta\partial_t f_\theta + \frac{\partial G}{\partial\{\partial_x f_\theta\}}\Delta\partial_x f_\theta + \cdots \tag{A.7}$$

$$= \sum_k \frac{\partial G}{\partial\{\partial_k f_\theta\}}\Delta\partial_k f_\theta, \quad k \in \{\emptyset, t, x, tt, tx, \cdots\} \tag{A.8}$$

$$=: a(\theta)^\top \phi, \tag{A.9}$$

where $a(\theta) = \left(\frac{\partial G}{\partial\{f_\theta\}}, \frac{\partial G}{\partial\{\partial_t f_\theta\}}, \frac{\partial G}{\partial\{\partial_x f_\theta\}}, \cdots\right)$ and $\phi = (\Delta f_\theta, \Delta\partial_t f_\theta, \Delta\partial_x f_\theta, \cdots)$.

**Mathematical details.** Here, we provide a systematic summary of the considerations underlying the validity of the employed estimation method.

1. **Reliability of the Fréchet derivative** $D_g[f]$**:** The existence of the Fréchet derivative requires the following sufficient conditions:
   - $G$ must be differentiable.
   - $f$ must be sufficiently differentiable with respect to $x$ and $t$.

   Both conditions are naturally satisfied in the context of our problem. This ensures that we can extract a vector that locally approximates the actual feature vector $\phi$ for each point $\mathbf{x} \in \Omega$, thereby facilitating a robust estimation process.

2. **Condition for the constancy of** $D_g[f]$**:** It is important to note that $G$ is generally a function of $(f, \partial_t f, \partial_x f, \cdots)$, and thus implicitly depends on $\mathbf{x}$. However, when the variables are not entangled with each other, the partial derivatives can exhibit constant behavior. For instance:
   - In our case, $\frac{\partial G}{\partial \{\partial_t f\}}$ is always 1.
   - Partial derivatives in the $x$-direction such as $\frac{\partial G}{\partial \{\partial_x f\}}$, $\frac{\partial G}{\partial \{\partial_{xx} f\}}$ depend on $\mathcal{N}_x$.

   If the output of the differential operator $\mathcal{N}_x$ entangles the partial derivatives in the $x$-direction (i.e, $\mathcal{N}_x[f]$ is non-linear), the assumption that $a$ acts as a constant may weaken.

We assume that $a(\theta)$ is constant or locally linear, enabling a unified linearized form of the residual. This simplification ensures consistency across Assumption 3.1 and Propositions 3.1–3.2, and underpins our curvature analysis. We will revise the manuscript to explicitly include this structural assumption in Assumption 3.1 and ensure that its implications are consistently reflected in all related propositions.

To provide further intuition, we illustrate this within the context of two representative PDEs.

For the **convection equation**,
$$g(f) = \partial_t f + \mu \, \partial_x f \quad \Rightarrow \quad a(\theta) = (0, 1, \mu, 0, 0, \dots) \text{ is constant.}$$

For the **Burgers' equation**,
$$g(f) = \partial_t f + f \, \partial_x f - \frac{0.01}{\pi} \, \partial_{xx} f \quad \Rightarrow \quad a(\theta) = \left( \partial_x f_\theta, \ 1, \ f_\theta, \ 0, \ 0, \ 0, \ -\frac{0.01}{\pi}, \ \dots \right)$$

is $\theta$-dependent.

Under the assumption that the neural network $f_\theta$ is sufficiently expressive and smooth with respect to $\theta$, we can locally linearize $a(\theta)$ around a fixed $\theta$.

When $a(\theta)$ is affine, its Jacobian
$$\mathbf{J}_\theta(a(\theta)) = \left( \frac{\partial a_i}{\partial \theta_j} \right) =: \mathbf{J}_a$$

is constant and its Hessian
$$\mathbf{H}_\theta(a(\theta)) = 0,$$

where the derivatives are defined elementwise.

## A.2 Heavy-Tailed Behavior of the Norm of Feature Vectors

Initially, we visualized the histogram of the norms of the extracted feature vectors across all feasible grid points in Figure 6, i.e., the histogram of $\{\|\phi(\mathbf{x})\| : \mathbf{x} \in \Omega = \mathcal{X} \times [0, T]\}$ for models with 4, 6, 8, and 10 layers, respectively.

From the provided histograms, it is evident that for each PDE, the distribution increasingly exhibits heavy-tail behavior as the layer depth grows. This tendency is particularly emphasized in the following two aspects:

1. **Heavy-tail characteristics resembling Pareto distribution:** As the layer depth increases, the distribution's tail becomes thicker, consistent with the heavy-tail properties of the Pareto distribution. In a Pareto distribution, the tail probability follows the form $\mathbb{P}(X > x) \propto x^{-\alpha}$, decaying slowly and exhibiting a high frequency of extreme values. This is reflected in the histograms, where deeper layers show data concentrated in certain regions while displaying more frequent extreme values.

2. **Increased concentration and frequency of extreme values:** As the number of layers increases, the data become densely concentrated within specific ranges (represented on the $y$-axis as frequency), while significantly more frequent occurrences of large values (depicted on the $x$-axis as extreme values) are observed. This behavior suggests a progressive shift towards heavy-tail distributions.

In addition to the previously obtained histograms, we also calculated two statistical estimates—Pareto tail index and Hill estimator—based on the samples to provide a more quantitative representation.

**Pareto tail index.** The Pareto tail index, denoted by $\alpha$, quantifies the heaviness of the tail of a distribution. For a random variable $X$ with a heavy-tailed distribution, the tail probability follows a power-law:

$$\mathbb{P}(X > x) \sim x^{-\alpha}, \quad \text{as } x \to \infty,$$

where $\alpha > 0$ represents the tail index. Therefore, a smaller value of $\alpha$ corresponds to a thicker tail, indicating a slower decay of the tail probability and a higher likelihood of extreme events. Conversely, a larger value of $\alpha$ corresponds to a thinner tail, where the tail probability decays more rapidly and extreme events are less likely.

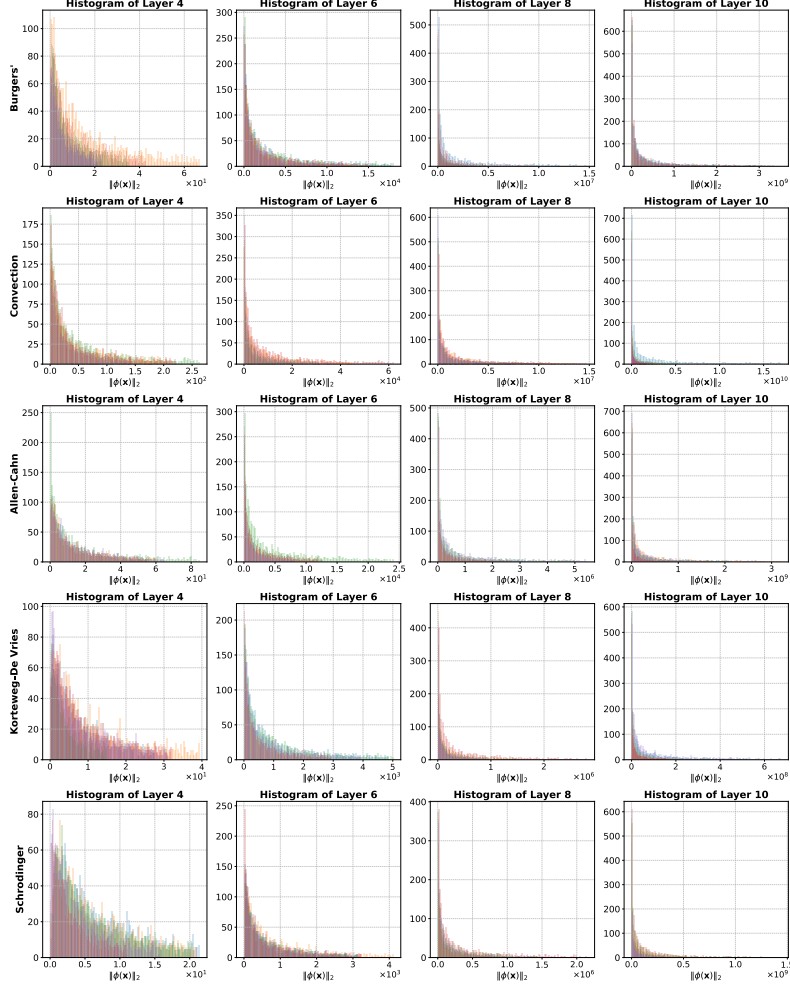

Figure 6: Each plot represents a (PDE, layer) pair, where the row corresponds to the type of PDE being solved (e.g., Burgers', Convection, Allen-Cahn, etc.), and the column indicates the model size by the number of hidden layers in the PINN (e.g., layer 4, 6, 8, 10). The histograms show the distributions of the feature vector norms $\|\phi(\mathbf{x})\|$ for each pair.

**Hill estimator.** The Hill estimator is specifically designed to estimate the inverse of the tail index, $\xi = \frac{1}{\alpha}$. Given a sample of $n$ independent and identically distributed observations $\{X_1, X_2, \ldots, X_n\}$, sorted in descending order as $X_{(1)} \geq X_{(2)} \geq \cdots \geq X_{(n)}$, the Hill estimator is defined as:

$$\hat{\xi}_k = \frac{1}{k} \sum_{i=1}^{k} \log \frac{X_{(i)}}{X_{(k+1)}},$$

where $k$ is the number of upper order statistics used for the estimation.

Figure 7 shows box plots of estimates across random seeds for various PDEs. The Pareto tail index decreases with depth, indicating heavier tails and a higher likelihood of extreme events. Across all PDEs, the index drops consistently from layer 4 to 10 and remains well below the heavy-tail threshold of 2. Even with fewer layers and 1,000 collocation points, feature norms display clear heavy-tailed behavior. The Hill estimator, as the inverse measure, increases with depth and exceeds the 0.5 threshold, further confirming the trend.

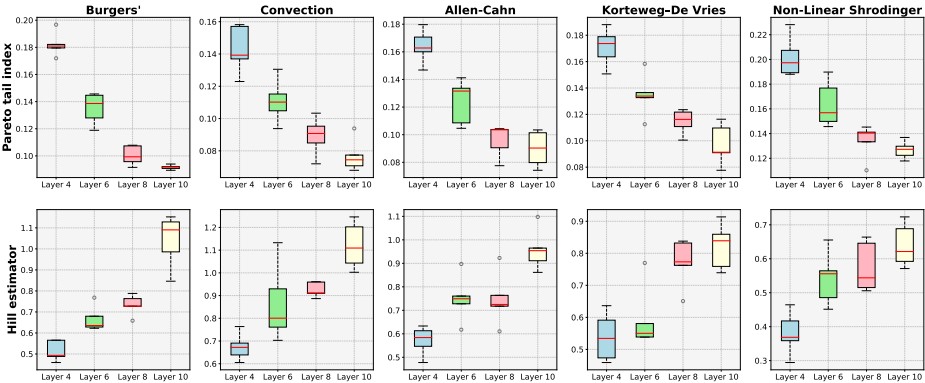

Figure 7: Two statistical estimates of the norms of the feature vectors.

### A.3 Emerging Disparities with Increasing Model Complexity

In the previous subsection, we conducted an empirical analysis of the distributional characteristics of feature vector norms. In this subsection, we aim to validate the hypothesis of the norm's emerging disparities with increasing model complexity (Assumption 3.1.3). For clarity, this relation can be expressed mathematically as $\left(\max_{\mathbf{x} \in \mathcal{P}} \|\phi(\mathbf{x})\|\right)^2 - \lambda_{\max}(\Sigma) \gg \left(\max_{\mathbf{x} \in \mathcal{P}} \|\phi(\mathbf{x})\|\right)^2 - \|\phi(\mathbf{x}^*)\|^2$, where $\mathbf{x}^* = \arg \max_{\mathbf{x} \in \Omega} |\mathcal{R}_\theta(\mathbf{x})|^2$. Here, due to the dominant scale of $\max \|\phi(\mathbf{x})\|$, we transformed the values into a logarithmic scale to investigate the relationship between $\lambda_{\max}$ and $\|\phi(\mathbf{x}^*)\|$.

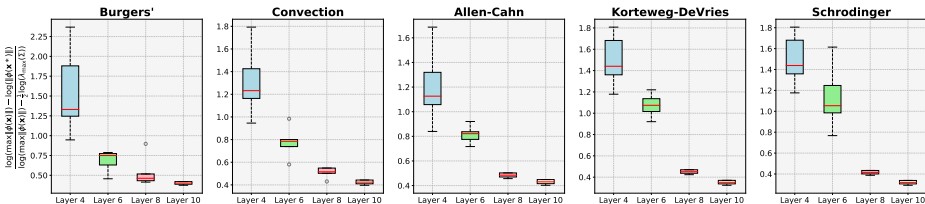

Figure 8: Disparity comparison of $\lambda_{\max}(\Sigma)$ and $\phi(\mathbf{x}^*)$ with respect to $\max \|\phi(\mathbf{x})\|$.

The Figure 8 illustrates the behavior of a logarithmic metric for various PDEs as the layer count increases. The x-axis represents the number of layers, shown as 4, 6, 8, and 10, while the y-axis represents a log-based value, denoted as $\frac{\log(\max \|\phi(\mathbf{x})\|) - \log \|\phi(\mathbf{x}^*)\|}{\log(\max \|\phi(\mathbf{x})\|) - \frac{1}{2} \log \lambda_{\max}(\Sigma)}$, which captures a ratio involving maximum values and scaled terms. Across all PDEs, the y-axis value decreases monotonically as the number of layers increases. This consistent decline in the log-metric across all PDEs suggests that as the layer count grows, the denominator in the ratio scales disproportionately compared to the numerator. This behavior indicates that the underlying system dynamics or representation becomes increasingly dominated by the factors represented in the denominator.

# B    Learning Rate Upper Bound Varying $\beta$

For the sake of simplicity, we will consider the situation at iteration $l$. Hence, in the forthcoming proof, we will omit the upper index related to iteration. i.e., denote $w_n^l$ as $w_n$. Under the assumption that the neural network $f_\theta$ is sufficiently expressive and smooth with respect to $\theta$, we can locally linearize $a(\theta)$ around a fixed $\theta$.

By the Assumption 3.1, when $a(\theta)$ is affine, its Jacobian $\mathbf{J}_\theta(a(\theta)) = \left( \frac{\partial a_i}{\partial \theta_j} \right) =: \mathbf{J}_a$ is constant and its Hessian $\mathbf{H}_\theta(a(\theta)) = 0$, where the derivatives are defined elementwise.

We consider the weighted loss

$$\mathcal{L}(\theta) = \sum_{n=1}^{N} w_n \left| a^\top \phi_n \right|^2, \quad a := a(\theta), \, \phi_n := \phi(\mathbf{x}_n). \tag{B.1}$$

Then the Hessian is

$$\mathbf{H}_\theta \mathcal{L} = 2 \sum_{n=1}^{N} w_n \left[ \left( \mathbf{J}_a^\top \phi_n \right) \left( \mathbf{J}_a^\top \phi_n \right)^\top + \left( a^\top \phi_n \right) \sum_{i=1}^{D} \phi_{n,i} \, H_\theta(a_i) \right]. \tag{B.2}$$

If $a = a(\theta)$ is linear, the second term vanishes, and we obtain

$$\mathbf{H}_\theta \mathcal{L} = 2 \mathbf{J}_a^\top \left( \sum_{n=1}^{N} w_n \, \phi_n \phi_n^\top \right) \mathbf{J}_a. \tag{B.3}$$

## B.1    Proof of Proposition 3.1

*Proof.* When the sampling concentration coefficient $\beta$ is sufficiently large, the weights $w_n$ are approximately uniform ($w_n \approx \frac{1}{N}$). Thus, the Hessian matrix $\mathbf{H}_\theta$ of the loss function with respect to $\theta$ can be approximated by:

$$\mathbf{H}_\theta(\mathcal{L}) = \mathbf{H}_\theta \left( \sum_{n=1}^{N} w_n |\mathcal{R}_\theta(\mathbf{x}_n)|^2 \right) \approx \mathbf{H}_\theta \left( \frac{1}{N} \sum_{n=1}^{N} |\mathcal{R}_\theta(\mathbf{x}_n)|^2 \right). \tag{B.4}$$

Since $\mathcal{R}_\theta(\mathbf{x}_n) = a^\top \phi_n$ and $\mathbf{H}_\theta \left( |a^\top \phi_n)|^2 \right) = 2 \mathbf{J}_a^\top \left( \sum_{n=1}^{N} w_n \, \phi_n \phi_n^\top \right) \mathbf{J}_a$, the Hessian of $\mathcal{L}$ satisfies:

$$\mathbf{H}_\theta(\mathcal{L}) \approx \frac{1}{N} \sum_{n=1}^{N} \mathbf{H}_\theta \left( |a^\top \phi_n|^2 \right) = \frac{2}{N} \sum_{n=1}^{N} \mathbf{J}_a^\top \phi_n \phi_n^\top \mathbf{J}_a. \tag{B.5}$$

This matrix approximates the product of the Jacobian and the sample covariance matrix of the feature vector $\phi(\mathbf{x})$. Therefore, for sufficiently large $N$, the maximum eigenvalue of the Hessian can be approximated as $2\lambda_{\max}(\Sigma)$. □

## B.2    Proof of Proposition 3.2

*Proof.* When the sampling concentration coefficient $\beta$ is sufficiently small, for the sample $\mathbf{x}^*$ with the largest residual (i.e., $\mathbf{x}^* = \arg\max_{\mathbf{x} \in \mathcal{P}} |\mathcal{R}_\theta(\mathbf{x})|^2$), we can consider all other weights to be zero except for $\mathbf{x}^*$. Therefore, the Hessian matrix of the loss function can be expressed as follows:

$$\mathbf{H}_\theta(\mathcal{L}) = \mathbf{H}_\theta \left( \sum_{n=1}^{N} w_n |\mathcal{R}_\theta(\mathbf{x}_n)|^2 \right) \approx \mathbf{H}_\theta \left( |\mathcal{R}_\theta(\mathbf{x}^*)|^2 \right). \tag{B.6}$$

The Hessian of the loss function can be expressed as $2 \mathbf{J}_a^\top \phi(\mathbf{x}^*) \phi(\mathbf{x}^*)^\top \mathbf{J}_a$. Therefore, in a similar manner, if we aim to compute $\lambda_{\max}$ here, we may consider the eigenvalue equation $Av = \lambda v$. Assuming $\mathbf{J}_a$ is independent of the dimension $D$, let $A = 2\phi(\mathbf{x})\phi(\mathbf{x})^\top$ and $v = \phi(\mathbf{x})$. It follows that $\phi(\mathbf{x}^)$ is an eigenvector of $A$, with the corresponding eigenvalue given by $2|\phi(\mathbf{x}^*)|^2$. Since $A$ is a rank-1 matrix, this eigenvalue is uniquely determined. □

## B.3 Proof of Theorem 3.1

*Proof.* Assume that the feature norm $\|\phi(\mathbf{x})\|$ follows a heavy-tailed distribution with tail index $\alpha > 2$, and that the number of samples $N$ satisfies $N \asymp D$.

**Step 1 (Low-temperature regime).** When the loss is dominated by one or a few extreme samples, the empirical Hessian satisfies

$$\mathbf{H}_\theta \mathcal{L} \approx 2\, w_{\max}\, \mathbf{J}_a^\top\, \phi_{\max}\phi_{\max}^\top\, \mathbf{J}_a, \quad \phi_{\max} = \max_{1 \le n \le N} \|\phi(\mathbf{x}_n)\|. \tag{B.7}$$

By extreme value theory, the maximum feature norm scales as

$$\phi_{\max} \asymp N^{1/\alpha} \asymp D^{1/\alpha}, \tag{B.8}$$

hence

$$\phi_{\max}^2 \asymp D^{2/\alpha}. \tag{B.9}$$

**Step 2 (Uniform sampling regime).** In contrast, under uniform sampling without dominance by extremes, we have
$$\mathbf{H}_\theta \mathcal{L} \approx 2\, \mathbf{J}_a^\top\, \Sigma\, \mathbf{J}_a, \quad \Sigma = \mathbb{E}[\phi(\mathbf{x})\phi(\mathbf{x})^\top], \tag{B.10}$$
where $\Sigma$ is the population covariance matrix of features.

**Step 3 (Scaling of covariance eigenvalues).** For $\alpha > 2$, $\Sigma$ is finite, and in a normalized isotropic setting $\lambda_{\max}(\Sigma) = O(1)$. Considering instead the sample covariance

$$\Phi_N = \frac{1}{N} \sum_{n=1}^{N} \phi(\mathbf{x}_n)\phi(\mathbf{x}_n)^\top, \tag{B.11}$$

results from random matrix theory for $2 < \alpha < 4$ imply

$$\lambda_{\max}(\Phi_N) \asymp \frac{\phi_{\max}^2}{N} \asymp N^{2/\alpha-1} \asymp D^{2/\alpha-1}. \tag{B.12}$$

**Step 4 (Comparison).** With $N \asymp D$, we obtain

$$\phi_{\max}^2 \asymp D^{2/\alpha} \quad \text{and} \quad \lambda_{\max}(\Phi_N) \asymp D^{2/\alpha-1}. \tag{B.13}$$

Therefore,

$$\frac{\phi_{\max}^2}{\lambda_{\max}(\Phi_N)} \asymp D \to \infty \quad \text{as} \quad D \to \infty, \tag{B.14}$$

which shows that $\phi_{\max}^2 \gg \lambda_{\max}(\Phi_N)$ for large $D$. $\square$

Finally, combining this fact with Assumption 3.3.1, which states $\|\phi(\mathbf{x}^*)\| \approx \phi_{\max}$, where $\mathbf{x}^* = \arg\max_{\mathbf{x} \in \mathcal{P}} |\mathcal{R}_\theta(\mathbf{x})|^2$, we can conclude the proof.

# C   Stationary distribution and Stability of Langevin Dynamics

## C.1   Proof of Theorem 4.1

*Proof.* Starting from the discretized Langevin dynamics:

$$\mathbf{x}^{l+1} = \mathbf{x}^l + \frac{\tau}{2}\nabla_{\mathbf{x}}|\mathcal{R}_\theta(\mathbf{x}^l)|^2 + \beta\sqrt{\tau}\mathbf{z}^l, \quad z^l \sim \mathcal{N}(\mathbf{z}^l; \mathbf{0}, \mathbf{I}). \tag{C.1}$$

We consider the continuous-time Langevin dynamics as the step size $\tau \to 0$.

$$d\mathbf{x}_s = \frac{1}{2}\nabla_{\mathbf{x}}|\mathcal{R}_\theta(\mathbf{x}_s)|^2\, ds + \beta\, dB_s, \tag{C.2}$$

where $B_s$ denotes standard Brownian motion in $\mathbb{R}^d$ and $\beta > 0$ is the diffusion coefficient. We assume the drift $\nabla_{\mathbf{x}}|\mathcal{R}_\theta(\mathbf{x})|^2$ is locally Lipschitz and satisfies a linear growth condition, ensuring the well-posedness of (C.2).

**Step 1: Fokker–Planck equation.** Let $p_s(\mathbf{x})$ denote the probability density of $\mathbf{x}_s$. The Fokker–Planck equation corresponding to (C.2) is

$$\frac{\partial p_s(\mathbf{x})}{\partial s} = -\nabla_{\mathbf{x}} \cdot \left(\frac{1}{2}\nabla_{\mathbf{x}}|\mathcal{R}_\theta(\mathbf{x})|^2\, p_s(\mathbf{x})\right) + \frac{\beta^2}{2}\Delta_{\mathbf{x}}p_s(\mathbf{x}), \tag{C.3}$$

where $\Delta_{\mathbf{x}}$ denotes the Laplacian.

**Step 2: Stationary equation.** At stationarity, $\partial_s p_s \equiv 0$, so (C.3) becomes

$$0 = -\nabla_{\mathbf{x}} \cdot \left(\frac{1}{2}\nabla_{\mathbf{x}}|\mathcal{R}_\theta(\mathbf{x})|^2\, p_\infty(\mathbf{x})\right) + \frac{\beta^2}{2}\Delta_{\mathbf{x}}p_\infty(\mathbf{x}), \tag{C.4}$$

where $p_\infty$ is the stationary density.

**Step 3: Candidate Gibbs-type solution.** We claim that

$$p_\infty(\mathbf{x}) = \frac{1}{Z}\exp\left(\frac{|\mathcal{R}_\theta(\mathbf{x})|^2}{\beta^2}\right), \tag{C.5}$$

where $Z$ is the normalizing constant, solves (C.4).

**Step 4: Verification.** Differentiating (C.5):

$$\nabla_{\mathbf{x}}p_\infty(\mathbf{x}) = \frac{1}{\beta^2}\nabla_{\mathbf{x}}|\mathcal{R}_\theta(\mathbf{x})|^2\, p_\infty(\mathbf{x}), \tag{C.6}$$

$$\Delta_{\mathbf{x}}p_\infty(\mathbf{x}) = \left(\frac{1}{\beta^2}\Delta_{\mathbf{x}}|\mathcal{R}_\theta(\mathbf{x})|^2 + \frac{1}{\beta^4}\left\|\nabla_{\mathbf{x}}|\mathcal{R}_\theta(\mathbf{x})|^2\right\|^2\right)p_\infty(\mathbf{x}). \tag{C.7}$$

Substituting (C.6) and (C.7) into (C.4) gives

$$-\frac{1}{2}\left[\Delta_{\mathbf{x}}|\mathcal{R}_\theta|^2\, p_\infty + \nabla_{\mathbf{x}}|\mathcal{R}_\theta|^2 \cdot \nabla_{\mathbf{x}}p_\infty\right] + \frac{\beta^2}{2}\left[\frac{1}{\beta^2}\Delta_{\mathbf{x}}|\mathcal{R}_\theta|^2 + \frac{1}{\beta^4}\left\|\nabla_{\mathbf{x}}|\mathcal{R}_\theta|^2\right\|^2\right]p_\infty$$

$$= -\frac{1}{2}\left[\Delta_{\mathbf{x}}|\mathcal{R}_\theta|^2 + \frac{1}{\beta^2}\left\|\nabla_{\mathbf{x}}|\mathcal{R}_\theta|^2\right\|^2\right]p_\infty + \frac{1}{2}\left[\Delta_{\mathbf{x}}|\mathcal{R}_\theta|^2 + \frac{1}{\beta^2}\left\|\nabla_{\mathbf{x}}|\mathcal{R}_\theta|^2\right\|^2\right]p_\infty$$

$$= 0.$$

Thus $p_\infty$ satisfies (C.4) exactly.

**Step 5: Conclusion.** By uniqueness of the stationary solution under the given assumptions, the stationary density of (C.2) is given by (C.5). $\qquad\square$

## C.2 Proof of Theorem 4.2

*Proof.* Rather than presenting a formal proof, we illustrate the underlying intuition using a simple toy example.

**Preference for flat high-residual regions under Langevin dynamics.** For a fixed neural network parameter $\theta$, we denote the residual as $\mathcal{R}_\theta(\mathbf{x}) := \mathcal{R}(\mathbf{x})$ and $\mathbf{x} \in \mathbb{R}$ for notational simplicity.

We analyze the residual function:

$$\mathcal{R}(\mathbf{x}) = -a(\mathbf{x}^2 - \theta^2)^2 - \epsilon \exp\left(-\frac{(\mathbf{x} - \theta)^2}{2\sigma^2}\right), \tag{C.8}$$

which has two local maxima: a broader maximum at $\mathbf{x}_{\text{flat}} = -\theta$ and a narrower, sharper maximum at $\mathbf{x}_{\text{sharp}} = \theta$ due to the additional Gaussian term.

We consider Langevin sampling governed by the stationary distribution:

$$\pi(\mathbf{x}) \propto \exp\left(\rho \mathcal{R}(\mathbf{x})\right), \tag{C.9}$$

where $\rho > 0$ is the inverse temperature. This distribution favors regions with higher residual values.

To compare the probability of sampling near each local maximum, we approximate the local probability mass by Laplace's method. Around each maximum $\mathbf{x}^*$, we expand the residual as:

$$\mathcal{R}(\mathbf{x}) \approx \mathcal{R}(\mathbf{x}^*) + \frac{1}{2}\mathcal{R}''(\mathbf{x}^*)(\mathbf{x} - \mathbf{x}^*)^2, \tag{C.10}$$

where $\mathcal{R}''(\mathbf{x}^*) < 0$ and the local curvature is defined as $\mathbf{H} = -\mathcal{R}''(\mathbf{x}^*) > 0$.

The probability mass near a local maximum $\mathbf{x}^*$ is approximated by:

$$\int_{\text{near } \mathbf{x}^*} \exp(\rho \mathcal{R}(\mathbf{x}))\, d\mathbf{x} \approx \exp(\rho \mathcal{R}(\mathbf{x}^*)) \cdot \sqrt{\frac{2\pi}{\rho \mathbf{H}}}. \tag{C.11}$$

Let:

$$\mathbf{H}_{\text{flat}} = -\mathcal{R}''(\mathbf{x}_{\text{flat}}) = 8a\theta^2, \tag{C.12}$$

$$\mathbf{H}_{\text{sharp}} = -\mathcal{R}''(\mathbf{x}_{\text{sharp}}) = 8a\theta^2 + \frac{\epsilon}{\sigma^2}, \tag{C.13}$$

so that $\mathbf{H}_{\text{flat}} \ll \mathbf{H}_{\text{sharp}}$ for small $\sigma$.

Assuming the residual values at both peaks are equal, i.e., $\mathcal{R}(\mathbf{x}_{\text{flat}}) = \mathcal{R}(\mathbf{x}_{\text{sharp}})$, the ratio of local probability masses becomes:

$$\frac{\mathbb{P}(\text{near } \mathbf{x}_{\text{flat}})}{\mathbb{P}(\text{near } \mathbf{x}_{\text{sharp}})} \approx \sqrt{\frac{\mathbf{H}_{\text{sharp}}}{\mathbf{H}_{\text{flat}}}}. \tag{C.14}$$

After normalization, the respective probabilities are:

$$\mathbb{P}(\text{near } \mathbf{x}_{\text{flat}}) = \frac{1/\sqrt{\mathbf{H}_{\text{flat}}}}{1/\sqrt{\mathbf{H}_{\text{flat}}} + 1/\sqrt{\mathbf{H}_{\text{sharp}}}} = \frac{\sqrt{\mathbf{H}_{\text{sharp}}}}{\sqrt{\mathbf{H}_{\text{flat}}} + \sqrt{\mathbf{H}_{\text{sharp}}}}, \tag{C.15}$$

$$\mathbb{P}(\text{near } \mathbf{x}_{\text{sharp}}) = \frac{\sqrt{\mathbf{H}_{\text{flat}}}}{\sqrt{\mathbf{H}_{\text{flat}}} + \sqrt{\mathbf{H}_{\text{sharp}}}}. \tag{C.16}$$

This analysis shows that, despite equal peak heights, Langevin dynamics assign higher probability to the broader maximum at $\mathbf{x}_{\text{flat}}$ due to its lower curvature and larger effective volume in the residual landscape.

For example, with parameters $a = 1$, $\theta = 2$, $\epsilon = 10$, and $\sigma = 0.1$, the curvature values are $\mathbf{H}_{\text{flat}} = 32$ and $\mathbf{H}_{\text{sharp}} = 1032$, yielding a probability ratio of approximately $5.68$. After normalization, we obtain:

$$\mathbb{P}(\text{near } \mathbf{x}_{\text{flat}}) \approx 0.85, \quad \mathbb{P}(\text{near } \mathbf{x}_{\text{sharp}}) \approx 0.15.$$

This quantitatively confirms that Langevin dynamics significantly prefer broader residual peaks under equal height conditions.

**Gradient sensitivity induced by sharp residual regions.** Residual peaks with high curvature, such as those near $\mathbf{x}_{\mathrm{sharp}}$, induce steep gradients with respect to the model parameters due to the amplified sensitivity of $\mathcal{R}$ to perturbations in $\mathbf{x}$. Formally, the chain rule

$$\frac{d}{d\theta}\mathcal{R}_\theta(\mathbf{x}) = \frac{\partial\mathcal{R}}{\partial\mathbf{x}} \cdot \frac{d\mathbf{x}}{d\theta}$$

indicates that when $\partial\mathcal{R}/\partial\mathbf{x}$ is large—as in highly localized peaks—even small variations in $\mathbf{x}$ can result in unstable updates in $\theta$.

Moreover, a concentration of samples around sharp residual peaks effectively sharpens the empirical loss landscape, making the optimization process highly sensitive to initialization and learning rate. In contrast, broader regions such as those around $\mathbf{x}_{\mathrm{flat}}$ yield smoother gradients and reduced curvature in the loss surface, enhancing the robustness of training against both gradient noise and model perturbations.

This observation highlights that the inherent sampling bias of Langevin dynamics toward flatter high-residual regions contributes not only to generalization but also to numerical stability during optimization.

$\square$

# D  Sample Trajectories of LAS and Benchmark Algorithms

We visualized the collocation point trajectories during the training under various adaptive sampling algorithms. The experimental settings follow the default configurations specified in the main text, with 4 layers and the number of Langevin iterations set to $l_{\mathrm{L}} = 1$. The background, shown as a heatmap using the plasma colormap, represents the solution errors, where dark purple indicates low values and bright yellow indicates high values. White points represent the collocation points aligned with solution errors to intuitively show how each algorithm works.

## D.1  Random-R Sample Trajectory

The figure below represents the sample trajectory of Random-R, where different collocation points are uniformly sampled at each iteration.

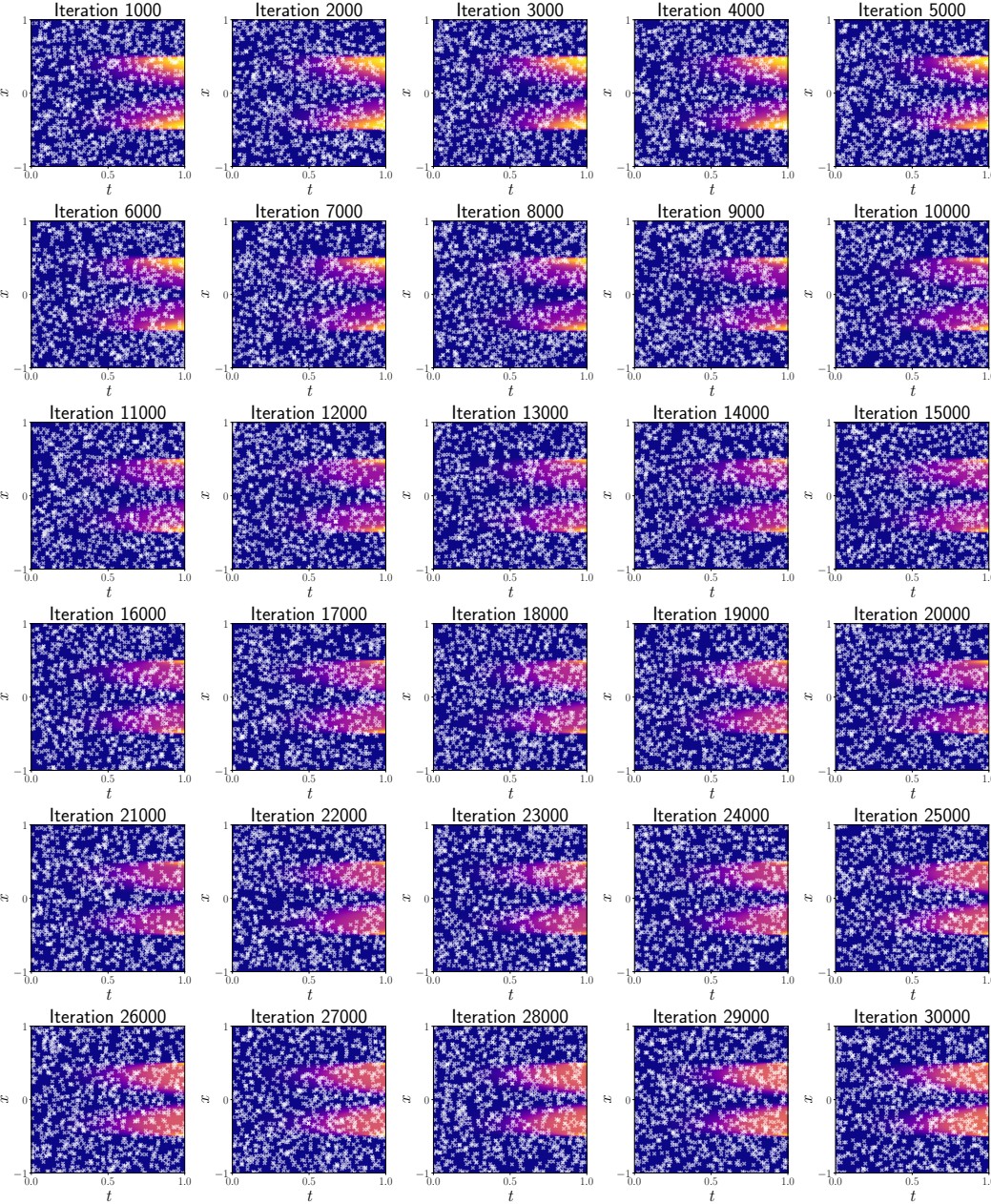

Figure 9: Random-R sample trajectory.

## D.2 RAD Sample Trajectory

This figure presents the RAD sample trajectory over multiple iterations, demonstrating a relatively stable pattern of sample distribution. To ensure a fair comparison, we fixed the number of collocation points for the denser set to be $N_{pde}$. As iterations progress, the sample points concentrate around regions of high errors, with some diversity maintained throughout. However, despite the overall stability, the RAD sampling method exhibits a distribution that is not significantly different from the Random-R approach. The clustering becomes more pronounced in certain areas, but the overall spread and distribution of samples remain similar, suggesting that RAD does not offer a distinct advantage over random-R in terms of improving sampling diversity.

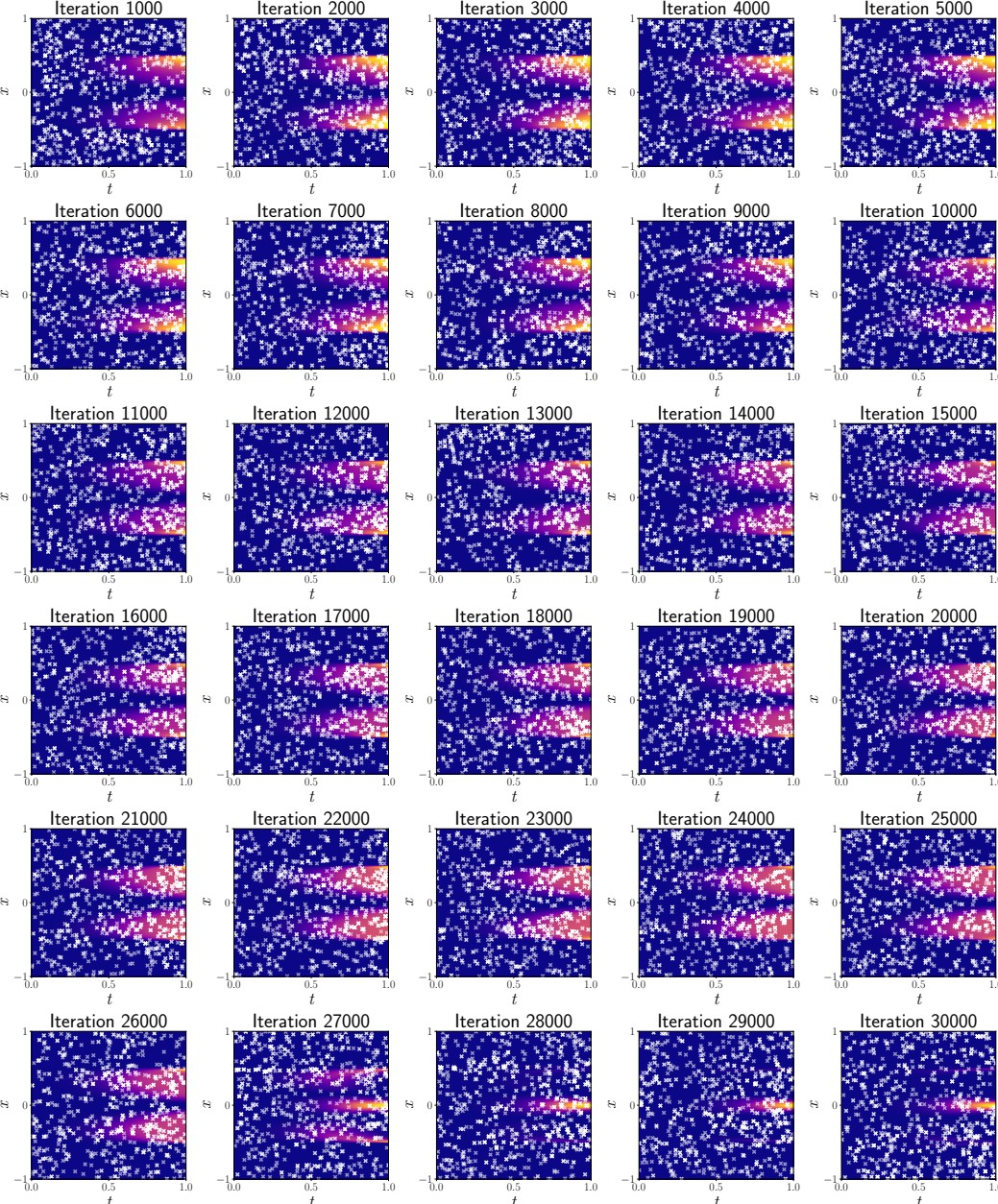

Figure 10: RAD sample trajectory.

### D.3 R3 Sample Trajectory

This figure illustrates the evolution of sample trajectories in the R3 algorithm, showing a clear concentration of samples in regions with high errors as the process progresses. While early iterations exhibit some scattering, the sample points increasingly cluster around specific areas of the solution errors, leading to a lack of diversity in later stages. Furthermore, this imbalance indicates instability in the sampling strategy, as it fails to maintain a continuous, balanced shift in the sample population across the entire domain. The discontinuous change in the sample population may result in instability from the perspective of the learning process.

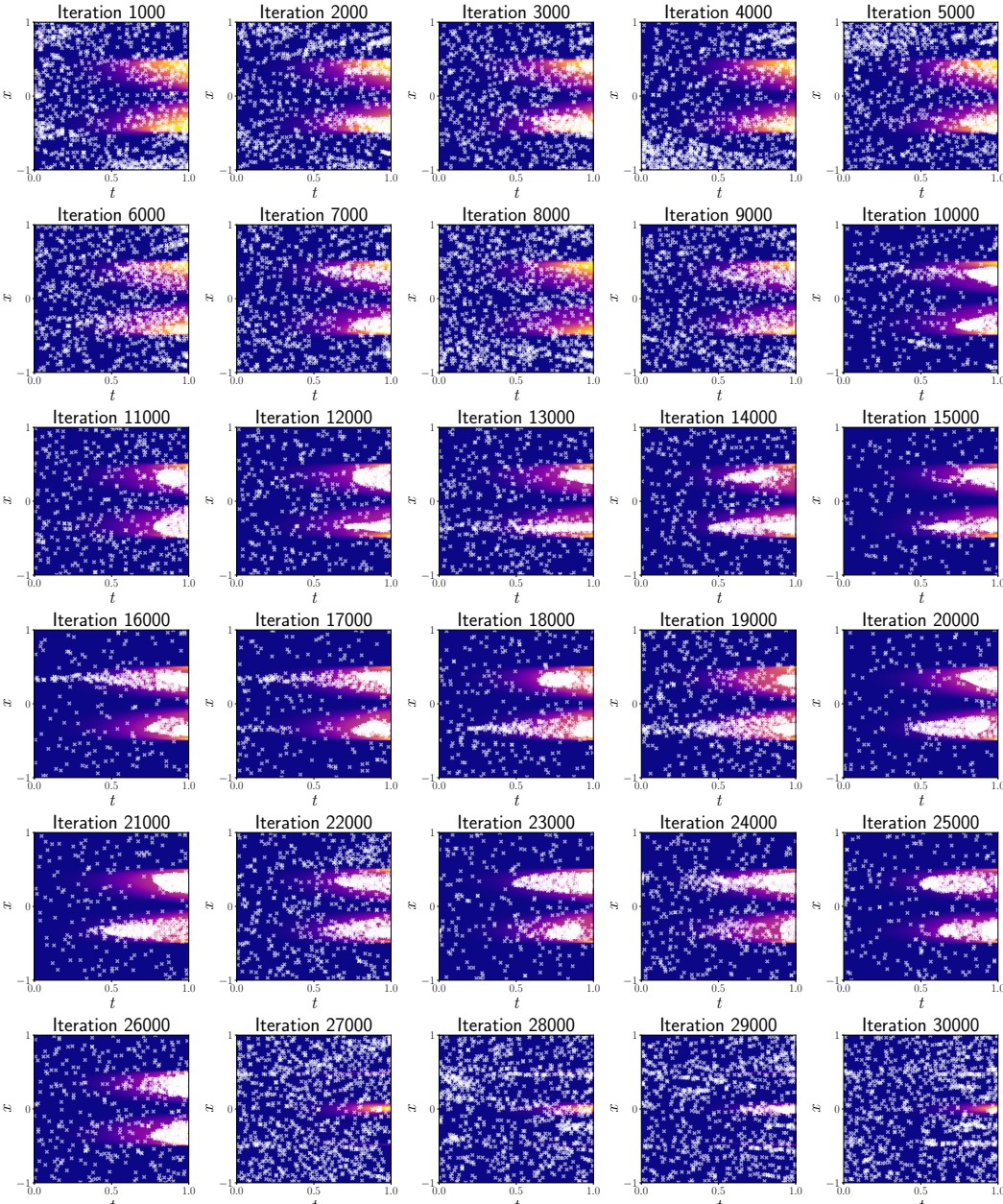

Figure 11: R3 sample trajectory.

## D.4 $L^\infty$ Sample Trajectory

This figure illustrates the evolution of sample trajectories in the $L^\infty$ algorithm. As the number of iterations increases, the samples become overly concentrated in regions with high errors, leading to a lack of diversity across the domain, particularly in areas with lower errors. This imbalance goes against the goal of maintaining a well-distributed sample set proportional to the solution errors. While some adaptation occurs, the excessive focus on extreme errors (small $\beta$ case) results in a skewed distribution, highlighting the need for more balanced and diverse sampling to improve the algorithm's performance.

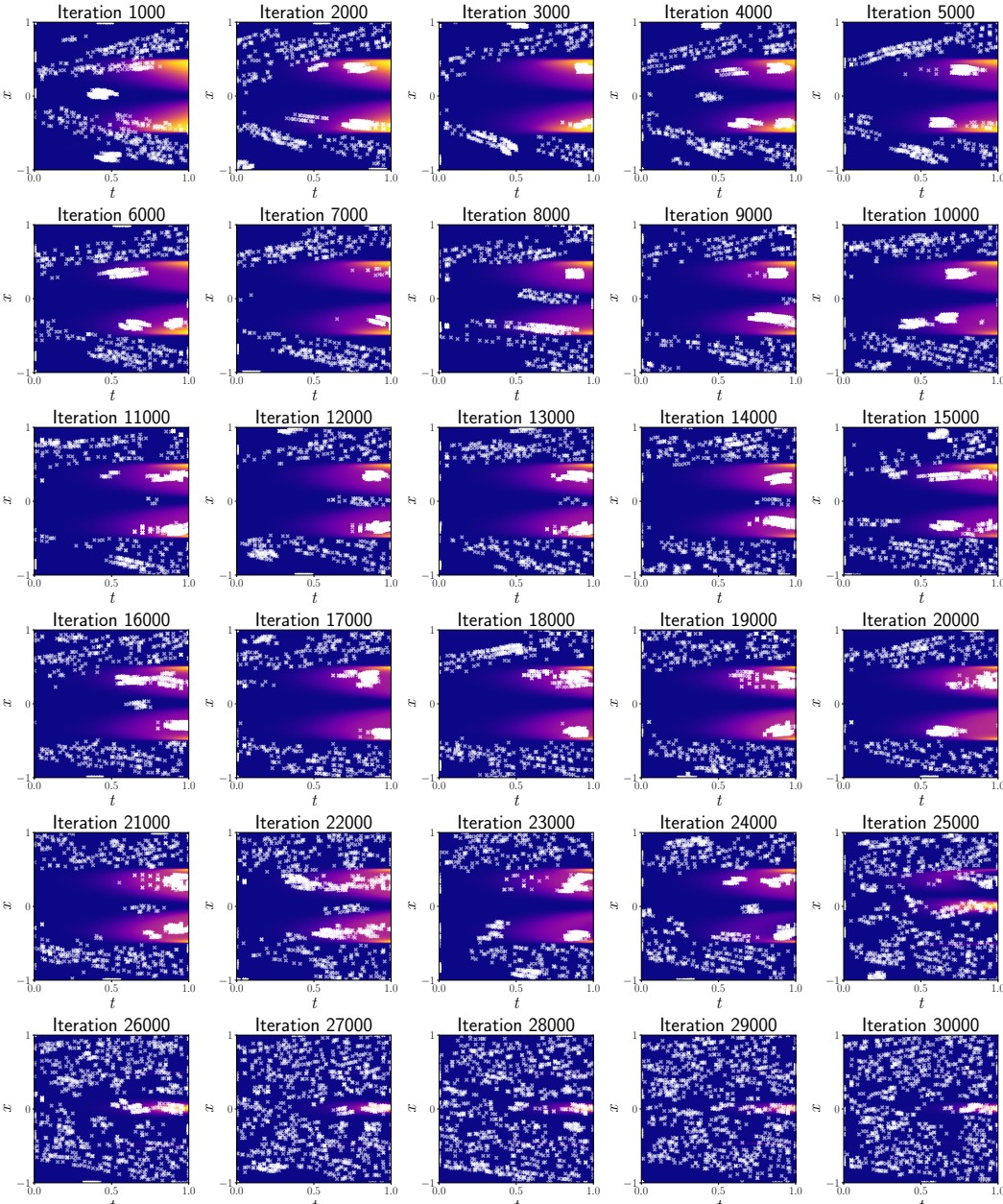

Figure 12: $L^\infty$ sample trajectory.

## D.5 LAS Sample Trajectory

This figure depicts the sample trajectory of the proposed LAS algorithm. As iterations progress, the sample points are proportionally distributed according to the solution errors, maintaining diversity across the domain. Unlike other methods, proposed LAS algorithm avoids over-concentration in regions of high errors, instead ensuring that sample points are scattered in a balanced manner. Additionally, the distribution adapts in line with the error peaks, with an appropriate portion of samples allocated based on the peak heights. This indicates that the LAS algorithm successfully addresses the key objectives of both proportionality and diversity in sample distribution, improving stability and overall performance.

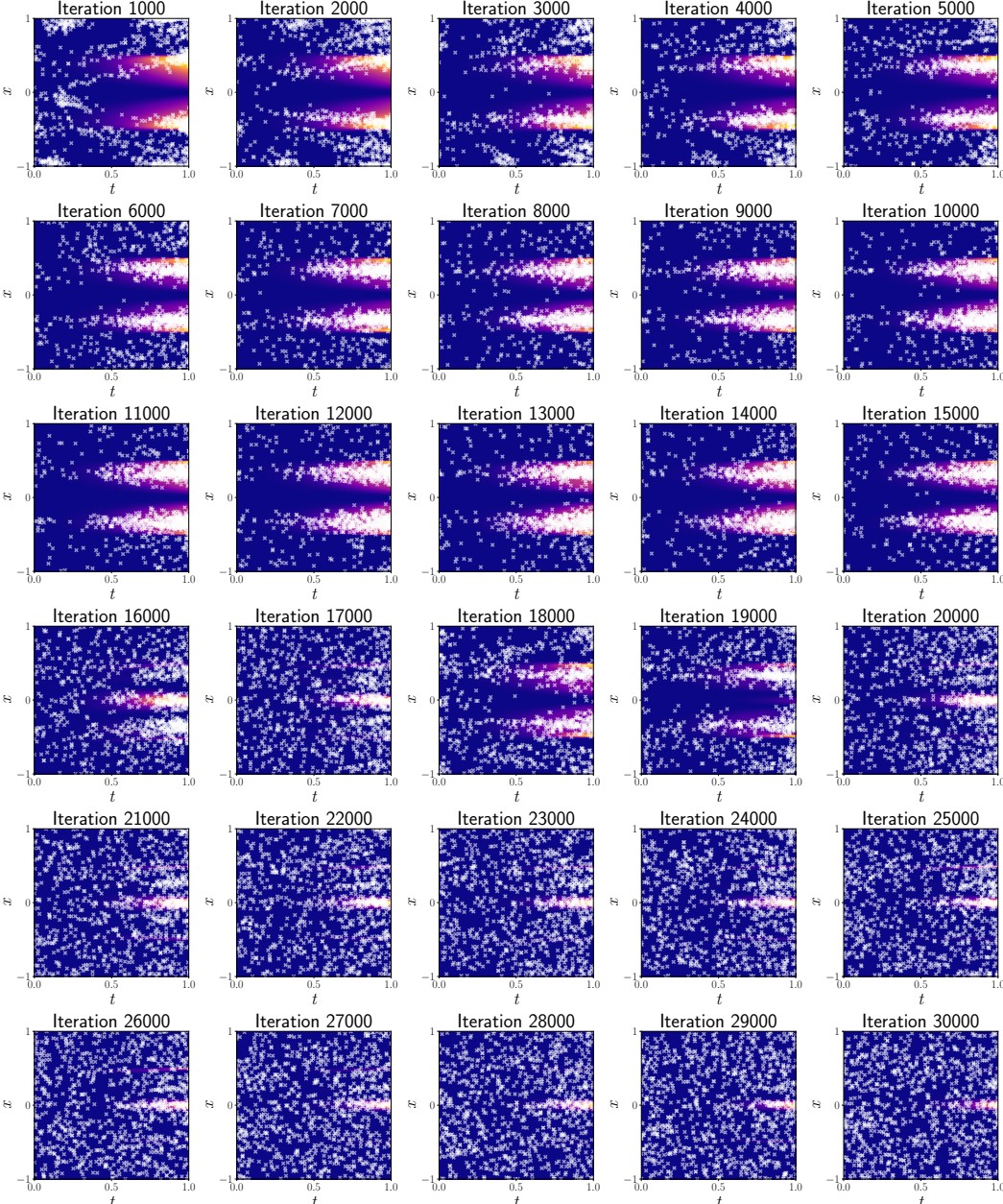

Figure 13: LAS sample trajectory.

# E  Relationship Between Learning Rate and Model Complexity

## E.1  Learning Curve Variation with Increasing Depth

Here, we aim to visualize and interpret the learning curves for the Allen-Cahn equation observed during the training process of the models, as reported in Table 1, with detailed experimental settings provided in Section 5, varying only the depth of the neural networks.

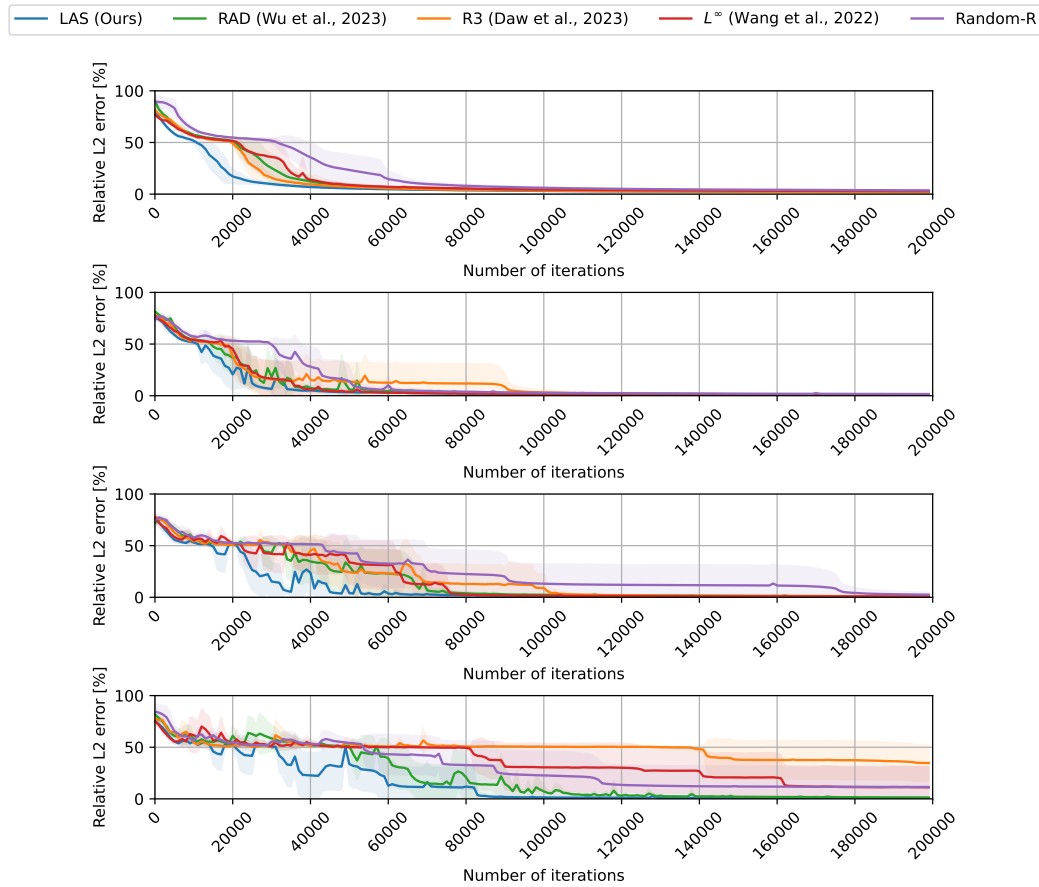

Figure 14: From top to bottom in the figure, the learning curves correspond to layers 4, 6, 8, and 10 for the Allen-Cahn equation.

Our primary observation is that most algorithms exhibit a slow learning progression until the learning rate reaches a specific value (which, of course, varies depending on the algorithm). This phenomenon appears to correlate with the degree of high residual concentration in the residual landscape of each algorithm. Specifically, the relative $L^2$ error in the learning curve requires more iterations to drop below a certain threshold (denoted as 50 in the figure) as the number of layers increases.

From an algorithmic perspective, most methods achieve the 50-threshold of the relative $L^2$ error crossing before iteration 40,000 with a 4 layer network. However, as the number of layers increases, particularly with 8 and 10 layers, the threshold-crossing iterations are significantly delayed. This delay is especially pronounced in algorithms such as R3 and $L^\infty$, which are highly focused on regions of extreme high residuals. This observation suggests that these algorithms are more affected by the increased complexity and residual concentration in deeper networks.

To verify whether this phenomenon depends on overall model complexity, in the following subsection, we also conducted experiments focusing on increasing the width rather than the depth of the model.

## E.2 Learning Curve Variation with Increasing Width

Simple calculations show that a neural network with 8 hidden layers and 128 nodes per layer has the same number of parameters as a neural network with 4 hidden layers and a width of 203. However, comparisons based solely on parameter count are inadequate, as depth introduces issues such as gradient vanishing.

Therefore, instead of viewing width solely from the perspective of parameter count, we conducted experiments by progressively doubling the width. The results showed that, similar to depth, increasing width also led to a gradual breakdown in learning stability. Consistent with the rankings observed in depth experiments, among adaptive sampling techniques, LAS reached the relative $L^2$ error threshold of 50 the fastest. Interestingly, Random-R demonstrated robustness in this setting, particularly with wide neural networks.

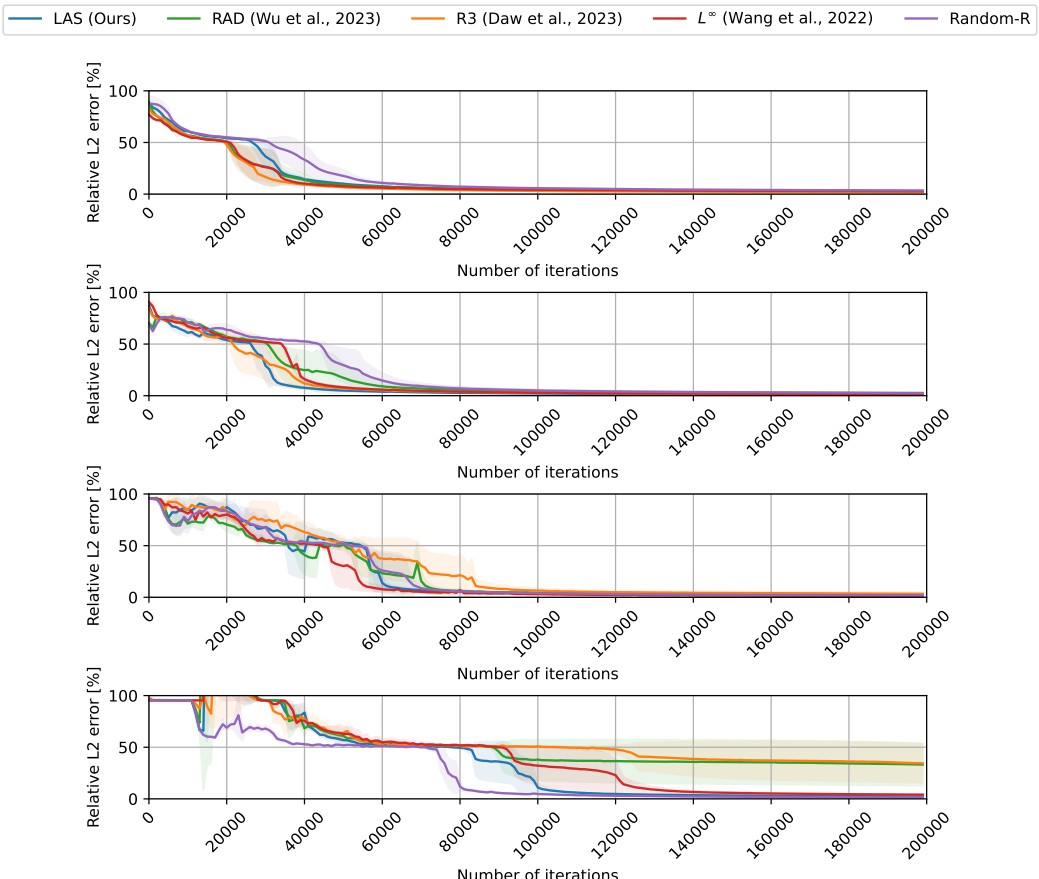

Figure 15: From top to bottom in the figure, the learning curves correspond to 128, 256, 512, and 1024 nodes per hidden layer, each with 4 layers, for the Allen-Cahn equation.

Through the experimental analyses described above, we argue that the proposed LAS demonstrates superior performance in terms of learning stability, particularly for models with high complexity.

# F Supplementary Details on Experimental Setup

## F.1 Details of Partial Differential Equations

Burgers': We set $(\lambda_{\text{ic}}, \lambda_{\text{bc}}, \lambda_{\text{pde}}) = (100, 1, 1)$ to solve the equation

$$\frac{\partial u}{\partial t} + u\frac{\partial u}{\partial x} - \frac{0.01}{\pi}\frac{\partial^2 u}{\partial x^2} = 0, \quad x \in [-1, 1], \quad t \in [0, 1]; \tag{F.1}$$
$$u(-1, t) = u(1, t) = 0; \tag{F.2}$$
$$u(x, 0) = -\sin \pi x. \tag{F.3}$$

Convection: We set $(\lambda_{\text{ic}}, \lambda_{\text{bc}}, \lambda_{\text{pde}}) = (100, 100, 1)$ to solve the equation

$$\frac{\partial u}{\partial t} + 50\frac{\partial u}{\partial x} = 0, \quad x \in [0, 2\pi], \quad t \in [0, 1]; \tag{F.4}$$
$$u(0, t) = u(2\pi, t); \tag{F.5}$$
$$u(x, 0) = \sin x. \tag{F.6}$$

Allen-Cahn: We set $(\lambda_{\text{ic}}, \lambda_{\text{bc}}, \lambda_{\text{pde}}) = (100, 1, 1)$ to solve the equation

$$\frac{\partial u}{\partial t} - 0.0001\frac{\partial^2 u}{\partial x^2} - 5(u - u^3) = 0, \quad x \in [-1, 1], \quad t \in [0, 1]; \tag{F.7}$$
$$u(-1, t) = u(1, t); \tag{F.8}$$
$$u_x(-1, t) = u_x(1, t); \tag{F.9}$$
$$u(x, 0) = x^2 \cos \pi x. \tag{F.10}$$

Korteweg-De Vries: We set $(\lambda_{\text{ic}}, \lambda_{\text{pde}}, \lambda_{\text{bc}}) = (100, 1, 1)$ to solve the equation

$$\frac{\partial u}{\partial t} + u\frac{\partial u}{\partial x} + 0.0025\frac{\partial^3 u}{\partial x^3} = 0, \quad x \in [-1, 1], \quad t \in [0, 1]; \tag{F.11}$$
$$u(-1, t) = u(1, t); \tag{F.12}$$
$$u(x, 0) = \cos \pi x. \tag{F.13}$$

Schrödinger: We set $(\lambda_{\text{ic}}, \lambda_{\text{bc}}, \lambda_{\text{pde}}) = (100, 1, 1)$ to solve the equation

$$i\frac{\partial h}{\partial t} + 0.5\frac{\partial^2 h}{\partial x^2} + |h|^2 h = 0, \quad x \in [-5, 5], \quad t \in \left[0, \frac{\pi}{2}\right]; \tag{F.14}$$
$$h(-5, t) = h(5, t); \tag{F.15}$$
$$h_x(-5, t) = h_x(5, t); \tag{F.16}$$
$$h(x, 0) = 2\text{sech}(x). \tag{F.17}$$

Heat 2D: We set $(\lambda_{\text{ic}}, \lambda_{\text{bc}}, \lambda_{\text{pde}}) = (1, 1, 1)$ to solve the equation:

$$\frac{\partial u}{\partial t} = \frac{\partial^2 u}{\partial x^2} + \frac{\partial^2 u}{\partial y^2}, \quad (x, y, t) \in \Omega \times [0, T], \tag{F.18}$$
$$u(x, y, 0) = h(x, y), \quad (x, y) \in \Omega, \tag{F.19}$$
$$u(x, y, t) = g(x, y, t), \quad (x, y, t) \in \partial\Omega \times [0, T], \tag{F.20}$$
$$h(x, y) = 3\sin(\pi x)\sin(\pi y) + \sin(3\pi x)\sin(\pi y), \tag{F.21}$$
$$g(x, y, t) = 3\sin(\pi x)\sin(\pi y)e^{-2\pi^2 t} + \sin(3\pi x)\sin(\pi y)e^{-10\pi^2 t}, \tag{F.22}$$
$$u(x, y, t) = 3\sin(\pi x)\sin(\pi y)e^{-2\pi^2 t} + \sin(3\pi x)\sin(\pi y)e^{-10\pi^2 t}, \tag{F.23}$$
$$\Omega = [0, 1]^2, \quad T = 1. \tag{F.24}$$

Burgers' 2D: We set $(\lambda_{\text{ic}}, \lambda_{\text{bc}}, \lambda_{\text{pde}}) = (1, 1, 1)$ to solve the equation:

$$\partial_t u + u\partial_x u + v\partial_y u = \frac{0.05}{\pi}(\partial_{xx} u + \partial_{yy} u), \quad (x, y, t) \in \Omega \times [0, T], \tag{F.25}$$

$$\partial_t v + u\partial_x v + v\partial_y v = \frac{0.05}{\pi}(\partial_{xx} v + \partial_{yy} v), \quad (x, y, t) \in \Omega \times [0, T], \tag{F.26}$$

$$u(x, y, 0) = h_1(x, y), \quad (x, y) \in \Omega, \tag{F.27}$$

$$v(x, y, 0) = h_2(x, y), \quad (x, y) \in \Omega, \tag{F.28}$$

$$h_1(x, y) = 0.75 - \frac{0.25}{1 + \exp\left(\frac{\pi}{0.05 \cdot 32}(-4x + 4y)\right)}, \tag{F.29}$$

$$h_2(x, y) = 0.75 + \frac{0.25}{1 + \exp\left(\frac{\pi}{0.05 \cdot 32}(-4x + 4y)\right)}, \tag{F.30}$$

$$u(x, y, t) = g_1(x, y, t), \quad (x, y, t) \in \partial\Omega \times [0, T], \tag{F.31}$$

$$u(x, y, t) = g_2(x, y, t), \quad (x, y, t) \in \partial\Omega \times [0, T], \tag{F.32}$$

$$g_1(x, y, t) = 0.75 - \frac{0.25}{1 + \exp\left(\frac{\pi}{0.05 \cdot 32}(-4x + 4y - t)\right)}, \tag{F.33}$$

$$g_2(x, y, t) = 0.75 + \frac{0.25}{1 + \exp\left(\frac{\pi}{0.05 \cdot 32}(-4x + 4y - t)\right)}, \tag{F.34}$$

$$u(x, y, t) = 0.75 - \frac{0.25}{1 + \exp\left(\frac{\pi}{0.05 \cdot 32}(-4x + 4y - t)\right)}, \tag{F.35}$$

$$v(x, y, t) = 0.75 + \frac{0.25}{1 + \exp\left(\frac{\pi}{0.05 \cdot 32}(-4x + 4y - t)\right)}, \tag{F.36}$$

$$\Omega = (0, 1)^2, \quad T = 1. \tag{F.37}$$

Dimension-flexible Heat: We set $(\lambda_{\text{ic}}, \lambda_{\text{bc}}, \lambda_{\text{pde}}) = (1, 1, 1)$ to solve the equation:

$$\frac{\partial u}{\partial t} - \Delta u = f(x, t), \quad (x, t) \in \Omega \times [0, T], \tag{F.38}$$

$$u(x, t) = g(x, t), \quad (x, t) \in \partial\Omega \times [0, T], \tag{F.39}$$

$$u(x, 0) = h(x), \quad x \in \Omega, \tag{F.40}$$

$$f(x, t) = \left(\frac{\omega^2}{d} - 1\right)\cos\left(\frac{\omega}{d}\sum_{i=1}^{d} x_i\right)\exp(-t), \tag{F.41}$$

$$g(x, t) = \cos\left(\frac{\omega}{d}\sum_{i=1}^{d} x_i\right)\exp(-t), \tag{F.42}$$

$$h(x) = \cos\left(\frac{\omega}{d}\sum_{i=1}^{d} x_i\right), \tag{F.43}$$

$$u(x, t) = \cos\left(\frac{\omega}{d}\sum_{i=1}^{d} x_i\right)\exp(-t), \tag{F.44}$$

$$\Omega = (-1, 1)^d, \quad T = 1, \tag{F.45}$$

$$\text{where} \quad (\omega = 20, d = 4), \quad (\omega = 30, d = 6), \text{and} \quad (\omega = 40, d = 8). \tag{F.46}$$

## F.2 Implementation Details of Baseline Algorithms

For all PDEs, we conducted experiments by fixing the algorithms' hyperparameters to the values specified in the original baseline code. Specifically, for R3, we set number of MaxIter = 1, for Annealed R-FIPINN, we set $\epsilon_r = 0.05, \epsilon_p = 0.01, a = 0.5, b = 1, p = 0.1$, for GAS-T, we set $N_G = 20, N_a = 20, \lambda = 1$, for RAD, we set $c = k = 1$, and for $L^\infty$, we fixed the number of gradient steps at 20 and step size 0.05.

| Equation | Solution | Number of $N_{\mathrm{ic}}$ | Number of $N_{\mathrm{bc}}$ | Number of $N_{\mathrm{pde}}$ | Evaluation Point |
|---|---|---|---|---|---|
| 1D Burgers' | Approximated | 100 | 100 | 1,000 | $N_x = 256, N_t = 100$ |
| 1D Allen-Cahn | Approximated | 100 | 100 | 1,000 | $N_x = 512, N_t = 200$ |
| 1D KdV | Approximated | 100 | 100 | 1,000 | $N_x = 200, N_t = 100$ |
| 1D Schrödinger | Approximated | 100 | 100 | 1,000 | $N_x = 256, N_t = 200$ |
| 1D Convection | Analytic | 100 | 100 | 1,000 | $N_x = 512, N_t = 200$ |
| 2D Burgers' | Analytic | 100 | 100 | 1,000 | $N_x = 100, N_t = 20$ |
| 2D Heat | Analytic | 100 | 100 | 1,000 | $N_x = 100, N_t = 20$ |
| 4D DF-heat | Analytic | 100 | 100 | 1,000 | $N_x = 20, N_t = 20$ |
| 6D DF-heat | Analytic | 250 | 250 | 1,000 | $N_x = 8, N_t = 8$ |
| 8D DF-heat | Analytic | 450 | 450 | 1,000 | $N_x = 5, N_t = 5$ |

Table 2: Summary of PDE solutions and points

# G Sensitivity of Adaptive Sampling Methods

## G.1 Different Loss Balance Terms ($\lambda_{ic}$, $\lambda_{bc}$, $\lambda_{pde}$)

| Sampling methods | | LAS (ours) | Random-R | RAD | R3 | $L^\infty$ | Annealed R-FIPINN | GAS-T |
|---|---|---|---|---|---|---|---|---|
| Number of layers | | 8 | 8 | 8 | 8 | 8 | 8 | 8 |
| 1D Allen-Cahn | 100,1,1 | 0.77 ± 0.20 | 2.54 ± 2.30 | 0.99 ± 0.29 | 0.97 ± 0.23 | **0.76 ± 0.07** | - | - |
| | 200,1,1 | 0.89 ± 0.03 | 1.69 ± 0.38 | 1.15 ± 0.18 | 10.62 ± 19.56 | **0.77 ± 0.05** | - | - |
| | 10,1,1 | **1.02 ± 0.20** | 1.07 ± 0.48 | 1.25 ± 0.27 | 75.81 ± 32.72 | 86.77 ± 7.53 | - | - |
| | 10,1,5 | **1.55 ± 0.80** | 2.06 ± 0.40 | 2.79 ± 0.24 | 65.51 ± 18.70 | 98.77 ± 0.79 | - | - |
| | 1,1,1 | 13.20 ± 22.52 | **3.21 ± 1.19** | 14.65 ± 18.34 | 64.32 ± 7.30 | 90.81 ± 12.27 | - | - |
| 4D DF-heat | 100,1,1 | **1.72 ± 0.23** | 9.73 ± 0.44 | 6.64 ± 2.72 | 9.53 ± 5.27 | 2.92 ± 1.95 | 8.87 ± 0.65 | 3.80 ± 1.70 |
| | 200,1,1 | **1.82 ± 0.42** | 8.87 ± 1.67 | 5.72 ± 0.42 | 6.50 ± 1.19 | 4.06 ± 1.87 | 9.25 ± 0.70 | 3.74 ± 2.43 |
| | 10,1,1 | **1.79 ± 0.20** | 7.98 ± 37.59 | 24.98 ± 37.59 | 16.08 ± 3.25 | 2.46 ± 0.67 | 6.75 ± 1.97 | 7.60 ± 3.30 |
| | 10,1,5 | **2.85 ± 0.34** | 25.18 ± 37.84 | 72.53 ± 32.73 | 15.57 ± 9.22 | 42.71 ± 46.79 | 29.86 ± 36.25 | 14.47 ± 8.07 |
| | 1,1,1 | **1.86 ± 0.20** | 7.73 ± 1.85 | 12.31 ± 1.24 | 13.56 ± 5.24 | 10.95 ± 14.41 | 26.36 ± 36.89 | 16.59 ± 11.01 |
| 6D DF-heat | 100,1,1 | **3.49 ± 0.20** | 6.14 ± 0.42 | 13.84 ± 2.08 | 29.91 ± 35.23 | 4.46 ± 0.42 | 5.75 ± 1.05 | 100.00 ± 0.00 |
| | 200,1,1 | **4.18 ± 0.58** | 5.21 ± 0.82 | 10.63 ± 0.93 | 30.51 ± 34.90 | 4.39 ± 0.61 | 5.80 ± 0.72 | 37.20 ± 36.19 |
| | 10,1,1 | **4.79 ± 0.33** | 5.53 ± 0.81 | 31.81 ± 34.30 | 49.47 ± 41.28 | 26.37 ± 37.05 | 5.49 ± 0.34 | 100.00 ± 0.00 |
| | 10,1,5 | **7.33 ± 1.37** | 70.30 ± 36.59 | 100.00 ± 0.00 | 69.37 ± 37.51 | 90.21 ± 19.57 | 100.00 ± 0.00 | 100.00 ± 0.00 |
| | 1,1,1 | **5.20 ± 0.92** | 53.18 ± 46.82 | 100.00 ± 0.00 | 61.37 ± 39.22 | 100.00 ± 0.00 | 62.57 ± 45.83 | 100.00 ± 0.00 |
| 8D DF-heat | 100,1,1 | **7.72 ± 0.59** | 52.41 ± 39.33 | 84.71 ± 30.56 | 100.00 ± 0.00 | 100.00 ± 0.00 | 49.22 ± 41.61 | 100.00 ± 0.00 |
| | 200,1,1 | **6.92 ± 0.31** | 17.63 ± 1.78 | 83.59 ± 32.80 | 100.00 ± 0.00 | 13.21 ± 4.46 | 15.01 ± 2.69 | 100.00 ± 0.00 |
| | 10,1,1 | **7.97 ± 0.30** | 100.00 ± 0.00 | 100.00 ± 0.00 | 100.00 ± 0.00 | 100.00 ± 0.00 | 100.00 ± 0.00 | 100.00 ± 0.00 |
| | 10,1,5 | **11.17 ± 0.65** | 100.00 ± 0.00 | 100.00 ± 0.00 | 100.00 ± 0.00 | 100.00 ± 0.00 | 100.00 ± 0.00 | 100.00 ± 0.00 |
| | 1,1,1 | **9.08 ± 0.50** | 100.00 ± 0.00 | 100.00 ± 0.00 | 100.00 ± 0.00 | 100.00 ± 0.00 | 100.00 ± 0.00 | 100.00 ± 0.00 |

Table 3: Relative $L^2$ error [%] performance comparison of different balance terms across varying PDEs when $N_{\text{pde}} = 1000$. **Bold** indicates best, underline second-best.

| Sampling methods | | LAS (ours) | Random-R | RAD | R3 | $L^\infty$ |
|---|---|---|---|---|---|---|
| Number of layers | | 8 | 8 | 8 | 8 | 8 |
| 1D Allen-Cahn | ic/bc/pde | (100/1/1) | (1/1/1) | (100/1/1) | (100/1/1) | (100/1/1) |
| | Rel L2 error [%] | 0.77 ± 0.20 | 1.07 ± 0.48 | 0.99 ± 0.29 | 0.97 ± 0.23 | **0.76 ± 0.07** |
| 4D DF-heat | ic/bc/pde | (100/1/1) | (1/1/1) | (200/1/1) | (200/1/1) | (10/1/1) |
| | Rel L2 error [%] | **1.72 ± 0.23** | 7.73 ± 1.85 | 5.72 ± 0.42 | 6.50 ± 1.19 | 2.46 ± 0.67 |
| 6D DF-heat | ic/bc/pde | (100/1/1) | (200/1/1) | (200/1/1) | (100/1/1) | (200/1/1) |
| | Rel L2 error [%] | **3.49 ± 0.20** | 5.21 ± 0.82 | 10.63 ± 0.93 | 29.91 ± 35.23 | 4.39 ± 0.61 |
| 8D DF-heat | ic/bc/pde | (200/1/1) | (200/1/1) | (200/1/1) | (200/1/1) | (200/1/1) |
| | Rel L2 error [%] | **6.92 ± 0.31** | 17.63 ± 1.78 | 83.59 ± 32.80 | 100.00 ± 0.00 | 13.21 ± 4.46 |

Table 4: Relative $L^2$ error [%] comparison across different PDEs, reported under the best balancing setting. **Bold** indicates best, underline second-best. $N_{\text{pde}} = 1000$.

## G.2 Different Loss Balance Terms with Smaller Number of Layers

| Sampling methods | | LAS (ours) | Random-R | RAD | R3 | $L^\infty$ |
|---|---|---|---|---|---|---|
| Number of layers | | 1 | 1 | 1 | 1 | 1 |
| 4D DF-heat | 100,1,1 | **12.46 ± 3.86** | 28.39 ± 5.90 | 44.14 ± 8.39 | 18.17 ± 2.38 | 22.53 ± 1.30 |
| | 200,1,1 | **13.18 ± 2.56** | 23.35 ± 1.62 | 44.55 ± 4.93 | 24.56 ± 3.20 | 22.96 ± 2.17 |
| | 10,1,1 | 34.88 ± 3.38 | **26.43 ± 3.97** | 63.67 ± 2.02 | 38.20 ± 5.71 | 48.23 ± 23.94 |
| | 10,1,5 | **59.14 ± 9.39** | 70.91 ± 24.78 | 100.00 ± 0.00 | 68.34 ± 6.02 | 100.00 ± 0.0 |
| | 1,1,1 | 82.51 ± 12.48 | **50.33 ± 6.90** | 100.00 ± 0.00 | 96.76 ± 4.72 | 96.17 ± 5.31 |
| Number of layers | | 5 | 5 | 5 | 5 | 5 |
| 4D DF-heat | 100,1,1 | 1.68 ± 0.44 | 5.42 ± 1.59 | 4.04 ± 1.25 | 4.27 ± 0.63 | **1.40 ± 0.33** |
| | 200,1,1 | 1.67 ± 0.33 | 5.41 ± 1.00 | 3.31 ± 1.36 | 3.82 ± 0.59 | **1.51 ± 0.13** |
| | 10,1,1 | **1.88 ± 0.45** | 6.95 ± 0.94 | 4.74 ± 1.58 | 5.16 ± 0.57 | 2.74 ± 1.68 |
| | 10,1,5 | **3.91 ± 1.28** | 8.13 ± 1.37 | 11.70 ± 4.32 | 7.46 ± 2.26 | 4.33 ± 1.93 |
| | 1,1,1 | **2.45 ± 0.44** | 6.58 ± 1.09 | 9.36 ± 1.11 | 5.63 ± 0.68 | 3.66 ± 2.16 |

Table 5: Relative $L^2$ error [%] performance comparison of different balance terms in a smaller number of layers when $N_{\text{pde}} = 1000$. **Bold** indicates best, underline second-best.

## G.3 Different Hyperparameters of MCI Approaches on the Best ic/bc/pde Balance Terms

| | | | | | | | |
|---|---|---|---|---|---|---|---|
| | DF-heat 4D, ic/bc/pde = (200/1/1) | | | | | | |
| | Hyper-parameters | k=0.5, c=1 | **k=1, c=1** | k=2, c=1 | k=3, c=1 | k=4, c=1 | k=5, c=1 |
| | Rel L2 error [%] | 6.92 ± 1.57 | **5.72 ± 0.42** | 10.07 ± 1.73 | 11.78 ± 1.59 | 12.94 ± 1.96 | 12.89 ± 1.19 |
| | DF-heat 6D, ic/bc/pde = (200/1/1) | | | | | | |
| | Hyper-parameters | **k=0.5, c=1** | k=1, c=1 | k=2, c=1 | k=3, c=1 | k=4, c=1 | k=5, c=1 |
| RAD | Rel L2 error [%] | **6.39 ± 1.14** | 10.63 ± 0.93 | 29.52 ± 8.07 | 46.47 ± 27.28 | 66.73 ± 27.86 | 55.58 ± 31.73 |
| | DF-heat 8D, ic/bc/pde = (200/1/1) | | | | | | |
| | Hyper-parameters | **k=0.5, c=1** | k=1, c=1 | k=2, c=1 | k=3, c=1 | k=4, c=1 | k=5, c=1 |
| | Rel L2 error [%] | **67.50 ± 39.83** | 83.59 ± 32.80 | 100.00 ± 0.00 | 77.71 ± 29.27 | 87.37 ± 25.24 | 90.40 ± 19.17 |
| | DF-heat 4D, ic/bc/pde = (200/1/1) | | | | | | |
| | Hyper-parameters | Max_i=1 | **Max_i=3** | Max_i=5 | Max_i=10 | Max_i=15 | Max_i=20 |
| | Rel L2 error [%] | 6.50 ± 1.19 | **2.15 ± 0.25** | 2.20 ± 0.17 | 2.24 ± 0.22 | 2.49 ± 0.45 | 2.37 ± 0.11 |
| | DF-heat 6D, Ic/bc/pde = (100/1/1) | | | | | | |
| | Hyper-parameters | Max_i=1 | **Max_i=3** | Max_i=5 | Max_i=10 | Max_i=15 | Max_i=20 |
| R3 | Rel L2 error [%] | 29.91 ± 35.23 | **4.98 ± 0.18** | 5.31 ± 0.86 | 5.91 ± 0.41 | 6.75 ± 1.02 | 20.55 ± 28.04 |
| | DF-heat 8D, Ic/bc/pde = (200/1/1) | | | | | | |
| | Hyper-parameters | Max_i=1 | Max_i=3 | Max_i=5 | Max_i=10 | **Max_i=15** | Max_i=20 |
| | Rel L2 error [%] | 100.00 ± 0.00 | 100.00 ± 0.00 | 100.00 ± 0.00 | 65.76 ± 42.41 | **65.03 ± 43.11** | 82.97 ± 34.35 |

Table 6: Effect of hyper-parameters on relative $L^2$ error [%] across MCI approaches when the best ic/bc/pde balance terms of DF-heat equations and $N_{\mathrm{pde}} = 1000$ are considered.

## G.4 Different Number of Collocation Points

| Sampling methods | LAS | Random-R | RAD | R3 | $L^\infty$ |
|---|---|---|---|---|---|
| $N_{\mathrm{pde}} = 100$ | **49.26 ± 30.18** | 100.00 ± 0.00 | 100.00 ± 0.00 | 100.00 ± 0.00 | 88.08 ± 23.83 |
| $N_{\mathrm{pde}} = 500$ | **8.62 ± 0.42** | 66.77 ± 40.75 | 93.37 ± 13.24 | 100.00 ± 0.00 | 49.85 ± 40.94 |
| $N_{\mathrm{pde}} = 1,000$ | **6.92 ± 0.31** | 17.63 ± 1.78 | 83.59 ± 32.80 | 100.00 ± 0.00 | 13.21 ± 4.46 |
| $N_{\mathrm{pde}} = 4,000$ | **5.94 ± 0.13** | 15.26 ± 3.26 | 19.49 ± 4.17 | 70.37 ± 37.49 | 7.61 ± 1.61 |
| $N_{\mathrm{pde}} = 10,000$ | **5.90 ± 0.39** | 20.09 ± 0.84 | 16.03 ± 1.94 | 35.45 ± 32.83 | 6.66 ± 1.38 |
| $N_{\mathrm{pde}} = 20,000$ | **5.27 ± 0.53** | 17.77 ± 3.18 | 11.96 ± 1.43 | 35.78 ± 32.21 | 6.64 ± 1.59 |

Table 7: Effect of $N_{\mathrm{pde}}$ on relative $L^2$ error [%] across different sampling methods when ic/bc/pde balance term of DF-heat 8D equation is (200/1/1). **Bold** indicates best, underline second-best.

# H Compatibility Issues with Different Neural Network Architectures

We aimed to experimentally evaluate the compatibility of the proposed adaptive sampling technique with architectures beyond MLPs, including self-attention and modified-MLPs [44]. Detailed descriptions of each architecture are provided in Table 8. Furthermore, to address the spectral bias issue commonly found in MLPs, we performed additional experiments incorporating random Fourier blocks (FB), as detailed in [38]. The FB hyperparameters were set with a Fourier feature scale of 2 and a Fourier block dimension of 64.

Table 8: Parameter configuration for different architectures

| Parameter | MLP | Self-attention | Modified MLP |
|---|---|---|---|
| Activation | *Tanh* | *Tanh* | *Tanh* |
| Embedding dimension | 128 | 128 | 128 |
| Number of layers | 4 | 4 | 4 |
| Multi-head number | N/A | 4 | N/A |
| Fully connected dimension | N/A | 256 | N/A |
| Attention dropout | N/A | 0.1 | N/A |
| Additional encoders $U$, $V$ | N/A | N/A | *Yes* |

The results are summarized in Table 9. All algorithms were evaluated using the default settings provided in the benchmark algorithm papers, and the same applies to our approach, as detailed in the experimental settings described in the main text. Analyzing the results, we observe that the proposed LAS demonstrates high compatibility in terms of relative $L^2$ error. Even in the less favorable architectures, such as MLP and self-attention, LAS achieved the second-best performance. Notably, in scenarios incorporating FB, the proposed LAS consistently exhibited superior compatibility across all cases.

Table 9: Relative $L^2$ error comparison for different architectures across sampling methods. **Bold** indicates best, underline second-best.

| Architecture | LAS | RAD | R3 | $L^\infty$ | Random-R |
|---|---|---|---|---|---|
| MLP | $\underline{2.15 \pm 0.12}$ | $2.65 \pm 0.47$ | $\mathbf{1.84 \pm 0.08}$ | $2.76 \pm 0.26$ | $3.56 \pm 0.20$ |
| MLP + FB | $\mathbf{0.56 \pm 0.14}$ | $0.69 \pm 0.05$ | $\underline{0.61 \pm 0.04}$ | $0.81 \pm 0.18$ | $0.94 \pm 0.06$ |
| Modified MLP | $\mathbf{0.43 \pm 0.10}$ | $\underline{0.51 \pm 0.05}$ | $0.66 \pm 0.07$ | $0.55 \pm 0.08$ | $0.56 \pm 0.07$ |
| Modified MLP + FB | $\mathbf{0.11 \pm 0.04}$ | $0.22 \pm 0.09$ | $0.24 \pm 0.03$ | $0.34 \pm 0.01$ | $\underline{0.21 \pm 0.04}$ |
| Self-attention | $\underline{2.13 \pm 0.13}$ | $2.30 \pm 0.13$ | $2.25 \pm 0.06$ | $\mathbf{1.98 \pm 0.03}$ | $2.91 \pm 0.69$ |
| Self-attention + FB | $\mathbf{1.29 \pm 0.09}$ | $1.53 \pm 0.03$ | $1.52 \pm 0.15$ | $\underline{1.33 \pm 0.12}$ | $1.35 \pm 0.12$ |

# I  Experimental Comparison of the Computational Complexities

A key limitation of the MCMC-based sampling approach lies in its high computational complexity, primarily due to the large number of iterations required to achieve convergence of the sampling distribution. To demonstrate the practical feasibility of this method, we present supporting experimental results.

To evaluate computational complexities, we measured the computational costs for training deep neural networks using each algorithm. Specifically, to validate the scalability of the algorithms, we conducted experiments to analyze their computational requirements in terms of the number of collocation points $N_{\text{pde}}$ and the dimensionality of the PDE.

Based on our observations, the runtime of the sampling algorithms was independent of the specific PDE. Thus, we utilized equations that allowed for a straightforward extension from 1D to 8D in dimensionality. More specifically, we experimented with different sizes of collocation points (100, 1,000, 10,000, 50,000, and 100,000) for both 1D, 2D Burgers' equations, and 4D to 8D DF-heat equations.

As part of the detailed experimental process, we calculated the elapsed time over 1,000 epochs. The measurement was repeated 10 times using 10 different random seeds, and the mean and standard deviation were computed. For additional clarity, the elapsed time was measured excluding auxiliary operations such as saving the model or storing data, focusing solely on the computations required to run the algorithms.

**Hardware specification.** NVIDIA RTX 4090 GPU with 24GB of memory.

**Changes with $N_{\text{PDE}}$.** As $N_{\text{PDE}}$ increases, the computational cost grows for all methods. However, the growth rate varies significantly between methods. Gradient-based algorithms such as LAS and $L^\infty$ show a particularly sharp increase in computational cost as $N_{\text{PDE}}$ grows. This is due to the iteration-intensive nature of their sampling processes. For example, with $N_{\text{PDE}} = 50,000$ in 2D, the computational cost of LAS ($l_{\text{L}} = 10$) reaches 63.98 seconds, whereas simpler methods like Fixed or Random-R remain below 35 seconds. For $N_{\text{PDE}} = 100,000$, LAS and $L^\infty$ run out of memory in the 2D case, highlighting their scalability limitations for very large PDE sample sizes.

**Changes with dimensionality.** Extending from 1D to 8D consistently increases the computational cost for all methods. While simple methods like Fixed or Random-R exhibit a relatively modest increase in cost when transitioning from 1D to 8D, gradient-based methods such as LAS and $L^\infty$ show disproportionately higher computational times. For example, in the 2D case with $N_{\text{PDE}} = 1,000$, LAS ($l_{\text{L}} = 10$) takes 45.74 seconds, compared to only 16.97 seconds in 1D. At $N_{\text{PDE}} = 10,000$, LAS ($l_{\text{L}} = 10$) takes 59.32 seconds in 2D versus 20.43 seconds in 1D.

However, despite the overall computational expense of LAS for higher $l_{\text{L}}$ values, the case of $l_{\text{L}} = 1$ demonstrates significantly lower computational costs, making it relatively practical and scalable. For example, at $N_{\text{PDE}} = 50,000$, LAS ($l_{\text{L}} = 1$) takes 63.98 seconds in 2D, which is manageable compared to the prohibitive 324.49 seconds for $l_{\text{L}} = 10$. Similarly, for smaller $N_{\text{PDE}}$, such as 1,000, LAS ($l_{\text{L}} = 1$) shows competitive runtimes (e.g., 26.62 seconds in 2D). An additional advantage of using $l_{\text{L}} = 1$ is that it avoids the out of memory issues observed for higher values of $l_{\text{L}}$, even in large-scale scenarios such as $N_{\text{PDE}} = 100,000$ in 2D. Furthermore, the relative $L_2$ error reported in our paper uses $l_{\text{L}} = 1$ as the baseline, demonstrating its effectiveness in balancing computational efficiency with accuracy. Thus, LAS ($l_{\text{L}} = 1$) remains a promising candidate for solving PDEs efficiently at scale.

Finally, while gradient-based methods like LAS exhibit high computational costs due to the overhead of gradient computation, future work optimizing the gradient operations could significantly enhance the scalability and practicality of these methods.

Table 10: Computational time (in seconds) for each method across varying problem dimensions. **OOM** denotes an out-of-memory failure during execution.

| Dimension | Method | $N = 100$ | $N = 1,000$ | $N = 10,000$ | $N = 50,000$ | $N = 100,000$ |
|---|---|---|---|---|---|---|
| 1D | Fixed | $2.58 \pm 0.03$ | $2.54 \pm 0.03$ | $3.07 \pm 0.05$ | $11.63 \pm 0.56$ | $24.93 \pm 0.02$ |
| | Random-R | $2.66 \pm 0.10$ | $2.61 \pm 0.01$ | $3.14 \pm 0.04$ | $11.66 \pm 0.05$ | $24.91 \pm 0.04$ |
| | R3 | $2.99 \pm 0.01$ | $2.94 \pm 0.00$ | $3.20 \pm 0.02$ | $11.97 \pm 0.04$ | $25.34 \pm 0.05$ |
| | RAD | $3.40 \pm 0.00$ | $3.46 \pm 0.13$ | $3.99 \pm 0.00$ | $14.92 \pm 0.03$ | $31.81 \pm 0.09$ |
| | LAS ($l_{\mathrm{L}} = 1$) | $3.90 \pm 0.03$ | $3.89 \pm 0.03$ | $5.01 \pm 0.01$ | $21.19 \pm 0.06$ | $46.36 \pm 0.05$ |
| | LAS ($l_{\mathrm{L}} = 5$) | $11.59 \pm 0.03$ | $9.70 \pm 0.02$ | $11.90 \pm 0.16$ | $56.86 \pm 0.06$ | $130.95 \pm 0.08$ |
| | LAS ($l_{\mathrm{L}} = 10$) | $15.55 \pm 0.05$ | $16.97 \pm 0.37$ | $20.43 \pm 0.10$ | $102.52 \pm 0.17$ | $236.95 \pm 0.11$ |
| | $L^{\infty}$ | $28.77 \pm 1.99$ | $29.03 \pm 2.20$ | $37.54 \pm 0.48$ | $197.24 \pm 0.17$ | $448.95 \pm 0.27$ |
| 2D | Fixed | $6.07 \pm 0.15$ | $6.45 \pm 0.14$ | $7.83 \pm 0.03$ | $34.83 \pm 0.04$ | $80.17 \pm 0.05$ |
| | Random-R | $6.22 \pm 0.21$ | $6.54 \pm 0.16$ | $7.85 \pm 0.03$ | $34.87 \pm 0.04$ | $80.26 \pm 0.04$ |
| | R3 | $6.15 \pm 0.19$ | $6.57 \pm 0.22$ | $7.97 \pm 0.03$ | $35.19 \pm 0.03$ | $80.61 \pm 0.06$ |
| | RAD | $7.61 \pm 0.02$ | $8.00 \pm 0.09$ | $10.04 \pm 0.01$ | $44.30 \pm 0.05$ | $99.67 \pm 0.10$ |
| | LAS ($l_{\mathrm{L}} = 1$) | $9.96 \pm 0.26$ | $10.42 \pm 0.31$ | $13.59 \pm 0.03$ | $63.98 \pm 0.08$ | $151.33 \pm 0.11$ |
| | LAS ($l_{\mathrm{L}} = 5$) | $25.67 \pm 0.51$ | $26.62 \pm 0.37$ | $33.40 \pm 0.14$ | $179.69 \pm 0.20$ | **OOM** |
| | LAS ($l_{\mathrm{L}} = 10$) | $44.68 \pm 0.52$ | $45.74 \pm 0.92$ | $59.32 \pm 0.27$ | $324.49 \pm 0.35$ | **OOM** |
| | $L^{\infty}$ | $82.75 \pm 0.88$ | $84.21 \pm 1.68$ | $115.29 \pm 0.21$ | $612.22 \pm 0.53$ | **OOM** |
| 4D | Fixed | $11.64 \pm 0.50$ | $12.01 \pm 0.06$ | $15.06 \pm 0.13$ | $67.23 \pm 0.03$ | $151.58 \pm 0.02$ |
| | Random-R | $11.83 \pm 0.26$ | $12.06 \pm 0.05$ | $15.03 \pm 0.06$ | $67.33 \pm 0.07$ | $151.60 \pm 0.00$ |
| | R3 | $11.89 \pm 0.34$ | $12.23 \pm 0.04$ | $15.43 \pm 0.32$ | $67.58 \pm 0.03$ | $152.03 \pm 0.03$ |
| | RAD | $15.53 \pm 0.70$ | $17.13 \pm 0.98$ | $19.56 \pm 0.05$ | $83.29 \pm 0.02$ | **OOM** |
| | LAS ($l_{\mathrm{L}} = 1$) | $19.17 \pm 0.81$ | $19.73 \pm 0.63$ | $25.33 \pm 0.65$ | $124.24 \pm 0.16$ | **OOM** |
| | LAS ($l_{\mathrm{L}} = 5$) | $42.31 \pm 2.41$ | $42.11 \pm 0.51$ | $61.20 \pm 0.23$ | **OOM** | **OOM** |
| | LAS ($l_{\mathrm{L}} = 10$) | $68.65 \pm 0.70$ | $71.24 \pm 2.18$ | $105.93 \pm 0.27$ | **OOM** | **OOM** |
| | $L^{\infty}$ | $133.81 \pm 1.75$ | $139.14 \pm 2.19$ | $193.54 \pm 0.17$ | **OOM** | **OOM** |
| 6D | Fixed | $17.65 \pm 0.64$ | $18.64 \pm 0.61$ | $21.10 \pm 0.12$ | $96.50 \pm 0.08$ | $218.68 \pm 0.20$ |
| | Random-R | $16.33 \pm 2.07$ | $16.44 \pm 1.66$ | $21.36 \pm 0.07$ | $96.63 \pm 0.03$ | $218.70 \pm 0.09$ |
| | R3 | $18.38 \pm 0.20$ | $18.36 \pm 1.01$ | $21.46 \pm 0.14$ | $96.85 \pm 0.07$ | $219.15 \pm 0.01$ |
| | RAD | $20.69 \pm 0.65$ | $20.05 \pm 0.77$ | $25.75 \pm 0.12$ | $119.51 \pm 0.11$ | **OOM** |
| | LAS ($l_{\mathrm{L}} = 1$) | $25.97 \pm 1.51$ | $27.13 \pm 1.98$ | $34.54 \pm 0.03$ | $179.94 \pm 0.32$ | **OOM** |
| | LAS ($l_{\mathrm{L}} = 5$) | $57.66 \pm 1.18$ | $61.36 \pm 2.76$ | $87.11 \pm 0.13$ | **OOM** | **OOM** |
| | LAS ($l_{\mathrm{L}} = 10$) | $102.97 \pm 3.22$ | $103.50 \pm 5.22$ | $153.10 \pm 0.25$ | **OOM** | **OOM** |
| | $L^{\infty}$ | $183.39 \pm 7.13$ | $195.98 \pm 6.34$ | $275.06 \pm 0.19$ | **OOM** | **OOM** |
| 8D | Fixed | $20.19 \pm 1.93$ | $21.93 \pm 0.38$ | $26.55 \pm 0.25$ | $127.65 \pm 0.16$ | $286.65 \pm 0.16$ |
| | Random-R | $22.08 \pm 0.13$ | $21.84 \pm 0.20$ | $26.95 \pm 0.18$ | $127.68 \pm 0.14$ | $286.61 \pm 0.01$ |
| | R3 | $20.99 \pm 0.04$ | $22.51 \pm 0.31$ | $27.15 \pm 0.09$ | $127.85 \pm 0.14$ | $287.26 \pm 0.05$ |
| | RAD | $26.48 \pm 0.88$ | $26.15 \pm 1.01$ | $32.15 \pm 0.36$ | $157.70 \pm 0.11$ | **OOM** |
| | LAS ($l_{\mathrm{L}} = 1$) | $33.03 \pm 0.99$ | $33.75 \pm 1.63$ | $44.89 \pm 0.15$ | **OOM** | **OOM** |
| | LAS ($l_{\mathrm{L}} = 5$) | $74.12 \pm 2.46$ | $78.77 \pm 1.76$ | $114.37 \pm 0.27$ | **OOM** | **OOM** |
| | LAS ($l_{\mathrm{L}} = 10$) | $128.65 \pm 5.59$ | $134.49 \pm 4.29$ | $201.64 \pm 0.26$ | **OOM** | **OOM** |
| | $L^{\infty}$ | $229.60 \pm 1.97$ | $240.58 \pm 5.01$ | $357.80 \pm 0.14$ | **OOM** | **OOM** |

# J  Additional Results with Varying LAS Hyperparameters

## J.1  Practical Implementation

In general, achieving successful Langevin sampling requires careful selection of hyperparameters (Langevin step size $\tau$, number of Langevin iterations $l_\mathrm{L}$, etc.). While running Langevin dynamics for many iterations with a small step size might allow sampling to be proportional to the actual residual landscape, it can significantly slow down the training speed of the PINN model in practical applications. Thus, we considered the following concepts when setting the hyperparameters, which are crucial for practical utilization of Langevin dynamics in PINN training.

**Adjusting Langevin step size and iteration.** To increase the computational efficiency, we adopted a strategy of increasing the step size $\tau$ and reducing the number of Langevin iterations $l_\mathrm{L}$. We anticipated that the temporal variation of the PINN model, $f_\theta$, would exhibit smooth behavior when utilizing adaptive sampling strategies with a large network size. Consequently, even with fewer Langevin iterations, minimal changes in the loss landscape suggest that the sample trajectory would resemble that of a fixed landscape.

**Normalizing the gradient size.** Since LAS leverages gradient information from the residual landscape, the step size $\tau$ needs to be set even smaller for stiff PDEs, i.e., the step size is dependent on the PDE. Additionally, empirical observations indicate that the gradient of the residual landscape exhibits substantial variations at the beginning of training. In contrast, towards the end of training, the residual landscape is characterized by relatively small gradients. This discrepancy restricts the movement of sample points, thereby making it challenging to secure reliable quality. To address these challenges, we normalized the magnitude of all residual gradients at each iteration relative to the largest residual gradient.

## J.2  Variation of $\beta$ and $l_\mathrm{L}$ with Fixed $\tau = 0.002$

1. **Instability of performance for small $\beta$ values as the layer increases:** As the layer depth increases, small $\beta$ values lead to unstable performance. For instance, in layer 4, a small $\beta = 0.001$ results in a relatively stable error value of $1.18 \pm 0.23$ at $l_\mathrm{L} = 1$, whereas in layer 10, the error rises significantly to $8.20 \pm 15.63$. This pattern suggests that small values of $\beta$ hinder performance stability in deeper layers.

2. **Increased instability with higher $l_\mathrm{L}$ values:** Generally, the performance deteriorates as $l_\mathrm{L}$ increases, particularly for small $\beta$ values. For example, in layer 6 with $\beta = 0.001$, the relative error increases from $0.58 \pm 0.08$ at $l_\mathrm{L} = 1$ to $22.73 \pm 22.08$ at $l_\mathrm{L} = 20$. This indicates that excessive Langevin iterations could lead to performance instability, especially when the concentration parameter $\beta$ is low.

Overall, the impact of $\tau$ and $l_\mathrm{L}$ on both relative $L^2$ error and variance is limited, indicating robustness of the method across a range of parameters. This robustness simplifies Langevin hyperparameter tuning, making the approach more practical for real-world applications.

## J.3  Variation of $\tau$ and $l_\mathrm{L}$ with Fixed $\beta = 0.2$

1. **Stability across $\tau$ values:** The relative $L^2$ error exhibits minor fluctuations across different values of $\tau$ for each layer suggesting a negligible dependency on $\tau$.

2. **Limited impact of $l_\mathrm{L}$:** While increasing $l_\mathrm{L}$ slightly reduces the variance of $L^2$ error in some cases, the effect is not consistent across layers showing only marginal improvement.

As a result, the main takeaway is that small values of $\beta$ are prone to instability as the layer depth increases and $l_\mathrm{L}$ becomes large. Conversely, higher $\beta$ values can ensure stable performance, even with varying $l_\mathrm{L}$ values. In shallow layers, however, lower $\beta$ values can be beneficial, providing a more precise error at lower $l_\mathrm{L}$ values.

| Layer and concentration parameter $\beta$ | | Langevin iteration $l_{\mathrm{L}}$ | | | |
|---|---|---|---|---|---|
| Layer | $\beta$ | 1 | 5 | 10 | 20 |
| Layer 4 | 0.001 | $1.18 \pm 0.23$ | $1.63 \pm 0.40$ | $1.66 \pm 0.20$ | $6.34 \pm 2.69$ |
| | 0.05 | $1.53 \pm 0.36$ | $1.41 \pm 0.43$ | $1.27 \pm 0.21$ | $1.25 \pm 0.16$ |
| | 0.1 | $1.08 \pm 0.37$ | $\mathbf{0.93 \pm 0.09}$ | $1.23 \pm 0.23$ | $1.06 \pm 0.13$ |
| | 0.3 | $2.50 \pm 0.28$ | $2.30 \pm 0.34$ | $2.54 \pm 0.35$ | $2.51 \pm 0.36$ |
| | 0.4 | $2.80 \pm 0.60$ | $3.14 \pm 0.22$ | $3.14 \pm 0.49$ | $2.90 \pm 0.44$ |
| Layer 6 | 0.001 | $0.58 \pm 0.08$ | $10.89 \pm 19.65$ | $10.10 \pm 17.80$ | $22.73 \pm 22.08$ |
| | 0.05 | $0.62 \pm 0.11$ | $0.64 \pm 0.12$ | $0.65 \pm 0.12$ | $0.67 \pm 0.15$ |
| | 0.1 | $0.74 \pm 0.12$ | $0.58 \pm 0.08$ | $0.64 \pm 0.07$ | $\mathbf{0.58 \pm 0.03}$ |
| | 0.3 | $1.05 \pm 0.27$ | $1.15 \pm 0.21$ | $1.16 \pm 0.11$ | $1.33 \pm 0.13$ |
| | 0.4 | $1.47 \pm 0.10$ | $1.49 \pm 0.08$ | $1.48 \pm 0.22$ | $1.58 \pm 0.13$ |
| Layer 8 | 0.001 | $0.65 \pm 0.14$ | $1.25 \pm 0.73$ | $10.66 \pm 19.50$ | $22.14 \pm 24.51$ |
| | 0.05 | $0.88 \pm 0.34$ | $\mathbf{0.46 \pm 0.05}$ | $0.59 \pm 0.10$ | $1.44 \pm 0.81$ |
| | 0.1 | $0.59 \pm 0.14$ | $0.58 \pm 0.08$ | $0.68 \pm 0.35$ | $0.86 \pm 0.56$ |
| | 0.3 | $1.06 \pm 0.16$ | $1.15 \pm 0.21$ | $1.08 \pm 0.24$ | $0.92 \pm 0.11$ |
| | 0.4 | $1.12 \pm 0.25$ | $1.09 \pm 0.19$ | $1.14 \pm 0.17$ | $1.12 \pm 0.27$ |
| Layer 10 | 0.001 | $8.20 \pm 15.63$ | $23.14 \pm 20.74$ | $40.14 \pm 19.78$ | $41.79 \pm 20.50$ |
| | 0.05 | $\mathbf{0.46 \pm 0.17}$ | $17.33 \pm 21.03$ | $17.21 \pm 21.02$ | $30.08 \pm 24.07$ |
| | 0.1 | $0.63 \pm 0.20$ | $0.73 \pm 0.23$ | $0.70 \pm 0.22$ | $10.23 \pm 18.99$ |
| | 0.3 | $0.94 \pm 0.18$ | $0.98 \pm 0.15$ | $0.83 \pm 0.25$ | $1.02 \pm 0.30$ |
| | 0.4 | $0.94 \pm 0.15$ | $0.96 \pm 0.22$ | $0.97 \pm 0.37$ | $1.12 \pm 0.33$ |

| Layer and Langevin step size $\tau$ | | Langevin iteration $l_{\mathrm{L}}$ | | | |
|---|---|---|---|---|---|
| Layer | $\tau$ | 1 | 5 | 10 | 20 |
| Layer 4 | 0.0001 | $1.98 \pm 0.35$ | $2.12 \pm 0.14$ | $\mathbf{1.66 \pm 0.06}$ | $1.74 \pm 0.19$ |
| | 0.0005 | $1.81 \pm 0.13$ | $1.74 \pm 0.40$ | $1.77 \pm 0.34$ | $1.74 \pm 0.35$ |
| | 0.001 | $2.35 \pm 0.37$ | $1.85 \pm 0.25$ | $1.98 \pm 0.18$ | $1.68 \pm 0.24$ |
| | 0.005 | $1.96 \pm 0.15$ | $2.08 \pm 0.18$ | $1.84 \pm 0.22$ | $1.71 \pm 0.17$ |
| | 0.01 | $1.93 \pm 0.28$ | $1.92 \pm 0.09$ | $1.48 \pm 0.09$ | $1.79 \pm 0.09$ |
| Layer 6 | 0.0001 | $0.96 \pm 0.11$ | $0.89 \pm 0.06$ | $0.96 \pm 0.11$ | $1.00 \pm 0.14$ |
| | 0.0005 | $0.92 \pm 0.13$ | $\mathbf{0.69 \pm 0.34}$ | $1.14 \pm 0.23$ | $1.17 \pm 0.25$ |
| | 0.001 | $0.85 \pm 0.07$ | $1.00 \pm 0.18$ | $0.70 \pm 0.09$ | $0.96 \pm 0.03$ |
| | 0.005 | $0.85 \pm 0.07$ | $0.82 \pm 0.22$ | $0.97 \pm 0.10$ | $0.78 \pm 0.37$ |
| | 0.01 | $0.90 \pm 0.04$ | $0.90 \pm 0.06$ | $1.03 \pm 0.16$ | $0.93 \pm 0.06$ |
| Layer 8 | 0.0001 | $\mathbf{0.62 \pm 0.04}$ | $0.79 \pm 0.11$ | $0.92 \pm 0.10$ | $0.82 \pm 0.05$ |
| | 0.0005 | $0.81 \pm 0.02$ | $0.85 \pm 0.12$ | $0.78 \pm 0.07$ | $0.75 \pm 0.06$ |
| | 0.001 | $0.91 \pm 0.13$ | $0.82 \pm 0.02$ | $0.71 \pm 0.04$ | $0.77 \pm 0.10$ |
| | 0.005 | $0.85 \pm 0.07$ | $0.83 \pm 0.04$ | $0.64 \pm 0.06$ | $0.63 \pm 0.08$ |
| | 0.01 | $0.64 \pm 0.14$ | $0.77 \pm 0.17$ | $0.88 \pm 0.13$ | $0.81 \pm 0.09$ |
| Layer 10 | 0.0001 | $\mathbf{0.41 \pm 0.18}$ | $0.75 \pm 0.08$ | $0.82 \pm 0.02$ | $0.68 \pm 0.12$ |
| | 0.0005 | $0.82 \pm 0.13$ | $0.64 \pm 0.04$ | $0.68 \pm 0.07$ | $0.82 \pm 0.06$ |
| | 0.001 | $0.91 \pm 0.21$ | $0.74 \pm 0.19$ | $0.87 \pm 0.25$ | $0.59 \pm 0.05$ |
| | 0.005 | $0.64 \pm 0.04$ | $0.53 \pm 0.32$ | $0.67 \pm 0.05$ | $0.64 \pm 0.12$ |
| | 0.01 | $0.56 \pm 0.29$ | $0.65 \pm 0.13$ | $0.71 \pm 0.14$ | $0.61 \pm 0.09$ |

Table 11: Relative $L^2$ error for varying $\beta$ and $\tau$ and $l_{\mathrm{L}}$ across different layers of the Allen–Cahn equation.

