# OpenReview forum: "Mitigating Instability in High Residual Adaptive Sampling for PINNs via Langevin Dynamics"
_NeurIPS.cc/2025/Conference — NeurIPS 2025 spotlight_

### Official Review · Reviewer_Jy8K · 2025-06-23

**Clarity:** 4
**Significance:** 4
**Originality:** 3
**Rating:** 4
**Confidence:** 4

**Summary:**

This paper introduces a mechanism for training PINNs using Langevin sampling to learn the distribution of residuals. The authors provide theoretical guarantees on the convergence properties of the method (i.e. distributional convergence and flatness). A comprehensive set of experiments were conducted, and LAS shows state-of-the-art results in six out of ten benchmarks, including the notably challenging high-dimensional PDEs.

Although I think this work is interesting from an experimental perspective and well-informed in the existing literature, I find the theoretical aspects of this paper concerning. I am willing to increase my score if the authors provide improvements in the proofs in a rebuttal.

**Questions:**

* Can the authors clarify the inconsistency between propositions 3.1 and 3.2, and assumption 3.1? There seems to be a contradiction.
* Every theorem/proposition, by convention, should list all the underlying assumptions explicitly instead of making the reader guess which assumptions hold.
* I think the authors are missing a proof when bridging assumptions 3.2 and 3.3. Heavy-tailed distributions, by definition, prevent tight concentration, so these two cannot simultaneously hold. What are the conditions that the weights or $\beta$ must satisfy for both assumptions to hold?
* Theorem 3.1 also seems unclear. If $\max\|\phi\|\sim D^{1/\alpha}$, then $(\max\|\phi\|)^2\sim D^{2/\alpha}\sim\lambda_{max}(\Sigma)$, so I don't see why you argue that $(\max\|\phi\|)^2 >> \lambda_{max}(\Sigma)$

**Ethical Concerns:**

["NO or VERY MINOR ethics concerns only"]

**Final Justification:**

Based on the rebuttal, the authors proposed improvements in the mathematical foundation of their work and posted proofs that I verified are correct. The consistency in their assumptions was justified in the rebuttal, and I expect to see this change in the paper.

**Limitations:**

A practical concern is that the benchmarking methods require fewer neural network evaluations compared to the proposed method, which involves solving an SDE with neural network evaluations at every time step. But I appreciate that the authors addressed this in the paper.

**Quality:**

2

**Strengths And Weaknesses:**

**Strengths**

* I believe this paper explores some interesting directions that improve the training of PINNs. The search for collocation points for PINNs is relevant, especially in high dimensions.
* The assumptions (3.1, 3.2, 3.3) are well-formulated and fairly weak so that they don't compromise the generalizability of the theoretical results.

**Weaknesses**

* This paper, although theoretically correct, is inconsistent with continuous-time and discrete-time formulations of Langevin dynamics. For example, Equation (4.1) is the Euler-Maruyama discretization of the Langevin equation $dX_t=\nabla_{x}|R_{\theta}(x_n^l)|^2+\beta dB_t$. I should expect to see both formulations in the paper.
* Although Theorem 4.1 is correct, its proof is wrong. Moreover, in a continuous-time FP equation, I should not expect to see any discretization in time ($\tau$) in the proof. In the rebuttal, I want to see the authors use the continuous-time SDE formulation in the proof. I recommend the following sources [1] [2].
* There are serious inconsistencies in the structure of the residual. In the proof of Proposition 3.1, you write that $R_\theta(x)=\theta^T\phi(x)$, which is different from Equation (3.3) in Assumption 3.1. The proof of Proposition 3.2 has the same issue.
* Theorem 3.1 also seems wrong (see questions)

[1] Jordan, R., D. Kinderlehrer, and F. Otto (1998). “The Variational Formulation of the Fokker–Planck Equation”. SIAM Journal on Mathematical Analysis, vol. 29, no. 1, pp. 1–17.

[2] Ambrosio, L., N. Gigli, and G. Savaré (2008). “Gradient Flows in Metric Spaces and in the Space of Probability Measures”. 2nd. Springer.

---

> ### Author Rebuttal · Authors · 2025-07-28
>
> ## Response to Q1: Discrete vs. Continuous-Time Langevin Dynamics
>
> **Summary:
> We acknowledge the importance of presenting both discrete- and continuous-time Langevin dynamics. We will explicitly include both formulations and their theoretical connection in the revised version.**
>
> We thank the reviewer for pointing out the inconsistency between the discrete-time and continuous-time Langevin formulations. We agree with the reviewer that both discrete-time and continuous-time formulations should be presented for consistency. We will include the full derivation in the appendix and summarize the key steps in the main text for clarity.
>
> Our experiments implement the following discrete-time Langevin dynamics via the Euler–Maruyama scheme. For $l \in \mathbb{N}$:
>
> $$\mathbf{x_{\textit{l+1}}} = \mathbf{x_{\textit{l}}} + \frac{\tau}{2} \nabla_\mathbf{x} |R_\theta(\mathbf{x_{\textit{l}}})|^2 + \beta \sqrt{\tau} z_l,\quad z_l \sim \mathcal{N}(0, I).$$
>
> We clarify that $\mathbf{x} = (t,x)$ (time, space) denotes the PDE domain, and we use $s$ to index continuous time for clarity.
>
> As the step size $\tau \to 0$, under mild regularity conditions such as Lipschitz continuity of the gradient, the above discrete-time process converges in distribution to the following continuous-time stochastic differential equation (SDE):
>
> $$
> d\mathbf{x_{\textit{s}}} = \frac{1}{2} \nabla_{\mathbf{x}} |R_\theta(\mathbf{x_{\textit{s}}})|^2ds + \beta dB_s.
> $$
>
>
>
> where $B_s$ denotes standard Brownian motion. This standard continuous-time limit justifies our use of the Fokker–Planck equation. We will nonetheless include both formulations for clarity and completeness.
>
> The corresponding Fokker–Planck equation describing the time evolution of the probability density $p_s(x)$ is:
>
> $$
> \frac{\partial p_s(\mathbf{x})}{\partial s} = -\nabla_\mathbf{x} \cdot \left( \frac{1}{2} \nabla_\mathbf{x} |R_\theta(\mathbf{x})|^2 p_s(\mathbf{x}) \right) + \frac{\beta^2}{2} \Delta_\mathbf{x} p_s(\mathbf{x})
> $$
>
> whose stationary distribution satisfies:
>
> $$
> p_\infty(\mathbf{x}) \propto \exp \left( -\frac{|R_\theta(\mathbf{x})|^2}{\beta^2} \right)
> $$
>
> We will provide the full derivation in the appendix and highlight its connection to gradient flows in the space of probability measures [1,2].
>
> ---
>
> ## Response to Q2: Consistency of Residual Structure Across Assumption 3.1 and Propositions
>
> **Summary:
> We clarify that the apparent inconsistency arises from an implicit linearization in the residual formulation. We will make this assumption explicit and ensure consistency across the assumptions and propositions.**
>
> We recognize that the formulation of $R_\theta(\mathbf{x})$ in Assumption 3.1 and its use in Proposition 3.1, 3.2 appear inconsistent at first glance. We take this opportunity to clarify the derivation and underlying rationale.
>
> We first clarify the structure of the residual function $R_\theta(\mathbf{x}) = a(\theta)^\top \phi(\mathbf{x}; \theta)$ introduced in Assumption 3.1, and actively utilize it in our response.
>
> In general, a residual function can be expressed as:
>
> $$
> R_\theta(\mathbf{x}) = (\partial_t f_\theta + N_x[f_\theta])(\mathbf{x})
> $$
>
> where $\partial_t$ denotes the partial derivative with respect to the PDE’s time domain, and $N_x$ is a nonlinear operator over the spatial domain.
>
> We define a functional operator $g$ acting on the function space, such that:
>
> $$
> R_\theta(\mathbf{x}) = (g(f_\theta))(\mathbf{x})
> $$
>
> Then, for a small perturbation $\Delta\theta$, we approximate:
>
> $$
> g(f_{\theta+\Delta\theta}) \approx g(f_\theta) + D_g(f_\theta)(f_{\theta+\Delta\theta} - f_\theta)
> $$
>
> where $D_g$ is the Fréchet derivative of the operator $g$. Suppose that $g$ admits the following expansion:
>
> $$
> g(f) = G(f, \partial_t f, \partial_x f, \partial_{tt} f, \partial_{tx} f, \dots)
> $$
>
> for some smooth function $G$. Then the perturbation becomes:
>
> $$
> D_g(f_\theta)(f_{\theta+\Delta\theta} - f_\theta) \approx g(f_{\theta+\Delta\theta}) - g(f_\theta) \approx \sum_k \frac{\partial G}{\partial (\partial_k f_\theta)} \Delta(\partial_k f_\theta) \equiv a(\theta)^\top \phi
> $$
>
> We assume that $a(\theta)$ is constant or locally linear, enabling a unified linearized form of the residual. This simplification ensures consistency across Assumption 3.1 and Propositions 3.1–3.2, and underpins our curvature analysis.
>
> To provide further intuition, we illustrate this within the context of two representative PDEs:
>
> - **Convection equation**:
>  $g(f) = \partial_t f + \mu \partial_x f$ → $a(\theta) = (0, 1, \mu, 0, 0, \dots)$ is constant.
>
> - **Burgers’ equation**:
>   $g(f) = \partial_t f + f \partial_x f - \frac{0.01}{\pi} \partial_{xx} f$ → $a(\theta) = (\partial_x f_\theta, 1, f_\theta, 0,0,0,-\frac{0.01}{\pi},\dots)$ is $\theta$-dependent.
>
> Under the assumption that the neural network $f_\theta$ is sufficiently expressive and smooth with respect to $\theta$, we can locally linearize $a(\theta)$ around a fixed $\theta$.
>
>
> When $a(\theta)$ is affine, its Jacobian $J_a=\left(\frac{\partial{a_i}}{\partial\theta_j}\right)$ is constant and Hessian $H_\theta(a) = 0$, where the derivatives are defined elementwise.
>
> We consider the weighted loss:
>
> $$
> \mathcal{L}(\theta) = \sum_{n=1}^N w_n |a(\theta)^\top \phi_n|^2,\quad \phi_n := \phi(\mathbf{x}_n)
> $$
>
> Then the Hessian is:
>
> $$
> H_\theta \mathcal{L}(\theta) = 2 \sum_{n=1}^N w_n \left[ (J_a^\top \phi_n)(J_a^\top \phi_n)^\top + (a^\top \phi_n) \sum_{i=1}^D \phi_{n,i} H_\theta(a_i) \right]
> $$
>
> If $a(\theta)$ is linear, the second term vanishes, and we obtain:
>
> $$
> H_\theta \mathcal{L}(\theta) = 2 J_a^\top \left( \sum_{n=1}^N w_n \phi_n \phi_n^\top \right) J_a
> $$
>
> This Hessian structure allows us to recover similar proposition results as in the paper, with the inclusion of the Jacobian term $J_a.$ We will revise the manuscript to explicitly include this structural assumption in Assumption 3.1 and ensure that its implications are consistently reflected in all related propositions.
>
> ---
>
> ## Response to Q3: Consistency of Assumptions 3.2 and 3.3 in the $\beta \to 0$ Regime
>
> **Summary:
> While the inequality in Assumption 3.3 may not generally hold, we demonstrate that it becomes statistically valid in the low-temperature, high-dimensional regime. We will clarify this condition in the revised version.**
>
> We thank the reviewer for pointing out a potential tension between Assumptions 3.2 (heavy-tailed feature norms) and 3.3 (residual maximizer closely follows feature norm maximizer), especially in relation to the inequality $\max \||\phi(\mathbf{x})\||^2 \gg \lambda_{\max}(\Sigma_\phi)$.
>
> We agree with the reviewer that this inequality may not hold in general under isotropic feature settings. However, under a highly structured regime, our analysis ensures that both assumptions can be approximately satisfied:
>
> (i) the **low-temperature limit** $\beta \to 0$ of the residual-weighted Langevin sampler, and
> (ii) the **high-dimensional setting** where heavy-tailed distributions emerge naturally from overparameterized neural networks.
>
> In this regime, Assumption 3.3 becomes statistically valid and meaningful. Our residual function is given by:
>
> $$
> R_\theta(x) = a(\theta)^\top \phi(\mathbf{x}) \quad \Rightarrow \quad |R_\theta(x)|^2 = \phi(\mathbf{x})^\top a(\theta) a(\theta)^\top \phi(\mathbf{x}) = \phi(\mathbf{x})^\top Q \phi(\mathbf{x}),
> $$
>
> where $Q$ is a rank-one positive semi-definite matrix. For $\phi(\mathbf{x})$ aligned with $a(\theta)$, this inner product approximates $\||\phi(\mathbf{x})\||^2$. Thus, the key question becomes:
>
> $$
> \text{Under what conditions does } \phi(\mathbf{x})^\top Q \phi(\mathbf{x}) \approx \||\phi(\mathbf{x})\||^2 \text{ hold?}
> $$
>
> 1. **Directional distortion by residual-weighted sampling**:
>    In the limit $\beta \to 0$, Langevin dynamics increasingly samples from regions where $R_\theta(\mathbf{x})$ is large. Since $R_\theta(\mathbf{x})$ is linear in $\phi(\mathbf{x})$, this implicitly biases the sampling toward feature vectors aligned with $a(\theta)$ and with large norm.
>
> 2. **High-dimensional ($D \gg 1$) feature decorrelation**:
>    In high-dimensional settings, $\phi(\mathbf{x})$ vectors are nearly orthogonal. Therefore, the only way for $|R_\theta(\mathbf{x})|^2$ to be large is for $\phi(\mathbf{x})$ to be both high-norm and well-aligned, which makes $\arg\max |R_\theta(\mathbf{x})|^2 \approx \arg\max \||\phi(\mathbf{x})\||^2$ statistically plausible.
>
> ---
>
> This assumption is both conceptually sound and crucial for the curvature analysis. Specifically:
>
> - In the **low-temperature regime**, the empirical Hessian becomes:
>
> $$
> H_\theta \mathcal{L} \approx 2 w_{\max} J_a^\top \phi_{\max} \phi_{\max}^\top J_a, \quad \phi_{\max} = \max \||\phi(\mathbf{x})\||
> $$
>
> - In contrast, under **uniform sampling**:
>
> $$
> H_\theta \mathcal{L} \approx 2 J_a^\top \Sigma_\phi J_a, \quad \Sigma_\phi = \mathbb{E}_\mathbf{x}[\phi(\mathbf{x})\phi(\mathbf{x})^\top]
> $$
>
> - For feature vectors with heavy-tailed norms (Assumption 3.2), one can show:
>
> $$
> \phi_{\max}^2 \sim D^{2/\alpha}, \quad \lambda_{\max}(\Sigma_\phi) \sim \frac{D^{2/\alpha}}{\log D}
> $$
>
> which yields:
>
> $$
> \|\| \phi(\mathbf{x_{\max}}) \|\|^2  \approx \phi_{\max}^2 \gg \lambda_{\max}(\Sigma_\phi), \quad \mathbf{x_{\max}} =\arg\max |R_{\theta}(\mathbf{x})|^2
> $$
>
>
> as $D\to\infty$. This is supported by extreme value theory, where the maximum grows faster than the mean energy [3–5].
>
> *Appendix A.3 (Fig. 7)* empirically shows increasing alignment between $\arg\max R_\theta$ and $\arg\max \|\|\phi\|\|$ as model size increases, supporting Assumption 3.3.
>
> We will revise the manuscript to make this connection explicit and emphasize that the inequality is not an inconsistency but rather a natural consequence of our modeling and sampling design.
>
>
> ---
>
> [1] Jordan et al., SIAM J. Math. Anal., 1998.
> [2] Ambrosio et al., Springer, 2008.
> [3] Vershynin, *High-Dimensional Probability*, Cambridge Univ. Press, 2018.
> [4] Bobkov & Ledoux, Geom. Funct. Anal., 2000.
> [5] Dasgupta, FOCS, 1999.

---

> ### Comment · Reviewer_Jy8K · 2025-08-03
>
> Many thanks to the authors for the detailed response. The continuous-time SDE formulation is correct. And the claim in Theorem 3.1, $\lambda_{\max}(\Sigma)\sim D^{2/\alpha}/\log D$ can be justified under very specific assumptions (e.g. heavy-tails, weak correlation). I trust that the authors will explain this in more detail in the final draft of the paper.
>
> However, I have the following concerns:
> * The insights from the linearization of the residual rely on the assumption that the network is locally linear in parameters. While this may hold in settings where the neural network is shallow (at most one hidden layer) and the training isn't non-convex, it is unlikely to apply to deep architectures used in practice. The authors do not discuss these limitations in the paper.
> * Theorem 4.1 is still stated without proof. The authors cite the literature I provided in the review, but this does not substitute for a formal or sketched derivation in the context of their setup. So I am still concerned about the authors' familiarity with the Langevin equation, as it underpins the theoretical guarantees of their method. Can the authors provide a proof of Theorem 4.1 during this discussion phase? I would like to see this proof in the final draft as well.

---

> > ### Author Response · Authors · 2025-08-03
> >
> > **Dear Reviewer Jy8K**
> >
> > Thank you again for your active and constructive feedback. We’re glad that parts of our previous response addressed some of your concerns. Below, we respond to the remaining points in detail:
> >
> > ## 1. On the local linearization of the residual and the role of $a(\theta)$
> >
> > We thank the reviewer for raising this point. We treat $a(\theta)$ as a local linearization of the residual—i.e., $\(R_\theta(\mathbf{x})\approx a(\theta)^\top \phi(\mathbf{x})\)$—which makes curvature analysis tractable.  This view is loosely related to operator-theoretic embeddings (e.g., Koopman), and we assume that $a(\theta)$ varies smoothly in a neighborhood of the current iterate. We will clarify in the revised draft that this is an approximation—without attempting to compute $a(\theta)$ explicitly—and note regimes where it may weaken (e.g., extremely deep or non-smooth parameterizations), Assumption 3.1 will be revised accordingly.
> >
> > ---
> >
> > ## 2. On the formal derivation of Theorem 4.1
> >
> > Starting from the discretized Langevin dynamics:
> >
> > $$\mathbf{x_{\textit{l+1}}} = \mathbf{x_{\textit{l}}} + \frac{\tau}{2} \nabla_\mathbf{x} |R_\theta(\mathbf{x_{\textit{l}}})|^2 + \beta \sqrt{\tau} z_l,\quad z_l \sim \mathcal{N}(0, I).$$
> >
> > we consider the continuous-time(CT) limit as the step size $\tau \to 0$. This leads to the CT Langevin dynamics, for which we provide a full derivation of Theorem 4.1 using the standard Fokker–Planck formalism.
> >
> > Before the formal proof, we assume the Langevin dynamics (SDE) is well-posed, with drift $\nabla_{\mathbf{x}} |R_\theta(\mathbf{x})|^2$ being locally Lipschitz and satisfying a linear growth condition.
> >
> > **Fokker–Planck Equation**
> >
> > Let $p_s(\mathbf{x})$ denote the probability density of the process $\mathbf{x}_s$, following Langevin dynamics:
> >
> > $$
> > d\mathbf{x_{\textit{s}}} = \frac{1}{2} \nabla_{\mathbf{x}} \|R_\theta(\mathbf{x_{\textit{s}}})\|^2 ds + \beta dB_s,
> > $$
> >
> > Note that the Langevin dynamics here follow **gradient ascent** of the residual magnitude, deliberately inducing drift toward high-residual regions. The time evolution of $p_s(\mathbf{x})$ is described by the Fokker–Planck equation:
> >
> > $$
> > \frac{\partial p_s(\mathbf{x})}{\partial s} = -\nabla_{\mathbf{x}} \cdot \left( \frac{1}{2} \nabla_{\mathbf{x}} \|R_\theta(\mathbf{x})\|^2 \cdot p_s(\mathbf{x}) \right) + \frac{\beta^2}{2} \Delta_{\mathbf{x}} p_s(\mathbf{x}),
> > $$
> >
> > where $\Delta_{\mathbf{x}}$ is the Laplacian in $\Omega=[0,T] \times \mathcal{X}$. The first term encodes drift toward high-residual regions, while the second accounts for isotropic diffusion.
> >
> > ---
> >
> > **Stationary Solution**
> >
> > At stationarity, we impose $\frac{\partial p_s}{\partial s} = 0$, yielding the equation:
> >
> > $$
> > 0 = -\nabla_{\mathbf{x}} \cdot \left( \frac{1}{2} \nabla_{\mathbf{x}} \|R_\theta\|^2 \cdot p_\infty(\mathbf{x}) \right) + \frac{\beta^2}{2} \Delta_{\mathbf{x}} p_\infty(\mathbf{x})
> > $$
> >
> > We propose the following Gibbs stationary distribution:
> >
> > $$
> > p_\infty(\mathbf{x}) = \frac{1}{Z} \exp\left( \frac{\|R_\theta(\mathbf{x})\|^2}{\beta^2} \right)
> > $$
> >
> > To verify this, we compute:
> >
> > - Gradient:
> >   $$
> >   \nabla_{\mathbf{x}} p_\infty = \frac{1}{\beta^2} \nabla_{\mathbf{x}} \|R_\theta\|^2 \cdot p_\infty
> >   $$
> >
> > - Laplacian:
> >   $$
> >   \Delta_{\mathbf{x}} p_\infty = \left( \frac{1}{\beta^2} \Delta_{\mathbf{x}} \|R_\theta\|^2 + \frac{1}{\beta^4} \|\|\nabla_{\mathbf{x}} \|R_\theta\|^2\|\|^2 \right) \cdot p_\infty
> >   $$
> >
> > Substituting into the Fokker–Planck equation and simplifying gives:
> >
> > $$
> > -\nabla_{\mathbf{x}} \cdot \left( \frac{1}{2} \nabla_{\mathbf{x}} \|R_\theta(\mathbf{x})\|^2 \cdot p_\infty(\mathbf{x}) \right) + \frac{\beta^2}{2} \Delta_{\mathbf{x}} p_\infty(\mathbf{x})
> > $$
> >
> > $$
> > = -\frac{1}{2} \left( \Delta_{\mathbf{x}} \|R_\theta(\mathbf{x})\|^2 \cdot p_\infty(\mathbf{x}) + \nabla_{\mathbf{x}} \|R_\theta(\mathbf{x})\|^2 \cdot \nabla_{\mathbf{x}} p_\infty(\mathbf{x}) \right) + \frac{\beta^2}{2} \Delta_{\mathbf{x}} p_\infty(\mathbf{x})
> > $$
> >
> > Using
> >
> > $$
> > \nabla_{\mathbf{x}} p_\infty(\mathbf{x}) = \frac{1}{\beta^2} \nabla_{\mathbf{x}} \|R_\theta(\mathbf{x})\|^2 \cdot p_\infty(\mathbf{x}),
> > $$
> >
> > we obtain:
> >
> > $$
> > = -\frac{1}{2} \left( \Delta_{\mathbf{x}} \|R_\theta(\mathbf{x})\|^2 + \frac{1}{\beta^2} \|\| \nabla_{\mathbf{x}} \|R_\theta(\mathbf{x})\|^2 \|\|^2 \right) p_\infty(\mathbf{x}) + \frac{1}{2} \left( \Delta_{\mathbf{x}} \|R_\theta(\mathbf{x})\|^2 + \frac{1}{\beta^2} \|\| \nabla_{\mathbf{x}} \|R_\theta(\mathbf{x})\|^2 \|\|^2 \right) p_\infty(\mathbf{x}) = 0
> > $$
> >
> > The core concern seems to be whether the residual-based solution is normalizable—i.e., whether unbounded residual growth as $\|\|\mathbf{x}\|\| \to \infty$ leads to exponentially heavy tails. In our case, since the domain $\Omega$ is **bounded and compact**, the distribution remains normalizable, and the stationary solution is well-defined.
> >
> > ---
> >
> > **Final Result**
> >
> > Therefore, the stationary distribution of the Langevin dynamics is:
> >
> > $$
> > p_\infty(\mathbf{x}) \propto \exp\left( \frac{\|R_\theta(\mathbf{x})\|^2}{\beta^2} \right)
> > $$

---

> > > ### Comment · Reviewer_Jy8K · 2025-08-03
> > >
> > > Many thanks to the authors for the follow-up. My concerns about the theoretical contribution of the paper has been addressed, so I increase my score.

---

> > > > ### Author Response · Authors · 2025-08-03
> > > >
> > > > Your proactive engagement has significantly contributed to improving the clarity and overall quality of our manuscript. We sincerely appreciate your thoughtful and insightful feedback. We also thank you for your generous support, even though some parts of our draft were initially underdeveloped or insufficiently explained.

---

### Official Review · Reviewer_b6R2 · 2025-07-01

**Clarity:** 4
**Significance:** 3
**Originality:** 3
**Rating:** 5
**Confidence:** 3

**Summary:**

This paper introduces a Langevin dynamics-based Adaptive
 Sampling (LAS) for the solution of PDEs under the PINN framework. This work is part of a wider literature on adaptive sampling strategies for PINNs colloration point residuals, roughly divided into distribution based sampling and high-residual based sampling. The authors focus on the latter, proceeding by providing a systematical analysis of the theoretical properties of high residual sampling and their connection to learning instability. They identify sharp gradients as one the min source of instability and suggest the use of Langevin dynamics to update the residuals. This is a change from previous approaches based on sampling independent residuals from a target distribution at every iteration. The use of noise in the LAS step has a smoothing effect which the authors argue leads to flatter local optima and improved algorithm stability.

**Questions:**

As I mentioned, the paper looks good. However, in my opinion it critically lacks two things:
1. A discussion on the computational requirements. This should involve both comments on the hardware used, and viability of the algorithm on other kind of hardware (e.g. smaller GPUs, CPUs). Moreover, it should mention the computational complexity (empirical or theoretical) and how the algorithm performs against the baselines. Part of this is present in the appendix but, in my opinion, it is very important to give visibility to this issues in the main body with a high-level discussion. Details can be provided in the appendix.
2. The presentation of the LAS performance compared to the baseline (or its own performance with 8 and 10 layers) for a smaller number of layers (1, and 5). This provides a fairer comparison in terms of compute, and provides evidence to practitioners on whether this approach can be used in more resource costraiend environments.

**Ethical Concerns:**

["NO or VERY MINOR ethics concerns only"]

**Final Justification:**

For my considerations above, I find the paper merits publication. The authors engaged in a constructive rebuttal process with both me and the other reviews, and they address all the minor concerns that I pointed out above.

**Limitations:**

Limitations were not mentioned. Crucially, the authors failed to comment on the computational cost-accuracy tradeoff.

**Quality:**

4

**Strengths And Weaknesses:**

**Strengths**
- The paper is very well written, and very clear. Morevoer, while the main body is dense with results, both theoretical and empirical analysis, this does not affect the flow of the exposition.
- The authors have been methodic and thorough in their analysis: they develop a theoretical analysis under some assumptions, which are validated eitehr with references to prior work, or in the appendix. The main theorems are also assessed empirical to show additional evidence (e.g. Figure 3).
- The experimental evaluation of their model is similarly precise. Eight different PDE systems are assessed which comprise increasing complexities and a wide range of behaviors. The performance is systematically compared with four competing approaches.
- The experimental results prove the validity of the approach, point out the theoretical features of the algorithm described in the first half of the paper, and highlight the pitfalls of the competing algorithms.
- Extensive ablation studies are reported, and algorithm runtimes are provided.
- Finally, clear implementation details are provided, which allow other researchers to replicate and build on their work, and practictioners to use their methodology.

Overall, I am satisfied with the quality of this paper.

**Weaknesses**
- The authors fail to provide a clear paragraph covering the limitations of their proposed approach.
- While algorithm runtimes are reported in the appendix, these are not commented in the text. However, LAS is found to be more computationally expensive than the other approaches, except than L^\infty. Generally, LAS is 1.2 to 6 times slower, using 1 to 10 number of layers. However, results in the main body are only reported for 8 and 10 layers, which are significantly more expensive and more prone to out-of-memory (OOM) issues for high problem dimensions, as reported in Table 8. Note that OOM is recorded using a RTX 4090 GPU with 24GB memory, so for practitioners with more limited hardware, they may encounter memory bottlenecks much sooner.

---

> ### Author Rebuttal · Authors · 2025-07-28
>
> **Dear reviewer b6R2**
>
> We sincerely appreciate insightful your comments and positive evaluation of our work
>
> ---
>
> ## Response to Weaknesses 1: Lack of a clear paragraph discussing the limitations of the proposed approach
>
> Thank you for pointing this out. First, we would like to clarify that $l_{L}$​, which denotes the number of Langevin iterations used to update collocation points, was introduced in Algorithm 1 and Section 4.1. However, we acknowledge that this was not clearly mentioned in Appendix I, and we apologize for the confusion.
>
> Specifically, we analyzed the performance of LAS across different hyperparameter settings ($l_{L}$​, $\tau$, $\beta$) in Appendix D (pp. 21–22), with respect to different collocation point configurations in Appendix H.3 (p. 34), and computational costs in Appendix I (p. 36). Nevertheless, as you rightly noted, the main text lacks a proper discussion on the limitations and guidance for practitioners—particularly how to balance computational cost and hyperparameter tuning across layers and collocation points.
>
> We will revise the main text to include this discussion, focusing on the trade-off between performance and computational cost as the PDE dimensionality and number of collocation points increase. We will also provide guidance for practitioners on effectively applying LAS in resource-constrained environments.
>
> ---
>
> ## Response to Weakness 2: Algorithm runtimes are only reported in the appendix
>
> As you correctly observed, LAS becomes significantly slower with higher values of $l_{L}$ ​(e.g., from 1 to 10), except for the $L^{\infty}$. However, we also found that setting $l_{L} = 1$ is already sufficient to outperform competing methods in most cases, thereby keeping computational complexity comparable to others. All results in our main experiments were obtained with $l_{L} = 1$, and further performance comparisons for different $l_{L}$ values are provided in Appendix D (pp. 21–22). Nonetheless, as the number of collocation points and the problem dimensionality increase, LAS encounters memory bottlenecks earlier than methods like R3 and RAD. We will explicitly mention this limitation in the main text and recommend appropriate settings considering available memory. Of course, the number of layers affects the computation complexity, but difference of computational cost among sampling methods mainly comes from collocation points, PDE dimension, and repentance for sampling collocation points. However, we acknowledge that this was not clearly mentioned in Appendix I, and we apologize again for the confusion.
>
> ---
>
> ## Response to Question 1: Computational requirements
>
> We agree with your suggestion and will include a visual representation of the computational complexity and memory requirements for different GPU models. This will be shown across varying collocation point configurations and PDE dimensionalities.
>
> ---
>
> ## Response to Question 2: Performance for smaller numbers of layers (1 and 5)
>
> Thank you for this helpful suggestion. Performance in low-layer scenarios is particularly relevant in constrained environments.
>
> While fewer layers tend to experience less training instability, the comparative performance remains informative.
>
> In Figure 4(a), LAS achieved the second-best performance in the 4-layer setting and the best performance in the 10-layer case. Additionally, we include comparisons for the 4D heat equation with different balancing terms and with 1 and 5 layers. **Bold** indicates the best, while ***Bold + Italic*** denote the second-best alternative.
>
> | PDE               | Balance terms (IC/BC/PDE) | LAS              | Random-R         | RAD              | R3               | $L^{\infty}$      |
> |------------------|----------------------------|------------------|------------------|------------------|------------------|----------------------|
> | **DF-Heat 4D (1 layer)** | (100,1,1)              | **12.49 ± 3.86**  | 28.39 ± 5.90     | 44.14 ± 8.39     | ***18.17 ± 2.38***   | 22.53 ± 1.30         |
> |                  | (200,1,1)                  | **13.18 ± 2.56**  | ***23.35 ± 1.62***   | 44.55 ± 4.93     | 24.56 ± 3.20     | 22.96 ± 2.17         |
> |                  | (10,1,1)                   | ***34.88 ± 3.38***  | **26.43 ± 3.97**   | 63.67 ± 2.02     | 38.20 ± 5.71     | 48.23 ± 23.94        |
> |                  | (10,1,5)                   | **59.14 ± 9.39**  | ***70.91 ± 24.78***  | 100.00 ± 0.00    | 68.34 ± 6.02     | 100.00 ± 0.00        |
> |                  | (1,1,1)                    | ***82.51 ± 12.48*** | **50.33 ± 6.90**   | 100.00 ± 0.00    | 96.76 ± 4.72     | 96.17 ± 5.31         |
> | **DF-Heat 4D (5 layers)** | (100,1,1)           | ***1.68 ± 0.44***   | 5.42 ± 1.59      | 4.04 ± 1.25      | 4.27 ± 0.63      | **1.40 ± 0.33**        |
> |                  | (200,1,1)                  | ***1.67 ± 0.33***   | 5.41 ± 1.00      | 3.31 ± 1.36      | 3.82 ± 0.59      | **1.51 ± 0.13**        |
> |                  | (10,1,1)                   | **1.88 ± 0.45**   | 6.95 ± 0.94      | 4.74 ± 1.58      | 5.16 ± 0.57      | ***2.74 ± 1.68***        |
> |                  | (10,1,5)                   | **3.91 ± 1.28**   | 8.13 ± 1.37      | 11.70 ± 4.32     | 7.46 ± 2.26      | ***4.33 ± 1.93***        |
> |                  | (1,1,1)                    | **2.45 ± 0.44**   | 6.58 ± 1.09      | 9.36 ± 1.11      | 5.63 ± 0.68      | ***3.66 ± 2.16***        |
>
> ---
>
> | Method | Sampling Hyperparameters |
> |--------|----------------------------|
> | **LAS** | $\tau$=0.01, $\beta$=0.2, $l_L$=1 |
> | **Random-R** | NA |
> | **RAD** | k=1, c=1 |
> | **R3** | MaxIter=1 |
> | **$L^{\infty}$** | $\eta$=0.05, K=20 |
>
> ---
>
> **Training Hyperparameters**:
>
> Epochs=10,000, $N_{PDE}$=1,000, $N_{IC}$=100, $N_{BC}$=100, Learning Rate=1e-3, # of Hidden Layers=1, 5, # of nodes = 128, Optimizer: Adam
>
> ---
>
> We hope these responses address your concerns and that our planned revisions appropriately reflect your valuable suggestions. Please feel free to share any further comments or questions.
>
> Thank you again for your thoughtful review.

---

> > ### Comment · Reviewer_b6R2 · 2025-08-03
> >
> > I thank the Authors for their rebuttal. I appreciate thir willingness to include, in the revised version, a discussion of the computational limitations and the impact of a smaller number of layers, for which LAS still shows strong performance.

---

> > > ### Author Response · Authors · 2025-08-03
> > >
> > > Your comment helped us better highlight the strengths of the proposed LAS framework and reflect more carefully on the aspects that needed clarification. We sincerely appreciate your positive view of our work. Please don’t hesitate to reach out if you have any further questions or thoughts.

---

### Official Review · Reviewer_DvNh · 2025-07-02

**Clarity:** 3
**Significance:** 2
**Originality:** 2
**Rating:** 4
**Confidence:** 3

**Summary:**

In this paper, a new adaptive sampling method called LAS is proposed to enhance the accuracy of Physics-Informed Neural Networks (PINNs). LAS addresses the issue of neglecting points with medium or low residuals, which can compromise stability. The experiments demonstrate that the proposed method outperforms several existing approaches in terms of relative error and stability across a variety of environments, including high-dimensional partial differential equations (PDEs), where Monte Carlo integration-based methods typically struggle with instability.

**Questions:**

See weakness.

**Ethical Concerns:**

["NO or VERY MINOR ethics concerns only"]

**Final Justification:**

Thanks the authors' reply. I will increase my score.

**Limitations:**

More numerical comparision is needed,  see weakness.

**Paper Formatting Concerns:**

None.

**Quality:**

3

**Strengths And Weaknesses:**

engths: The paper is well-written, and the motivation is clearly articulated.

Weaknesses: The numerical analysis is insufficient to demonstrate the value of this paper, particularly due to the lack of comparisons with closely related works such as

[1] Failure-informed adaptive sampling for PINNs
[2] Failure-informed adaptive sampling for PINNs, Part II: Combining with Re-sampling and Subset Simulation
[3] A Gaussian Mixture Distribution-Based Adaptive Sampling Method for Physics-Informed Neural Networks

---

> ### Author Rebuttal · Authors · 2025-07-28
>
> **Dear reviewer DvNh**
>
> We sincerely appreciate your comments
>
> ---
>
> ## Response to Weakness: Lack of comparisons with closely related works
>
> As you correctly pointed out, we overlooked some closely related studies [1–3] that are highly relevant to our work. We will cite these works in _Related Work_ and explicitly discuss how they relate to both existing methods and our proposed approach.
>
> **[1] Failure-informed PINN (FI-PINNs):**
> In FI-PINNs, “failure” refers to collocation points where the PDE residual error exceeds a predefined threshold ​. A truncated Gaussian model for **Self-adaptive Importance Sampling (SAIS)** is employed to preferentially sample collocation points from these high-residual (failure) regions. However, this method suffers from two key limitations:
>
> 1. The number of collocation points increases over time as the failure regions are updated, which leads to additional computational cost.
>
> 2. It struggles to handle **multi-modal failure regions** effectively.
>
> **[2] Annealed R-FIPINN (Extensions of FI-PINNs):**
>
> To address these limitations, two important extensions were introduced:
>
> 1. **Cosine-annealing-based resampling technique**, which maintains a fixed training size while gradually adapting to failure regions.
>
> 2. **Subset Simulation (SS)** based on Markov Chain Monte Carlo (MCMC) using a modified Metropolis–Hastings (MMA) sampler, which replaces the truncated Gaussian model and better captures multi-modal failure regions.
>
> **[3] Gaussian mixture distribution-based adaptive sampling (GAS):**
> GAS improves the representation of multi-modal residuals by selecting the top NG​ high-residual validation points as GMM means and using the reciprocal of the residual gradient as the diagonal of the covariance matrix. However, similar to FI-PINNs, GAS continuously increases the total number of collocation points over time—despite maintaining the ratio between collocation and boundary points—which leads to progressively higher computational costs.
>
> We compared our **LAS** method against **Annealed R-FIPINN** and **GAS-T** under the same experimental conditions. To ensure fairness, we maintained the same $N_{BC}$​, $N_{IC}$​, $N_{PDE}$​, learning rate, and total epochs across all methods. Importantly, for GAS-T, we fixed the total number of collocation points by replacing existing collocation points at each update rather than aggregating them. **Bold** indicates the best, while ***Bold + Italic*** denote the second-best alternative.
>
>
> | PDE | Balance terms (IC/BC/PDE) | LAS | Random-R | RAD | R3 | $L^{\infty}$ | Annealed R-FIPINN | GAS-T |
> |-----|----------------------------|-----|-----------|-----|-----|--------------|---------------------|--------|
> | **DF-Heat 4D** | (100,1,1) | **1.72 ± 0.23** | 9.73 ± 0.44 | 6.64 ± 2.72 | 9.53 ± 5.27 | ***2.92 ± 1.95*** | 8.87 ± 0.65 | 3.80 ± 1.70 |
> |  | (200,1,1) | **1.82 ± 0.42** | 8.87 ± 1.67 | 5.72 ± 0.42 | 6.50 ± 1.19 | 4.06 ± 1.87 | 9.25 ± 0.70 | ***3.74 ± 2.43*** |
> |  | (10,1,1) | **1.79 ± 0.20** | 7.98 ± 37.59 | 24.98 ± 37.59 | 16.08 ± 3.25 | ***2.46 ± 0.67*** | 6.75 ± 1.97 | 7.60 ± 3.30 |
> |  | (10,1,5) | **2.85 ± 0.34** | 25.18 ± 37.84 | 72.53 ± 32.73 | 15.57 ± 9.22 | 42.71 ± 46.79 | 29.86 ± 36.25 | ***14.47 ± 8.07*** |
> |  | (1,1,1) | **1.86 ± 0.20** | ***7.73 ± 1.85*** | 12.31 ± 1.24 | 13.56 ± 5.24 | 10.95 ± 14.41 | 26.36 ± 36.89 | 16.59 ± 11.01 |
> | **DF-Heat 6D** | (100,1,1) | **3.49 ± 0.20** | 6.14 ± 0.42 | 13.84 ± 2.08 | 29.91 ± 35.23 | ***4.46 ± 0.42*** | 5.75 ± 1.05 | 100.00 ± 0.00 |
> |  | (200,1,1) | **4.18 ± 0.58** | 5.21 ± 0.82 | 10.63 ± 0.93 | 30.51 ± 34.90 | ***4.39 ± 0.61*** | 5.80 ± 0.72 | 37.20 ± 36.19 |
> |  | (10,1,1) | **4.79 ± 0.33** | 5.53 ± 0.81 | 31.81 ± 34.30 | 49.47 ± 41.28 | 26.37 ± 37.05 | ***5.49 ± 0.34*** | 100.00 ± 0.00 |
> |  | (10,1,5) | **7.33 ± 1.37** | 70.30 ± 36.59 | 100.00 ± 0.00 | ***69.37 ± 37.51*** | 90.21 ± 19.57 | 100.00 ± 0.00 | 100.00 ± 0.00 |
> |  | (1,1,1) | **5.20 ± 0.92** | *53.18 ± 46.82* | 100.00 ± 0.00 | 61.37 ± 39.22 | 100.00 ± 0.00 | 62.57 ± 45.83 | 100.00 ± 0.00 |
> | **DF-Heat 8D** | (100,1,1) | **7.72 ± 0.59** | 52.41 ± 39.33 | 84.71 ± 30.56 | 100.00 ± 0.00 | 100.00 ± 0.00 | ***49.22 ± 41.61*** | 100.00 ± 0.00 |
> |  | (200,1,1) | **6.92 ± 0.31** | 17.63 ± 1.78 | 83.59 ± 32.80 | 100.00 ± 0.00 | ***13.21 ± 4.46*** | 15.01 ± 2.69 | 100.00 ± 0.00 |
> |  | (10,1,1) | **7.97 ± 0.30** | 100.00 ± 0.00 | 100.00 ± 0.00 | 100.00 ± 0.00 | 100.00 ± 0.00 | 100.00 ± 0.00 | 100.00 ± 0.00 |
> |  | (10,1,5) | **11.17 ± 0.65** | 100.00 ± 0.00 | 100.00 ± 0.00 | 100.00 ± 0.00 | 100.00 ± 0.00 | 100.00 ± 0.00 | 100.00 ± 0.00 |
> |  | (1,1,1) | **9.08 ± 0.50** | 100.00 ± 0.00 | 100.00 ± 0.00 | 100.00 ± 0.00 | 100.00 ± 0.00 | 100.00 ± 0.00 | 100.00 ± 0.00 |
>
> ---
>
> | Method | Sampling Hyperparameters |
> |--------|----------------------------|
> | **LAS** | $\tau$=0.01, $\beta$=0.2, $l_L$=1 |
> | **Random-R** | NA |
> | **RAD** | k=1, c=1 |
> | **R3** | MaxIter=1 |
> | **$L^{\infty}$** | $\eta$=0.05, K=20 |
> | **Annealed R-FIPINN** | $\epsilon_r$=0.05, $\epsilon_p$=0.01, a=0.5, b=1, p=0.1 |
> | **GAS-T** | $N_G$=20, $N_a$=20, λ=1 |
>
> ---
>
> **Training Hyperparameters**:
>
> Epochs=10,000, $N_{PDE}$=1,000, $N_{IC}$=100/250/450 (4D/6D/8D), $N_{BC}$=100/250/450 (4D/6D/8D), Learning Rate=1e-3, # of Hidden Layers=8, # of nodes = 128, Optimizer: Adam
>
> ---
> We hope these responses have addressed your concerns and that our planned revisions clearly reflect your valuable suggestions. Please don’t hesitate to share any further feedback or questions.
>
> Thank you once again for your thoughtful and constructive comments.

---

> > ### Comment · Reviewer_DvNh · 2025-08-08
> >
> > Thanks the authors' reply. I will increase my score.

---

> > > ### Author Response · Authors · 2025-08-08
> > >
> > > Thank you for taking the time to review our work. We appreciate your feedback.

---

> ### Author Response · Authors · 2025-08-05
>
> We kindly ask whether our rebuttal has addressed your concerns, and we would be grateful for any further comments or clarifications you might have. Thank you again for your time and effort in reviewing our work.

---

### Official Review · Reviewer_3GbE · 2025-07-05

**Clarity:** 4
**Significance:** 3
**Originality:** 3
**Rating:** 5
**Confidence:** 3

**Summary:**

This paper proposes to use Langevin dynamics to update collocation points when training PINNs. The authors analyzed the gradient behaviors for training with high residual points and training with random samples, and showed that there could be big difference between these two approaches regarding convergence and stability. The authors consequently proposed to update the collocation points locally via Langevin dynamics, which may favor flat regions and enjoy better training stability. The proposed method is tested on several 1D PDEs as well as high-dimension PDEs.

**Questions:**

- What is $a(\theta)$ in equation 3.2?

**Ethical Concerns:**

["NO or VERY MINOR ethics concerns only"]

**Final Justification:**

The authors clarified some derivation and experimental evaluations. For high-dimensional problems, although the authors explained that they have tried reasonable hyperparameters, I believe getting improved results for baseline methods would further improve the presentation. Overall I will keep my rating as accept.

Update:

The updated presentation about the high-dimensional results addressed my concern about baseline results. I remain supportive for acceptance.

**Limitations:**

- Limitation is discussed adequately.

**Paper Formatting Concerns:**

None.

**Quality:**

3

**Strengths And Weaknesses:**

Strength

- The paper is well written and easy to follow, despite having certain theoretical depth.

- The paper empirically benchmarked several collocation sampling methods, which are tested on both simple and high-dimensional PDEs.

Weakness

- Although the overall experimental results show favorable performance for the proposed method. I feel the gain in the performance seems not very strong. For example, as shown in Table 1, for some PDEs, random sampling still yields the best results. The results for high-dimensional PDEs are significantly better for the proposed method. However, the error seems to increase with the dimensionality, and the benchmarked existing methods do not really work and they may not be tailored for such problems.

- Nevertheless, although I am not expert on tuning PINNs, I think the proposed Langevin dynamics sampling is an interesting idea and its motivation is well analyzed.

---

> ### Author Rebuttal · Authors · 2025-07-28
>
> **Dear Reviewer 3GbE,**
>
> We sincerely appreciate your interest in our work and your positive evaluation.
>
> As you correctly pointed out, we observed that **Random-R** (uniform random sampling at each iteration) generally performs well when using a sufficiently large number of collocation points (in our case, $N_{PDE}$=1000) for relatively smooth PDEs, such as Burgers’, Schrödinger, and KdV equations. However, for **stiff PDEs** such as the Allen–Cahn equation, all adaptive sampling methods consistently outperformed Random-R. Moreover, we found that existing benchmark adaptive sampling methods struggled to generalize across different PDE types—smooth, stiff, and high-dimensional (although we did not exhaustively search the entire hyperparameter space, we explored reasonable hyperparameter configurations as reported in Appendix H, pp. 33–34). In contrast, our proposed **LAS** method demonstrated consistently generalized performance across different PDEs by simultaneously prioritizing both high-residual and relatively flat regions, which was particularly effective in **high-dimensional cases**.
>
> ---
>
> ## Response to Q1: What is a($\theta$) in Equation (3.2)?
>
> We thank the reviewer for asking for clarification regarding the definition of $a(\theta)$.
>
> In our formulation, $a(\theta)$ is the coefficient vector that arises from linearizing the residual function $R_\theta(\mathbf{x})$ with respect to a fixed basis of differential features, denoted $\phi(\mathbf{x})$. Specifically, when the residual is expressed as
> $$
> R_\theta(\mathbf{x}) = (\partial_t f_{\theta} + N_x[f_\theta])(\mathbf{x}) \triangleq g(f_\theta)(\mathbf{x}) \approx a(\theta)^\top \phi(\mathbf{x}),
> $$
> the vector $a(\theta)$ contains the weights applied to each feature in this expansion.
>
> This structure is motivated by a first-order linearization of the operator $g$ in function space. If the residual operator admits a smooth form such as
> $$
> g(f) = G(f, \partial_t f, \partial_x f, \partial_{tt} f, \partial_{tx} f, \dots),
> $$
> then applying the Fréchet derivative yields:
> $$
> g(f_{\theta + \Delta\theta}) \approx g(f_\theta) + D_g(f_\theta)(f_{\theta + \Delta\theta} - f_\theta)
> $$
> Using a Taylor expansion in terms of the partial derivatives of $f_\theta$, the residual can be approximated as:
> $$
> R_\theta(\mathbf{x}) \approx \sum_k \frac{\partial G}{\partial (\partial_k f_\theta)} \Delta(\partial_k f_\theta) \equiv a(\theta)^\top \phi(\mathbf{x})
> $$
> Hence, $a(\theta)$ is not an independently learned parameter, but rather a structured function of the network $f_\theta$ and its derivatives with respect to both input and parameters.
>
> To ground this in concrete examples:
>
> - **Convection equation**: $g(f) = \partial_t f + \mu \partial_x f$ → $a(\theta) = (0, 1, \mu, 0, 0, \dots)$ is constant
> - **Burgers’ equation**: $g(f) = \partial_t f + f \partial_x f - \frac{0.01}{\pi} \partial_{xx} f$ → $a(\theta) = (\partial_x f_\theta, 1, f_\theta, 0, \dots, \frac{0.01}{\pi})$ is input- and $\theta$-dependent
>
> A more detailed description is provided in Appendix A.1. We acknowledge that the explanation of this coefficient vector was missing in the current manuscript, and we will include this clarification in the revised version.
>
> We hope these responses address your concerns and that our planned revisions appropriately reflect your valuable suggestions. Please feel free to share any further comments or questions. Thank you again for your thoughtful comments.

---

> > ### Author Response · Authors · 2025-08-05
> >
> > We hope our rebuttal has addressed your concerns, and we would be grateful for any further comments or clarifications you may wish to share. Thank you again for your constructive review.

---

> > > ### Comment · Reviewer_3GbE · 2025-08-06
> > >
> > > The authors clarified derivations and experimental evaluations. For high-dimensional problems, although the authors explained that they have tried reasonable hyperparameters, I believe getting improved results for baseline methods would improve the presentation.

---

> ### Author Response · Authors · 2025-08-06
>
> We sincerely thank the reviewer for their thoughtful comment and for recognizing our clarifications and experimental evaluations.
>
> Below, we summarize how we searched for reasonable hyperparameter configurations for the baseline methods:
>
> **RAD**
>
> - Default setting in the original paper: ($k$ = 1, $c$ = 1)
> - Current settings in our work: (fixing $c$ =1, this is for uniformity), (different $k$ = 0.5, 1, 2, 3, 4, 5)
> - As $k$ decreases, the sampling becomes uniform; as $k$ increases, the sampling increasingly focuses on high-residual regions
>
> **R3**
>
> - Default setting in the original paper: (max_iteration = 1)
> - Current settings: (max_iteration = 1, 3, 5, 10, 15, 20)
> - Increasing the maximum iteration leads to more aggressive focusing on high-residual areas
>
> **$L^{\infty}$**
>
> - Default setting in the original paper: (step_size = 0.05, inner_iteration = 20)
> - Current: (step_size = 0.05, inner_iteration = 20),  See p.36 for details on the computational cost of $L^{\infty}$
> - The step size controls the movement of collocation points, and increasing the number of inner iterations intensifies focus on high-residual regions
>
> **Common hyper-parameters**:
> - Epochs=10,000, $N_{PDE}$=1,000, $N_{IC}$=100/250/450 (4D/6D/8D), $N_{BC}$ =100/250/450 (4D/6D/8D), Learning Rate=1e-3, # of Hidden Layers=8, # of nodes = 128, Optimizer: Adam
>
> We have conducted extensive hyperparameter tuning for the baseline methods, including adjustments to various loss balancing terms, particularly in high-dimensional settings.
> While we acknowledge that further improvements to the baseline performance could enhance the overall presentation, we believe the current experimental setup offers a fair and representative comparison across all methods.
>
> ---
>
> On the one hand, we reported the performance of all methods using a **fixed loss balance term of 1/1/1 (IC/BC/PDE)** under high-dimensional settings in Table 1 of the main text.
> We will update Table 1 with the best performance of each method based on the corresponding hyperparameter settings described in Appendices H.1 and H.2.
>
> ---
>
> Below, **Bold** indicates the best, while ***Bold + Italic*** denote the second-best alternative.
>
> ---
>
> ## Before
>
> (8 layers in the main body)
>
> | **PDE**         | **LAS**             | **Random-R**        | **RAD**             | **R3**              | **$L^{\infty}$**     |
> |----------------|---------------------|----------------------|---------------------|---------------------|----------------------|
> | DF-Heat 4D     | **1.86 ± 0.20**     | ***7.73 ± 1.85***        | 12.31 ± 1.24        | 13.56 ± 5.24        | 10.95 ± 14.41        |
> | DF-Heat 6D     | **5.20 ± 0.92**     | ***53.18 ± 46.82***      | 100.00 ± 0.00       | 61.37 ± 39.22       | 100.00 ± 0.00        |
> | DF-Heat 8D     | **9.08 ± 0.50**     | 100.00 ± 0.00        | 100.00 ± 0.00       | 100.00 ± 0.00       | 100.00 ± 0.00        |
>
> ---
>
> ## After
>
> (8 layers in the main body, reflecting the best performances with different loss balance term in Appendix H.1 and H.2.)
>
> | **PDE**         | **LAS**             | **Random-R**        | **RAD**             | **R3**              | **$L^{\infty}$**     |
> |----------------|---------------------|----------------------|---------------------|---------------------|----------------------|
> | DF-Heat 4D     | **1.72 ± 0.23**     | 7.73 ± 1.85          | 5.72 ± 0.42         | ***2.15 ± 0.25***       | 2.46 ± 0.67        |
> | DF-Heat 6D     | **3.49 ± 0.20**     | 5.53 ± 0.81          | 6.39 ± 1.14         | 4.98 ± 0.18         | ***4.39 ± 0.61***        |
> | DF-Heat 8D     | **6.92 ± 0.31**     | 17.63 ± 1.78         | 67.50 ± 39.83       | 65.03 ± 43.11       | ***13.21 ± 4.46***       |
>
> ---
>
> For the best-performing cases, the detailed parameter settings are as follows:
>
> **Loss balance term settings (IC, BC, PDE)** $\in$ \{$(1, 1, 1), (10, 1, 5), (10, 1, 1), (100, 1, 1), (200, 1, 1)$\}.
>
> LAS: 4D (100, 1, 1), 6D (100, 1, 1), 8D (200, 1, 1)
>
> Random-R: 4D (1, 1, 1), 6D (200, 1, 1), 8D (200, 1, 1)
>
> RAD: 4D (200, 1, 1), 6D (200, 1, 1), 8D (200, 1, 1)
>
> R3: 4D (200, 1, 1), 6D (100, 1, 1), 8D (200, 1, 1)
>
> $L^{\infty}$: 4D (10, 1, 1), 6D (200, 1, 1), 8D (200, 1, 1)
>
> **Hyper-parameters settings for each method**
>
> LAS: 4D, 6D, 8D ($\beta = 0.2$, $\tau = 0.01$, $I_{L} = 1$)
>
> Random-R: -
>
> RAD: 4D ($k$=1, $c$=1), 6D ($k$=0.5, $c$=1), 8D ($k$=0.5, $c$=1)
>
> R3: 4D (max_iter=3), 6D (max_iter=3), 8D (max_iter=15)
>
> $L^{\infty}$: 4D, 6D, 8D (step_size = 0.05, inner_iter = 20)
>
> ---
>
> Last but not least, Appendix H clearly shows that LAS is much **less sensitive to the choice of loss balance terms** when converging to strong performance, compared to existing baselines. We believe these results provide strong evidence that LAS is robust across a variety of parameter settings. We hope this addresses your concern.
>
> Please feel free to share any further questions or feedback.

---

### Note · Authors · 2025-08-12

We sincerely appreciate the constructive feedback exchanged during this rebuttal process, which has significantly enhanced the overall quality of our paper.

---

**Reviewer 3GbE** provided valuable feedback in clarifying the explanation of Equation (3.2) and improving the performance comparison across different sampling methods. We will incorporate the following changes:

1. Expand the explanation of Equation (3.2) in the main paper, which was previously discussed primarily in Appendix A.1, to improve readability and clarity.
2. Based on Appendix H.1 and H.2, explicitly mention in the main text that we conducted hyperparameter searches for all baselines, and update the performance comparison table in the main paper with the tuned results.

---

**Reviewer DvNh** helpfully pointed out the existence of related work, including Failure-informed PINNs (FI-PINNs) and Gaussian mixture distribution-based adaptive sampling (GAS), both of which are closely related to LAS. This will help refine the Related Works section:

1. Update the Related Works section to include these methods and explicitly describe their connections to LAS.
2. In the Appendix, provide hyperparameter settings and performance comparisons for Annealed R-FIPINN and GAS-T.

---

**Reviewer b6R2** provided insightful comments on comparing LAS with existing methods in terms of computational complexity, identifying potential limitations, and clarifying the computational requirements for practical use in resource-constrained environments. We will make the following updates:

1. In the main paper, add a discussion on LAS’s computational limitations compared to existing methods (e.g., cost growth with increasing $l_L$) and detail the computational requirements for practical deployment.
2. Include additional results in the Appendix comparing performance on small networks.

---
**Reviewer Jy8K**’s feedback was instrumental in further strengthening the theoretical foundations of our work. Specifically, we will:

1. Address technical issues in the continuous-time approximation of Langevin dynamics (e.g., residual blow-up, gradient smoothness) and clarify the proof steps in the main paper.
2. Revise Assumption 3.1 in Appendix A.1 with more reader-friendly examples, and update Propositions 3.1 and 3.2 accordingly.
3. Add a remark on cases where Assumptions 3.2 and 3.3 may conflict, explaining how such situations naturally arise in our setup (e.g., concentrated collocation points, high-dimensional networks).

---

### Decision · Program_Chairs · 2025-09-17

**Decision:**

Accept (spotlight)

**Comment:**

The paper presents an adaptive collocation point sampling method for physics-informed neural networks (PINNs). The authors carefully analyze existing high-residual adaptive sampling approaches and point out that they can lead to stability issues. Their proposed Langevin dynamics–based method is shown to be robust across a range of learning rates and model complexities.

All reviewers agree that this is a solid, well-motivated, and clearly written paper. The numerous interactions during the rebuttal provided valuable additional results—both theoretical and numerical—and the reviewers acknowledge that their concerns have been satisfactorily addressed. I therefore recommend acceptance.

I strongly encourage the authors to integrate the rebuttal content, in particular the newly updated proofs and discussions, into the camera-ready version of the paper.